# Integrated proteomic and transcriptomic landscape of macrophages in mouse tissues

Jingbo Qie[1,8], Yang Liu[1,8], Yunzhi Wang [1,8], Fan Zhang[1,8], Zhaoyu Qin [1], Sha Tian [1], Mingwei Liu [2], Kai Li [2], Wenhao Shi[2], Lei Song [2], Mingjun Sun[1], Yexin Tong[1], Ping Hu [3], Tao Gong[4], Xiaqiong Wang[4], Yi Huang[4], Bolong Lin [4], Xuesen Zheng[4], Rongbin Zhou [4], Jie Lv[5], Changsheng Du[5], Yi Wang[2,6], Jun Qin[1,2,6], Wenjun Yang[3] ✉, Fuchu He[1,2,7] ✉ & Chen Ding [1,2] ✉

Macrophages are involved in tissue homeostasis and are critical for innate immune responses, yet distinct macrophage populations in different tissues exhibit diverse gene expression patterns and biological processes. While tissue-specific macrophage epigenomic and transcriptomic profiles have been reported, proteomes of different macrophage populations remain poorly characterized. Here we use mass spectrometry and bulk RNA sequencing to assess the proteomic and transcriptomic patterns, respectively, of 10 primary macrophage populations from seven mouse tissues, bone marrow-derived macrophages and the cell line RAW264.7. The results show distinct proteomic landscape and protein copy numbers between tissue-resident and recruited macrophages. Construction of a hierarchical regulatory network finds cell-type-specific transcription factors of macrophages serving as hubs for denoting tissue and functional identity of individual macrophage subsets. Finally, Il18 is validated to be essential in distinguishing molecular signatures and cellular function features between tissue-resident and recruited macrophages in the lung and liver. In summary, these deposited datasets and our open proteome server (http://macrophage.mouseprotein.cn) integrating all information will provide a valuable resource for future functional and mechanistic studies of mouse macrophages.

Macrophages are myeloid-lineage cells that participate in the innate immune response and populate niches or territories in multiple organs, serving as auxiliary cells for tissue development and homeostasis and possessing a broad spectrum of immune- and non-immune-related tissue-supporting activities[1,2]. Lineage-tracing and fate-mapping studies performed in the C57BL/6 mouse strain have indicated that tissue macrophage populations develop throughout life in a manner dependent on macrophage colony-stimulating factor and

[1]State Key Laboratory of Genetic Engineering, Institutes of Biomedical Sciences, Human Phenome Institute, School of Life Sciences, Zhongshan Hospital, Fudan University, Shanghai 200433, China. [2]State Key Laboratory of Proteomics, Beijing Proteome Research Center, National Center for Protein Sciences, 102206 Beijing, China. [3]Department of Pediatric Orthopedics, Xin Hua Hospital Affiliated to Shanghai Jiao Tong University, School of Medicine, Shanghai 200092, China. [4]Hefei National Laboratory for Physical Sciences at Microscale, The CAS Key Laboratory of Innate Immunity and Chronic Disease, School of Life Sciences, University of Science and Technology of China, Hefei 230027, China. [5]Putuo District People's Hospital, Shanghai Key Laboratory of Signaling and Disease Research, School of Life Sciences and Technology, Tongji University, Shanghai 200092, China. [6]Alkek Center for Molecular Discovery, Verna and Marrs McLean Department of Biochemistry and Molecular Biology, Department of Molecular and Cellular Biology, Baylor College of Medicine, Houston, TX 77030, USA. [7]Research Unit of Proteomics Driven Cancer Precision Medicine, Chinese Academy of Medical Sciences, Beijing 102206, China. [8]These authors contributed equally: Jingbo Qie, Yang Liu, Yunzhi Wang, Fan Zhang. ✉e-mail: wjyang@sibcb.ac.cn; hefc@nic.bmi.ac.cn; chend@fudan.edu.cn

the CSF1/CSF1R signaling axis in three waves[3]: this development is initiated with phagocytes derived from a transient hematopoietic wave of erythro-myeloid progenitors (EMPs) in the yolk sac, followed by EMP-derived fetal monocytes seeded in the fetal liver, and finally, the blood monocyte wave derived from hematopoietic stem cells (HSCs)[4,5].

Tissue-resident macrophages seeded in different tissues have been found to exhibit disparate functions and gene expression profiles[2,6]. For example, microglia are the only true central nervous system parenchymal macrophages and play important roles in synapse modulation, synapse pruning, and neurogenesis[7,8]. Alveolar macrophages in the lungs are responsible for surfactant clearance, which is important in the alveolar space[9]. Adipose macrophages are involved in regulating insulin resistance and adaptive thermogenesis, buffering the concentrations of locally required substrates, such as catecholamines, lipids, and iron[10]. Even within a certain tissue, macrophage populations, especially for tissue-resident and recruited macrophages, are highly heterogeneous with specific marker expression, as revealed by single-cell proteome or transcriptome (single-cell RNA-sequencing, scRNA-seq), mass cytometry, and epigenetic studies[11–14]. These findings highlight the particular plasticity of macrophages and the close relationship between the environmental niche and macrophage heterogeneity, especially in the postnatal period. Circulating monocytes were demonstrated to be recruited and occupy the macrophage pool after the loss of resident macrophages in a CSF1R-dependent manner[15], and the heterogeneity among monocyte-derived macrophages within a certain tissue is well recognized[16]. However, a systematic comparison between tissue-resident and recruited macrophages has not been reported.

Mechanistically, the heterogeneity in the cellular functions and expression profiles of macrophages depends on precise transcriptional regulation[6]. Lineage-determining transcription factors (LDTFs), such as PU.1 and C/EBPs, are ubiquitously expressed in macrophages in different tissues[17]. The cell-type-specific transcription factors (TFs) of macrophages, such as GATA6 in peritoneal macrophages[18], SALL1[19] in microglia, and PPARg in lung macrophages[20], can work coordinately with LDTFs to determine macrophage functions[21]. Recently, a large-scale meta-analysis based on network clustering of 466 available mouse RNA-sequencing (RNA-seq) datasets systematically elucidated the transcriptional atlases of mononuclear phagocyte system (MPS) cells[22], providing an overview of the TF expression profile of macrophages at the transcriptional level.

Omic studies provide a rapid and systematic way to identify the integral heterogeneity of macrophages. Emerging pieces of evidence from epigenomic and transcriptomic studies focusing on macrophage heterogeneity have generally demonstrated the importance of environmental effects on tissue-resident macrophage specialization[21–23]. Previous studies on cell type-resolved liver, brain, and heart proteomes[24–26] have emphasized the poor correlation between mRNA and protein profiling, revealing the importance of post-transcriptional processes that regulate protein synthesis and degradation. Therefore, an analysis of the heterogeneity of the macrophage proteome is required, but such a research effort has not yet been undertaken to the best of our knowledge. Fortunately, the rapid development of high-resolution mass spectrometry (MS) for proteome analysis[27] and of a bioinformatics platform[28] has allowed researchers to achieve in-depth coverage of protein expression in various cell types.

Here, integrated proteomic and transcriptomic analysis on ten primary mouse tissue macrophages and bone marrow-derived macrophages (BMDMs) uncover the diverse gene expression features and specific cellular functions of different macrophage populations, relating to the physiological activities of the tissues in which the macrophages resided. The TF-centered hierarchical crosstalk networks between macrophages and the relevant tissues provide insights into the mechanism underlying macrophage heterogeneity. Further

proteomic analysis reveals the significant differences between tissue-resident and recruited macrophages, with Il18 being validated as an essential molecule in distinguishing them, especially in the lung and liver. Collectively, our datasets provide a valuable resource and an opportunity to view the macrophage regulatory network, especially for the TF and microenvironment regulation in orchestrating macrophage identity.

## Results

### The proteome and transcriptome atlases of ten primary macrophage populations, BMDMs, and RAW264.7 cells

To probe the global spectrum of protein expression profiles among tissue-resident macrophages in homeostasis, we isolated seven primary tissue-resident macrophage populations including brain microglia, lung alveolar macrophages, liver Kupffer cells, splenic red pulp macrophages, peritoneal macrophages, and small and large intestinal macrophage populations from C57BL/6N mice (8–12 weeks old) through enzymatic digestion and fluorescence-activated cell sorting (FACS). To identify global differences between tissue-resident and recently recruited macrophages, we isolated three tissue-recruited macrophage populations in the lung, liver, and spleen simultaneously. A total of 13 surface markers, including CD45, F4/80, CD11b, CD117, Siglec-F, CD11c, Cx3cr1, MHCII, CD64, CD115, CD24, B220, and Ly6g, were used for cell sorting according to the published literature (Supplementary Fig. 1a and Supplementary Tables 1 and 2)[1,8,29]. F4/80 and CD11b bright and dim phenotypes were used to distinguish tissue-resident and recruited populations in the lung, liver, and spleen (Supplementary Fig. 1a). Evidence obtained by back-testing through flow cytometry suggested that the average purity of the macrophage populations was 98% (Supplementary Fig. 1b). However, it is worth noting that the possible contamination due to factors such as phagocytosis activity of macrophages and encapsulation of other cells by macrophage remnants during tissue digestion were inevitable. The cell line RAW264.7 and BMDMs were also collected, as these populations, although maintained in culture, are widely used as immortal cell lines or models of macrophage biology (Fig. 1a).

For each macrophage population, equal numbers of cells (1.5E6) were subjected to LC-MS/MS analysis using a high-resolution mass spectrometer (Orbitrap Fusion Lumos) after tryptic digestion of cell lysates and fractionation of the resulting peptides into six fractions through high-pH reversed-phase liquid chromatography (RPLC). The raw MS files were processed with the Firmiana[28] platform for protein identification and quantification based on the Mascot search engine against the NCBI murine Refseq protein database. Peptides (minimum length of seven amino acid residues) with 1% FDR and a Mascot ion score greater than 20 were selected for protein identification. Then, proteins with 1% FDR (with at least one unique peptide) were selected for further analysis. All identified peptides were quantified with the area under the curve (AUC) of a peptide feature. The intensity-based absolute quantification (iBAQ) algorithm[30] and "proteomic ruler"[31] methods were employed for protein quantification and protein copy number calculation (Methods).

As a result, a total of 12,205 proteins were identified, with more than 7000 proteins being identified in every single replicate and an average of 8000 proteins being discovered in each macrophage population (Fig. 1b and Supplementary Data 1). The identified proteins in the three replicates for each macrophage population showed high overlap (Supplementary Fig. 2a), and the average Pearson correlation coefficient $r$ of the biological replicates was greater than 0.9 (Supplementary Fig. 2c). Proteins that are critical for macrophage development or serve as identity markers, such as Sfpi1 (PU.1), Itgam (CD11b), Adgre1 (F4/80), Myd88, and Mertk, were found to be highly or moderately abundant (Supplementary Fig. 3).

To compare gene expression diversities at the protein and transcript levels, we carried out RNA-seq analysis on the 12

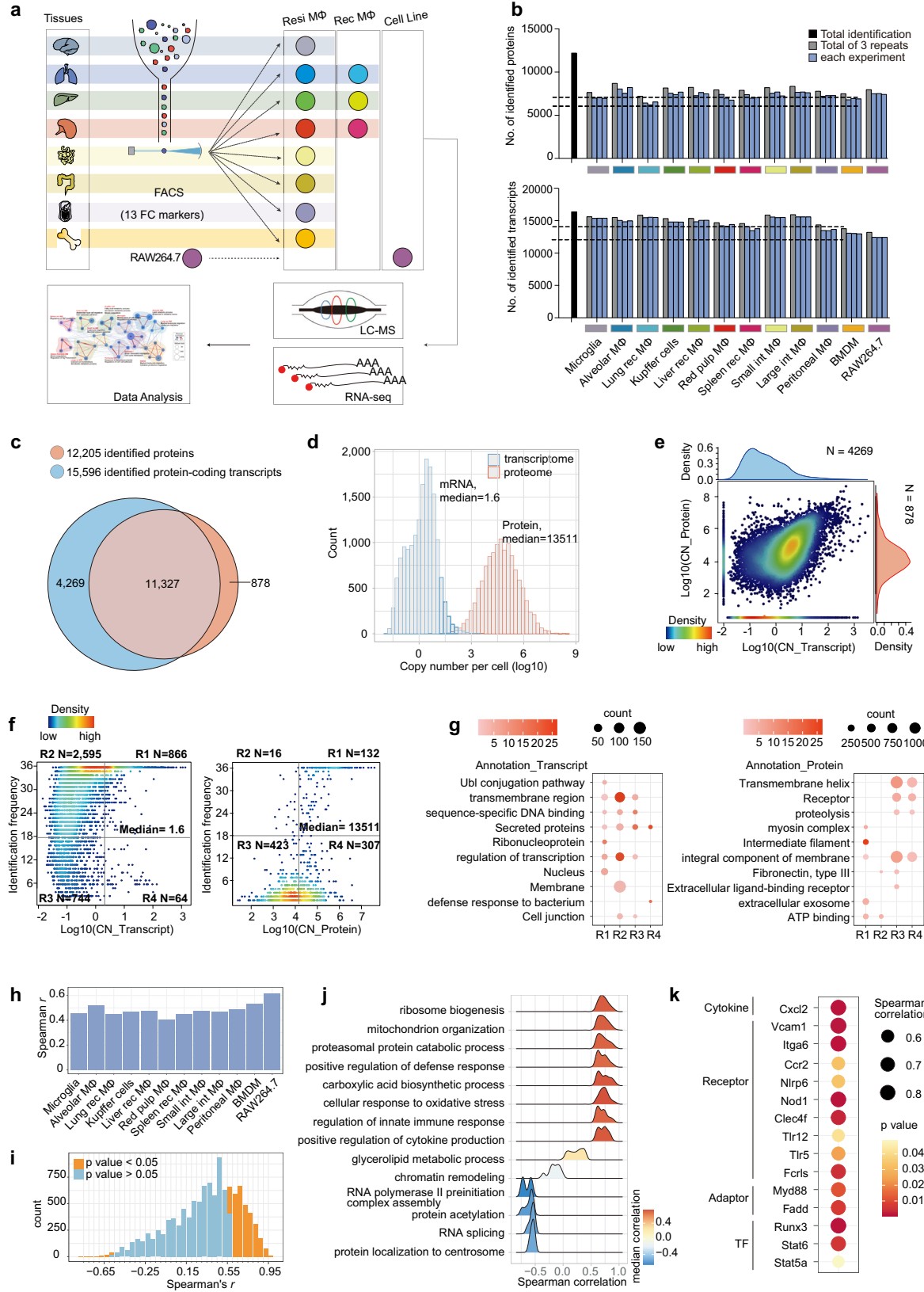

abovementioned macrophage populations with three parallel biological replicates (Fig. 1a). The transcriptome analysis identified 12,690 to 15,247 protein-coding genes with more than one fragment per kilobase of exon model per million fragments mapped (FPKM) for one population, leading to 15,596 protein-coding genes being

detected with high repeatability (Fig. 1b and Supplementary Fig. 2b, d and Supplementary Data 2).

We then systematically compared our transcriptome data with a published large-scale meta-data (Methods), in which a total of 466 RNA-seq libraries were well integrated[22]. As a result, the spearman

**Fig. 1 | The proteomic and transcriptomic atlases of ten primary macrophage populations from seven tissues, BMDMs, and RAW264.7 cells. a** Experimental design and workflow for determining the mouse macrophage proteome and transcriptome. **b** Numbers of identified proteins in LC−MS/MS measurements and protein-coding genes in RNA-seq analysis of the 12 macrophage populations. **c** Venn diagram of the identified gene numbers at the protein (red) and mRNA (blue) levels among the 12 macrophage populations. **d** Distributions of the macrophage proteomic (red) and transcriptomic (blue) landscapes based on the mean values of estimated copy number values across the 12 macrophage populations. **e** Density scatterplot of protein intensities versus mRNA intensities, based on the mean values of copy numbers across the 12 macrophage populations. **f** Density distribution for the transcripts of 4269 missing proteins (left) and the protein products of 878 missing mRNA transcripts (right) according to their average expression values and identification frequencies across the 12 populations. The genes were divided into four groups based on the defined cut-off (median value of the protein expression and half the identification frequencies). **g** Representative GOBP/KEGG functional annotations for genes in the four regions in **f** (one-sided Fisher's exact test, $p < 0.05$). The dot size represents the number of proteins involved in the relevant term. The color bar indicates the enrichment significance. **h** Histogram of Spearman correlation coefficients between the average copy numbers for the triplicate proteome and triplicate transcriptome data for each of the 12 macrophage populations (two-sided Spearman's rank correlation test, $p < 0.05$). **i, j** Histogram (**i**, two-sided Spearman's rank correlation test, $p < 0.05$) and GOBP enrichment (**i**, one-sided Fisher's exact test, BH FDR < 0.05) for high or low gene-wise RNA-to-protein correlations. **k** Bubble plot of the immune-related proteins with a significantly high RNA-to-protein correlation (two-sided Spearman's rank correlation test, $p < 0.05$). Source data are provided as a Source Data file.

correlation coefficients between the two datasets of the same macrophage population were around 0.75 (*p* value <0.05), which was significantly higher than the correlation coefficients between the two datasets of different cell types (Supplementary Fig. 4a and Supplementary Data 3). Notably, large intestinal macrophages served as an exception, which may be due to the fact that the large intestinal macrophages in the meta-data analysis were derived from monocyte-transferred populations in a CD11c-DTR mouse model (BioProject ID: PRJNA591465)[32], resulting in the poor correlation between the transcriptome of the samples and transcriptome of all macrophage populations from normal mice in our study. Similarly, the transcriptomes of microglia derived from bone marrow (BM) transfer (BioProject ID: PRJNA506249)[33]/hybridized mice (BioProject ID: PRJNA529095)[34], or the transcriptomes of small intestinal macrophages derived from monocyte transfer (BioProject ID: PRJNA591465)[32]/a conjoined mouse model (BioProject ID: PRJNA325288)[35], were significantly less correlated with transcriptomes of macrophage populations derived from normal C57BL/6N mice in our study (Supplementary Fig. 4b). We also compared published macrophage proteomes[24,25,36] and our proteome datasets, and found that the average correlation coefficient between the data for the same macrophages was 0.7, which was significantly higher than that for different macrophages (Supplementary Fig. 4c). Through comparing the quantification values of the signature genes among the published meta-data, our transcriptome and proteome data, we discovered good consistency of the expression of cell-type specific genes among the three datasets (Supplementary Fig. 4d, e). Furthermore, the expression levels of some potential core signatures of different macrophages, such as Mef2a, Mef2c, Mef2d, and Sall1 in microglia; Pparg, Stat5a, and Stat6 in alveolar; Rxra, Nr1h3, and Vcam1 in Kupffer cells; and Gata6, Itga6, and Tgfb2 in peritoneal macrophages, showed high consistency across published epigenome and transcriptome[37], as well as our proteome (Supplementary Fig. 4f).

The results indicated that our transcriptomic and proteomic data captured the characteristic profiles of different macrophage populations, providing an opportunity for comparisons between the proteome and transcriptome of mouse macrophage populations.

## Comparison of the proteomic and transcriptomic data of 12 macrophage populations

Integrated analysis of our proteomic and transcriptomic datasets revealed high overlap in identifications. A total of 11,327 out of 15,596 protein-coding transcripts were detected at the protein level, representing deep coverage of the proteomic dataset (Fig. 1c). We found that macrophage transcript copies spanned approximately five orders of magnitude and were on average ~8000-fold lower than the protein copy numbers (Fig. 1d). Consistently, previous studies on the cell-type-resolved liver proteome also indicated a comparatively lower copy number for transcripts than for proteins in various cell types[24].

To systematically determine the differences between the transcriptome and proteome in gene identification and quantification, we compared the expression levels of co-identified and exclusively identified genes between two datasets. As shown in Fig. 1e, low-abundance transcripts (less than 1 copy per cell) were often under-detected at the protein level (accounting for more than 78% of missing proteins), including a large number of olfactory receptors that were not expected to be functionally relevant to macrophage populations (Supplementary Data 2). However, we also found some missing proteins whose transcripts have higher copy numbers (e.g., above 10). To explore the reasons for the missing detection of proteins, we divided the missing proteins into four groups according to their average expression values (cut-off: 1.6) and identification frequencies (cut-off: 18, half the number of MS experiments) at the transcriptomic level (Fig. 1f). Through gene annotation analysis (Fig. 1g), we found the genes with high quantification values and identification frequencies were associated with ubiquitination modification, transcriptional regulation, ribonucleoproteins, transmembrane proteins and secretory proteins. The proteins encoded by these genes are difficult to be identified by MS due to their biological function characteristics[38], physicochemical properties (half-life)[39] or specific cellular localization. Sixty-four genes in the R4 region (Fig. 1f) represent specifically expressed transcript, most of which are secreted proteins or associated with specific immune responses, such as Mup family genes identified in the Kupffer cell transcriptome and Defa family genes in the gut (Supplementary Fig. 4h).

In parallel, we also analyzed the proteomic properties of missing transcripts with the same strategies (Fig. 1f, g). The results showed that proteins encoded by 16% of the missing transcripts were identified with high frequency (cut-off: 18), and were significantly enriched in cytoskeleton-related functions. Previous studies have shown that as structural proteins, cytoskeletal proteins are stable, while the half-lives of the corresponding transcripts are short[39]. Unlike the missing proteins, most missing transcripts (84%) were moderately expressed at the protein level with low identification frequencies. Among these moderately quantified proteins without transcripts signals (Supplementary Fig. 4g), we noticed that (1) ~35% of proteins were identified only once or sporadically in different cells, reflecting the transient protein expression or noise in the MS measurement.; (2) half of the proteins were specifically identified in one or two populations, such as Cbln2/3/4 in microglia and the Slc proteins in Kupffer cells or intestinal macrophages, may indicating the phagocytic ability of macrophages in relevant tissue. The enrichment of the protein products of some missing transcripts in exosomes also suggested the phagocytic activity of macrophages (Supplementary Fig. 4h).

We then surveyed both the across-gene and within-gene correlations of 11,327 co-identified genes in the transcriptome and proteome. The results revealed moderate correlations between quantified transcripts and proteins among different macrophage populations (Spearman correlation coefficient *r* values from 0.41 to 0.63) (Fig. 1h),

consistent with previous research[36]. Only 28% of the proteins displayed a significant correlation with the cognate RNA (3033 proteins, Spearman $r > 0.57$ or Spearman $r < -0.57$, $p < 0.05$) (Fig. 1i), suggesting that the proteomic patterns can be explained to only a limited extent by the transcriptomic patterns. Gene function annotation revealed that 0.5% of the proteins that showed a significant negative correlation with the cognate RNA (54 proteins, Spearman $r < -0.57$, $p < 0.05$) were enriched in RNA splicing or transcription initiation, while 27% of the proteins that positively correlated with the cognate RNA (2979 proteins, Spearman $r > 0.57$, $p < 0.05$) were involved in housekeeping functions and classic immune pathways (Fig. 1j). For example, proteins involved in the inflammatory response, such as cytokines (Cxcl2), adapters (Fadd and Myd88), receptors (Ccr2, Tlr5, and Tlr12), and TFs (Stat5a, Stat6 and Runx3), showed high correlations (Spearman $r > 0.63$, $p < 0.05$) with their corresponding mRNA (Fig. 1k).

## Comparative proteome analysis of different macrophage populations revealed the tissue-specific and immune-specific functions of different macrophage populations

To explore the function properties of different macrophage populations, we carried out weighted gene co-expression network analysis (WGCNA)[40] to define co-expression gene modules for the 12 macrophage populations. The algorithm resulted 40 cell-type-specific modules (CTMs) containing 35–1080 proteins from 11,298 identifiers across the 12 macrophage populations (Supplementary Fig. 5a–c and Supplementary Data 4). To precisely identify the functional features of different macrophage populations based on the CTMs, a total of 6103 high-confidence genes in modules with a cut-off of gene significance (GS) ≥ 0.6 and module membership (MM) ≥ 0.5 (Supplementary Fig. 5d) were selected and subjected to Gene Ontology (GO) enrichment analysis (Fig. 2a and Supplementary Data 4). The analysis successfully grouped the gene signatures of different macrophage populations into their relevant CTMs based on their proteomic features (Fig. 2b). Representative markers (Slc1a3, Mef2a/c, Sall1/3, and P2ry12 in microglia; Car4 and Siglecf in alveolar macrophages; Clec4f, Vsig4, and Timd4 in Kupffer cells; and Spic in splenic red pulp macrophages) were accurately assigned (Fig. 2b).

The gene annotation analysis revealed that the properties of CTMs were highly consistent with functions of resided tissues, suggesting the involvement of macrophage populations in relevant organ-supporting activities (Fig. 2c and Supplementary Fig. 6a). For example, proteins in the CTMs of alveolar macrophages were indicated to be involved in the "response to oxidative stress" and "regulation of lipid metabolism", connecting alveolar macrophages to lung protection and surfactant balance. The CTMs of Kupffer cells were enriched in "fatty acid/xenobiotic metabolic process" and "blood coagulation", suggesting supportive roles for Kupffer cells in metabolism and blood coagulation modulation in the liver. The CTMs of intestinal macrophages were characterized by the "mucosal immune response" and "leukocyte migration", indicating a potential role for intestinal macrophages in regulating intestinal homeostasis (Fig. 2c). These results suggest that there are close connections between tissue-resident macrophages and the specific functions of relevant tissues. Undeniably, the "contamination" derived from phagocytosis activity and re-encapsulation of unrelated cells during macrophage isolation may also partly explain the existence of abundant tissue-specific detection in different macrophage populations, such as Alb in the liver, Villin (Vil1) in the gut, Vwf in endothelial cells.

In addition to non-immune-related tissue-supporting activities, macrophages extensively express proteins that participate in the innate immune response, in which the recognition of pathogen-associated molecular patterns (PAMPs) by pattern recognition receptors (PRRs) play a central role. The molecules in PRR pathways have been well explored, but their expression patterns in different macrophage populations remain unknown. Our CTM system captured the expression heterogeneity of 133 PRR signaling pathway-related proteins across the 12 macrophage populations (Supplementary Fig. 6b). Most of the PRR pathway proteins were widely expressed among the populations, while their expression levels varied in different macrophages. For instance, Tlr5, Nlrc4, Naips, etc. were significantly identified CTMs of alveolar macrophages, Nod1, Trmps, etc. were highly expressed in Kupffer cells, and Tlr1, Tlr12, P2x7, etc. were highlighted in intestinal macrophages (Fig. 2d). The tightly correlated proteins in the network may suggest a functional diversity of macrophage responses to different PAMPs.

Among the PRR signaling pathways, the Toll-like receptor (TLR) pathways have been identified as a hotspot[41]. Our study provided a rich resource for comprehensively dissecting the entire TLR family distributed in macrophages throughout the body. Almost all TLRs from TLR1 to TLR13 (except for TLR10 and TLR11) were identified in this study. Most of the TLRs were expressed in all macrophages; however, certain TLRs were significantly enriched in specific types of macrophages, suggesting that different macrophages have diverse properties in the innate immune response. Intestinal macrophages were found to express almost all TLRs (except for TLR10 and 11), revealing their extensive immune recognition of different kinds of pathogens in the intestinal tract (Fig. 2e). TLR5 was found to be predominantly enriched in alveolar macrophages, suggesting activated flagellin recognition in the lung. TLR4 was overrepresented in Kupffer cells, indicating the high ability of Kupffer cells to respond to lipopolysaccharide (LPS) stimulation (Fig. 2f). The results were well supported by published FACS data for mouse tissue macrophages[42].

## Macrophage proteome pattern revealed a TF regulatory network and hierarchical crosstalk between macrophages and relevant tissues

Previous studies based on cellular function[18,19], fate mapping[43], and omics technologies[37] highlighted the important role of TFs in maintaining the cell identities of different macrophages. The deep coverage of the proteome in our current study enabled us to profile the TF patterns of different macrophage populations. In the proteome data, a total of 510 TFs were detected, ranging from 233 TFs in the BMDMs to 338 TFs in the microglia. We determined which TFs are cell-type-specific based on the criterion that the expression level of a TF in a certain cell type was found to be five times greater than the geometric median of the expression levels in the other cell types (Methods). As a result, a panel of cell-type-specific TFs was defined (Fig. 3a and Supplementary Data 5). Supplementary Fig. 7a shows the representative specific TFs in the 12 studied macrophage populations. Some well-known cell-type specific TFs, such as Sall1 in microglia[19], Pparg in alveolar macrophages[20], Gata6 in peritoneal macrophages[44], were successfully captured by our proteome data. In contrast, typical TFs that are related to immunity, such as NF-κB, Irfs, and Stats, and LDTFs such as Pu.1 and Cebpb, were ubiquitously expressed in the 12 macrophage populations.

Based on the deep coverage of TFs in the macrophage proteome data, we constructed a TF interaction network of the 12 macrophage populations (Supplementary Fig. 7b). We found a significant positive correlation between the universality of a TF and the number of its engaged PPIs among the 12 macrophage populations (Supplementary Fig. 7c, d and Supplementary Data 6; Pearson correlation coefficient $r = 0.88$, $p = 0.00016$). Ubiquitous TFs, especially immune-related TFs, including Pu.1, NF-κB, Stats, etc., were found to be involved in many TF-TF interactions, implying the participation of ubiquitous TFs in a wide variety of cellular transcriptional programs through TF-TF interactions (Supplementary Fig. 7e). As shown in Supplementary Fig. 7f, the ubiquitously identified lineage TFs, such as Cebpb and Sfpi1 (Pu.1), could serve as hubs in the network and connect to the other cell-type-specific TFs, including Smad3 and Cebpa in microglia, Srf and Notch1in Kupffer cells, etc.

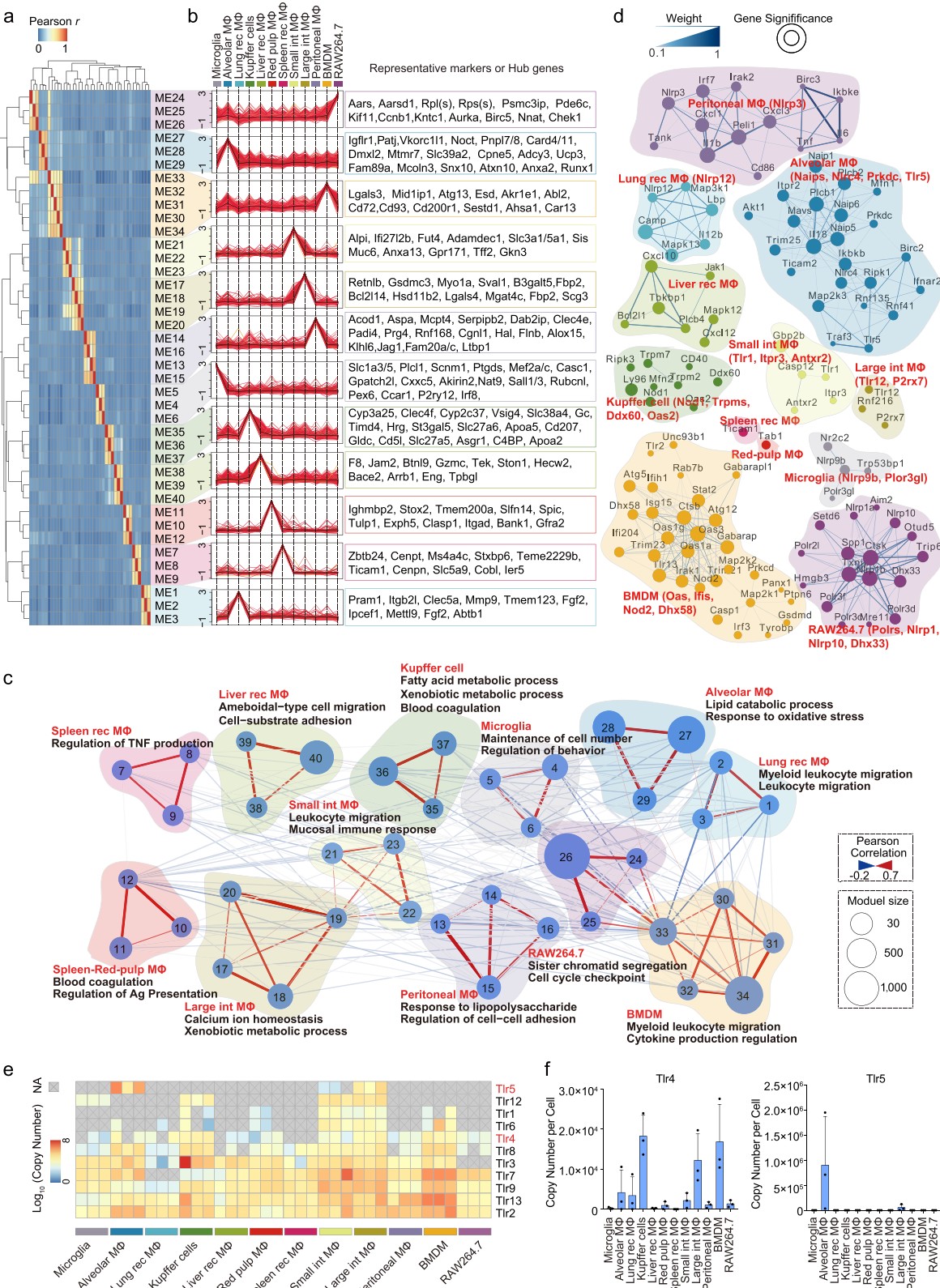

To further functionally explore the critical TFs, we set out to identify the cell-type-maintenance TFs (ctmTFs), i. e., those that would be required to maintain the identities of different macrophage types. We employed the TF-downstream target gene (TG) database from CellNET[45] with the TF patterns in this study. We reasoned that ctmTFs should not only be specifically enriched in the macrophage populations but also predominantly control the transcription of their

downstream genes in the macrophages (Supplementary Fig. 7g, Methods)[46]. As a result, 92 ctmTFs were identified in the 12 tested macrophage populations, ranging from 2 TFs in the BMDMs to 24 TFs in the RAW264.7 cell line (Supplementary Fig. 7h). As shown in Fig. 3b, a ctmTF-TG network was derived based on ctmTF alignment and the filter CTMs to portray the potential mechanism driving the heterogeneity within each of the different macrophage populations. For

**Fig. 2 | Comparative proteome analysis of different macrophage populations revealed the involvement of macrophages in the regulation of the function of resident tissues. a** Heatmap of Pearson correlation analysis results for different modules defined by carrying out WGCNA. **b** Protein expression profiles and representative signatures or hub genes of CTMs across different macrophage populations. Each line represents one protein. The average profile is shown in black. **c** Macrophage module network. Edges represent Pearson correlation coefficients coded by gradually varied color and thickness. Module size is represented by node size and the characteristics of each module are noted (one-sided Fisher's exact test,

$p < 0.05$). **d** Network depicting the diverse PRR proteins in different macrophages. Edges represent weighted correlations (weight) shown with gradually varied colors. Only network connections whose weighted correlations are above the threshold of 0.10 are shown. **e** Proteomic quantifications of TLRs across the 12 macrophage populations. The expression values are $log_{10}$-transformed copy numbers per cell. The gray blocks with a cross represent missing values. **f** Bar plots of Tlr4/Tlr5 expression across the 12 macrophage populations. $n = 3$ biologically independent experiments (Data are presented as mean ± SD). Source data are provided as a Source Data file.

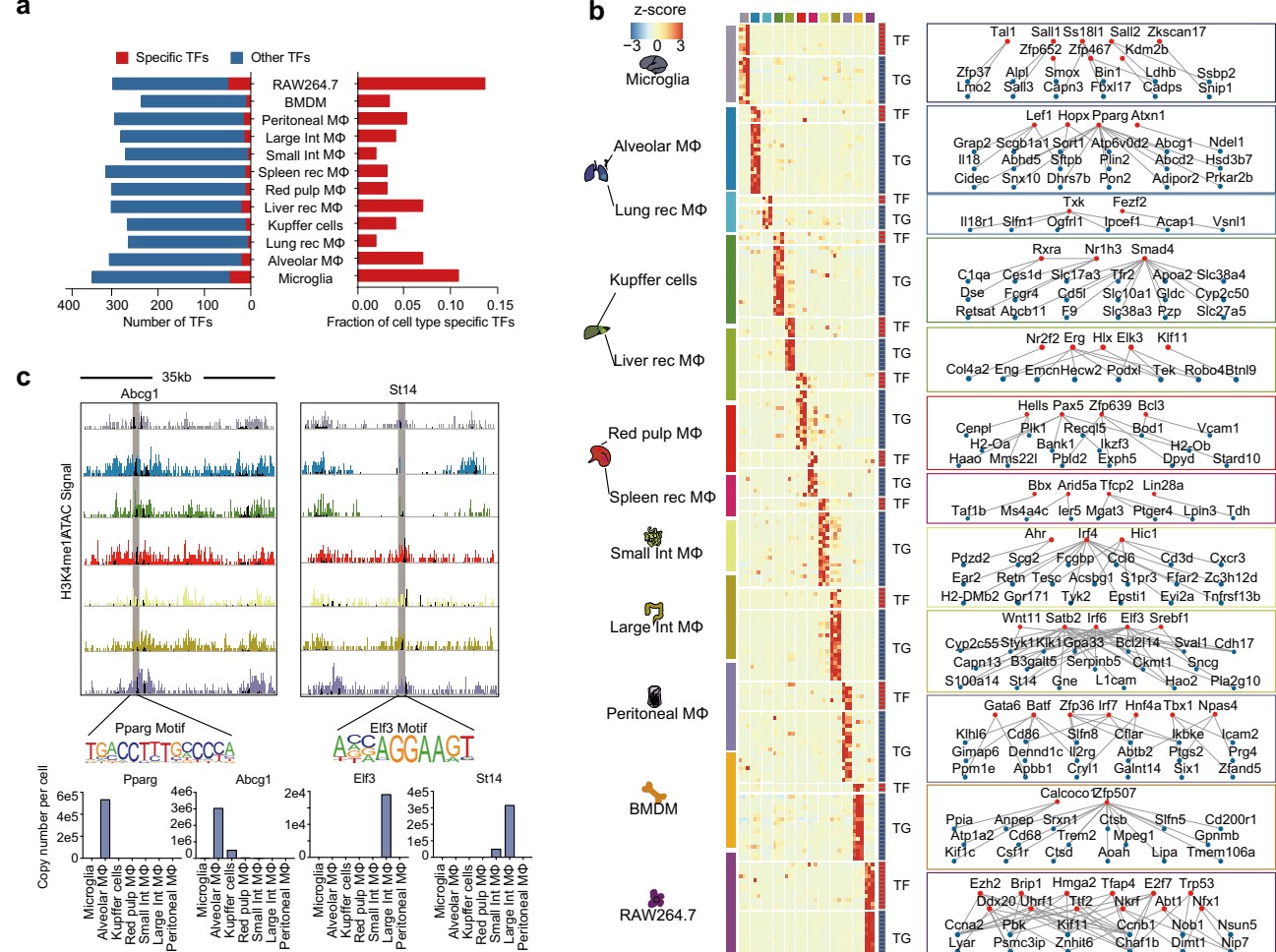

**Fig. 3 | Macrophage proteome pattern revealed a TF regulatory network and hierarchical crosstalk between macrophages and relevant tissues. a** The numbers of cell-type-specific TFs and their fractions among the total identified TFs in different macrophage populations. **b** Expression pattern and regulatory network of ctmTFs with high gene significant values in CTMs. Values for each protein (ranked along with genes from top left to bottom right in the relevant network) expression across all populations are color-coded based on z-scored copy numbers per cell in

the heatmap. The red nodes indicate ctmTFs and the blue nodes indicate relevant TGs in the network. **c** Profiles of the H3K4me1 signal in 35 kb regions with ATAC-seq peaks (shown in black) overlaid on the profiles, with enhancers around the indicated proteins contained. Shaded regions indicate the locations of the relevant motifs. The bar plots depict the protein expression levels of Pparg and Abcg1 in the alveolar macrophages, and of Elf3 and St14 in the large intestinal macrophages. Source data are provided as a Source Data file.

example, proteins in CTMs of alveolar macrophages, including Abcg1, Plin2, Abhd5, Adipor2 etc. participating in lipid metabolism, were regulated by the ctmTFs of alveolar macrophages, including Hopx, Lef1, Pparg, and Atxn1. The ctmTFs of Kupffer cells, including Rxra, Nr1h3, and Smad4, along with their target genes participating in metabolism or blood coagulation (Slcs, Cyp2c50, Apoa2, C1qa, F9, etc.), were overrepresented in the Kupffer cells. Moreover, we found consistent signals of ctmTFs and TGs in multiple omics layers, including ChIP-seq, ATAC-seq, and RNA-seq datasets[37], and the proteome data (Supplementary Fig. 7i). For instance, Pparg and its target

gene *Abcg1* were specifically expressed in the alveolar macrophages (fold change >10, $p < 0.01$). Elf3 and its target gene *St14* were specifically expressed in the large intestinal macrophages (fold change >10, $p < 0.01$). Consistent with these results, the epigenomic activation signals of Abcg1 and St14, revealed using H3K4me1 ChIP-seq, and ATAC-seq, were overrepresented in the alveolar macrophages and large intestinal macrophages, respectively (Fig. 3c). The ctmTFs are summarized in Table 1.

Ontogeny and local environmental signals can shape cell identity and are responsible for the heterogeneity of macrophages[29,47]. Our

**Table 1 | CtmTFs in 12 different macrophages**

| Cell type | ctmTFs |
|---|---|
| Microglia | Adnp2, Kdm2b, Kdm5b, **Sall1**, Sall2, Sp4, Ss18l1, Tal1, **Zfp467**, Zfp652, Zkscan17 |
| Alveolar MΦ | Atxn1, Hopx, Lef1, **Pparg** |
| Lung-recruited MΦ | Ets2, Fezf2, Tet1 |
| Kupffer Cells | Arx, Nr1h3, **Rxra**, Smad4 |
| Liver-recruited MΦ | Elk3, Erg, Foxo1, Gata4, Hlx, Klf11, Nfib, Nr2f1, Nr2f2, Tfap2c, Yap1, Zhx3 |
| Spleen red pulp MΦ | Bahd1, Bcl3, Hells, **Pax5**, Zfp639 |
| Spleen-recruited MΦ | Arid5a, Bbx, Lin28a, Sox6, Taf1b, Tfcp2, Zfp521, Zfy583, Zic3 |
| Small intestinal MΦ | **Ahr**, Hic1, **Irf4**, Zfp287 |
| Large intestinal MΦ | Aebp1, **Elf3**, Irf6, Satb2, Sphk1, Srebf1, Wnt11 |
| Peritoneal MΦ | **Batf**, Cebpd, **Fosl2**, **Gata6**, Hnf4a, Irf7, Nfe2l2, Npas4, Taf12, Tbx1, Tbx22, Zfp36 |
| BMDM | Calcoco1, Zfp507 |
| RAW264.7 | Abt1, Arnt, Brip1, Ddx20, Dhx33, E2f7, Ezh2, Gpbp1l1, Hmga2, Ing2, Med26, Mybbp1a, Nfx1, Nkrf, Psmc3ip, Rfx2, Sall4, Tfap4, Trp53, Ttf2, Uhrf1, Zfat, Zfp553, Zfp668 |

Molecules marked in bold indicated TFs that overlapped with the published epigenomic study.

functional module analysis and TF-centered cell-type-specific networks of the 12 macrophage populations indicated potential crosstalk between tissue-resident macrophages and relevant tissues. To investigate the tissue-macrophage crosstalk network, we profiled the proteomes of eight tissues (brain, lung, liver, spleen, small intestine, large intestine, peritoneum, BM) with three repeats, and identified between 3956 (peritoneal lavage fluid) and 6762 (lung) proteins per tissue, and a total of 10,669 proteins from all eight tissues (Supplementary Fig. 8a–c and Supplementary Data 7).

We employed CCCEXPLOR[48] to derive hierarchical crosstalk network by determining how ligands found in tissues are connected to receptors, specific TFs, and their downstream TGs found in macrophages (Fig. 4a). The pathways enriched in the crosstalk networks were predominantly related to the tissue function regulation of the macrophages, suggesting the effect of the tissue environment in shaping and maintaining the respective identities of different macrophage populations (Fig. 4b). Taking the crosstalk network between the brain and microglia as an example, ligands (Cx3cl1, Sema7a, Fgf8, Gnas, etc.) from the brain, and receptors (Cx3cr, Itgb1, Fgfr4, Adcy7), TFs (Jun, Mef2a, Mef2c, Smad3) and TGs (Bin1, Pacsin1, Duoxa1, etc.) from microglia, formed a crosstalk network participating in growth hormone synthesis, circadian entrainment, and positive regulation of neurogenesis (Fig. 4d; one-sided Fisher's exact test, $p < 0.001$). With the same approach, we determined the structure of the crosstalk network of the 8 tissues and the relevant 11 macrophage populations and described their featured pathways (Supplementary Figs. 9–11 and Supplementary Data 8; one-sided Fisher's exact test, $p < 0.001$), including hypoxia response (alveolar macrophages), lipid storage and iron ion transport (Kupffer cells), blood coagulation (spleen red pulp macrophages), leukocyte differentiation (intestinal macrophages), and myeloid cell differentiation (BMDMs), etc. The TFs-centered network provided evidence for the prominent role played by the tissue microenvironment in establishing macrophage identity.

**Hierarchical clustering of the proteome patterns distinguished different macrophage populations and revealed the significant differences between tissue-resident and recruited macrophages**

Based on the above-derived tissue-macrophage crosstalk network, we counted the link numbers (significantly enriched pathways, $p < 0.05$) between the macrophages and their tissues of residence. Interestingly, we found that the link numbers between the resident macrophages and their tissues of residence were greater than that between the recruited macrophages and the liver and lung (Fig. 4c; $p < 0.05$). This observation suggested the different effects of the tissue environments on shaping the identities of tissue-resident and recruited macrophages.

To further investigate the differences between tissue-resident and recruited macrophages, we performed a principal component analysis (PCA) on the macrophage proteomes, this analysis also revealed considerable differences between tissue-resident and recruited macrophage (Fig. 5a). An unsupervised hierarchical clustering analysis of the proteome patterns yielded three subclusters: cluster I included microglia, Kupffer cells and alveolar macrophages, i.e., typical tissue-resident macrophages; cluster II mainly comprised of HSC-derived macrophages, including tissue-recruited macrophages, small and large intestinal macrophages, peritoneal macrophages, and BMDMs; and Cluster III included the RAW264.7 cell line (Fig. 5b). A notable exception to the above pattern regarded the spleen: unlike the proteome patterns of typical resident macrophages in cluster I, the proteome pattern of the spleen red pulp macrophages was relatively similar to that of the spleen-recruited macrophages. Also, the classification of RAW264.7 into a separate cluster (i.e., cluster III), further indicated the difference between primary macrophages and cell lines.

We next explored the differences in function between the macrophage populations of the three clusters. Here, we used GO annotation to carry out an enrichment analysis of differentially expressed proteins (fold change > 5, $p < 0.05$) of cluster I (typical tissue-resident macrophages), cluster II (non-typical tissue-resident macrophages) and cluster III (RAW264.7). The macrophages in cluster I were found to be enriched in tissue regulatory functions, including "cellular response to metal ion" (Trf, Tfr2, etc.) and "carboxylic acid biosynthetic process" (Lpl, Abhd3, Ptgds, etc.). Macrophage populations in cluster II were featured with immune characteristics, especially cell chemotaxis (Ccr2, Ccl6, etc.) and adhesion (Elane, Selp, etc.). RAW264.7 in cluster III was characterized with "ribosome biogenesis" (Pop7, Riok2, Nsun5, etc.) and "cell cycle checkpoint" (Chfr, Trk, etc.) (Fig. 5c–e). This diversity revealed non-typical-tissue-resident macrophages to be involved in cell locomotion, while the typical tissue-resident macrophages to be mainly involved in tissue homeostasis and functional regulation.

Chemotaxis and adhesion are critical processes for the homing of the immune cells and ultimately determining their locations of residence. We found higher expression levels of chemokines, chemokine receptors and adhesion molecules in the recruited macrophages than in the resident macrophages in the lung and liver (Fig. 5f, g; fold change > 10). Conversely, the resident macrophages expressed higher amounts of PRRs than did the recruited macrophages in the lung and liver (Fig. 5h; fold change > 10, $p < 0.01$). Therefore, we speculated that compared to the resident macrophages, the recruited macrophages are more mobile but have lower PAMP response capabilities. To further validate this speculation, we investigated the correlation between

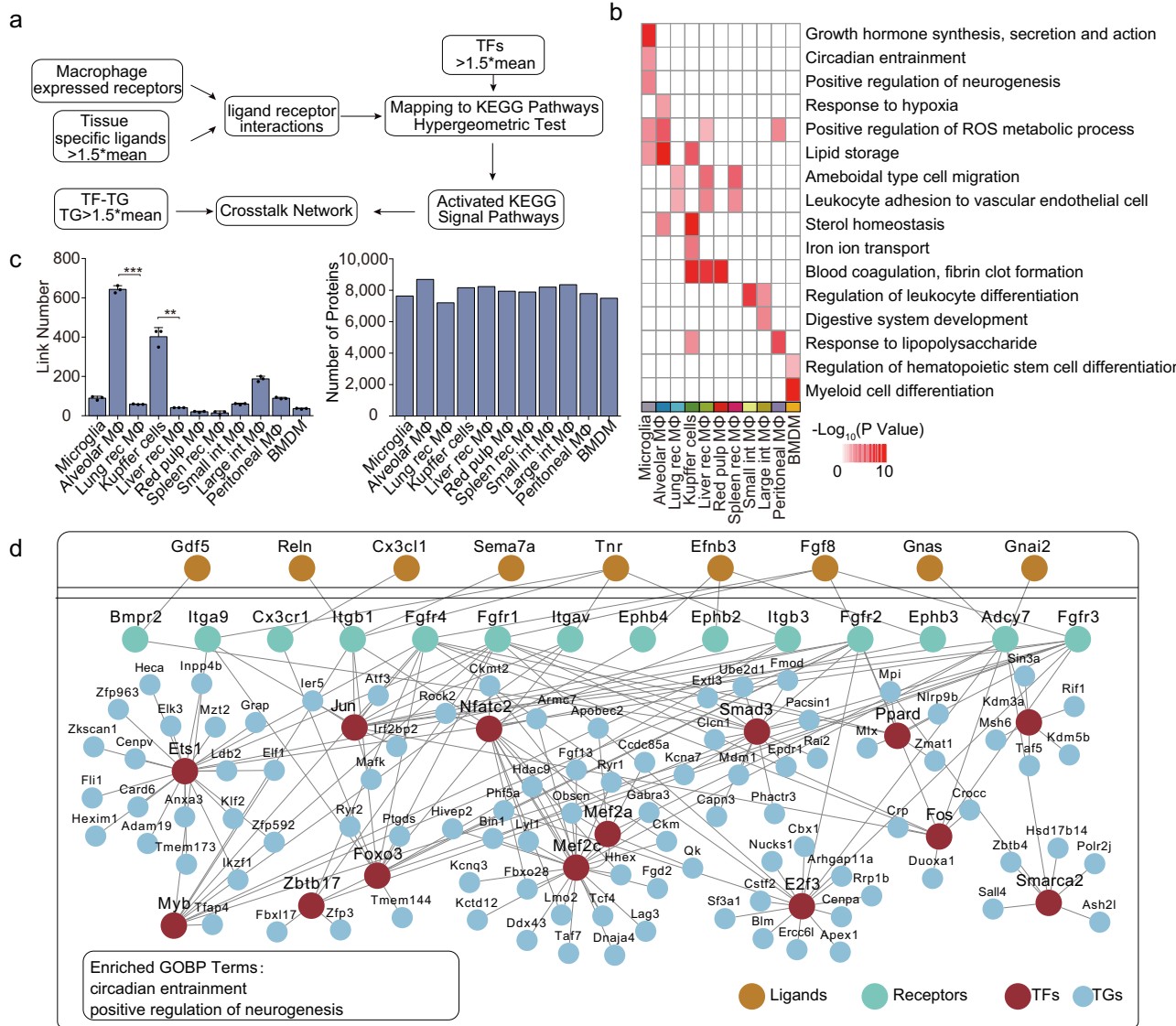

**Fig. 4 | Hierarchical proteome crosstalk networks from ligand-receptor to TF-TG target the cellular regulations between macrophages and tissues.**
**a** Workflow of the crosstalk network between macrophages and the relevant tissues. **b** The function annotation for proteins involved in macrophage-tissue crosstalk networks (ligands from tissues are excluded). Values for each term across all populations are color-coded based on log$_{10}$-scaled $p$ values in the heatmap (one-sided Fisher's exact test, $p < 0.05$). **c** Link numbers in the crosstalk network (left panel) and the total identified proteins (right panel) in 11 macrophage populations.

$n = 3$ biologically independent experiments (Data are presented as mean ± SD), \*\*\*$p = 0.0003$, \*\*$p = 0.0054$ (two-sided Student's $t$ test), from left to right. **d** Network of the crosstalk between microglia and the brain. Orange indicates ligands, green indicates receptors, red indicates TFs, and blue indicates TGs. Proteins between receptors and TFs are ignored. Functions illustrated in **b** for indicated macrophage populations are marked at the bottom. Source data are provided as a Source Data file.

expression levels of chemokine receptors and PRRs in the paired resident and recruited macrophages for three organs (the liver, lung and spleen). As shown in Fig. 5i, the correlation coefficient of the expression between the chemokine receptors and PRRs was −0.76, indicating a negative correlation between cell mobility and PAMP response capability. These results defined the distinct molecular phenotypes of tissue-resident and tissue-recruited macrophage populations.

**The functional diversity of the tissue-resident and recruited macrophages in the lung and liver**
To explore the differences between the resident and recruited macrophages in the same tissues, we surveyed the Pearson correlation coefficients for the relationship between the proteome patterns of the three paired tissue-resident and recruited macrophages in the lung, liver, and spleen. As shown in Fig. 6a, the correlation coefficient

between the proteome patterns of the spleen red pulp and recruited macrophages was 0.66, a value markedly higher than the values between the patterns of the alveolar and recruited macrophages (0.49), and between the Kupffer cell and liver-recruited macrophages (0.55). This difference might be due to the spleen being a buffering immune tissue, in which both resident and recruited macrophages are highly mobile. As shown in Fig. 6b, the recruited macrophages and the resident macrophages were determined to be co-clustered, respectively, in the lung and liver, revealing that the differences between the origins of the macrophages were greater than the diversities derived by the tissue environments.

We then investigated the functional differences between the resident and recruited macrophages in the lung and liver, respectively. Compared to the liver-recruited macrophages, the Kupffer cells were mainly involved in the biological processing and functional regulation of the liver (Fig. 6c), involving processes such as iron ion homeostasis

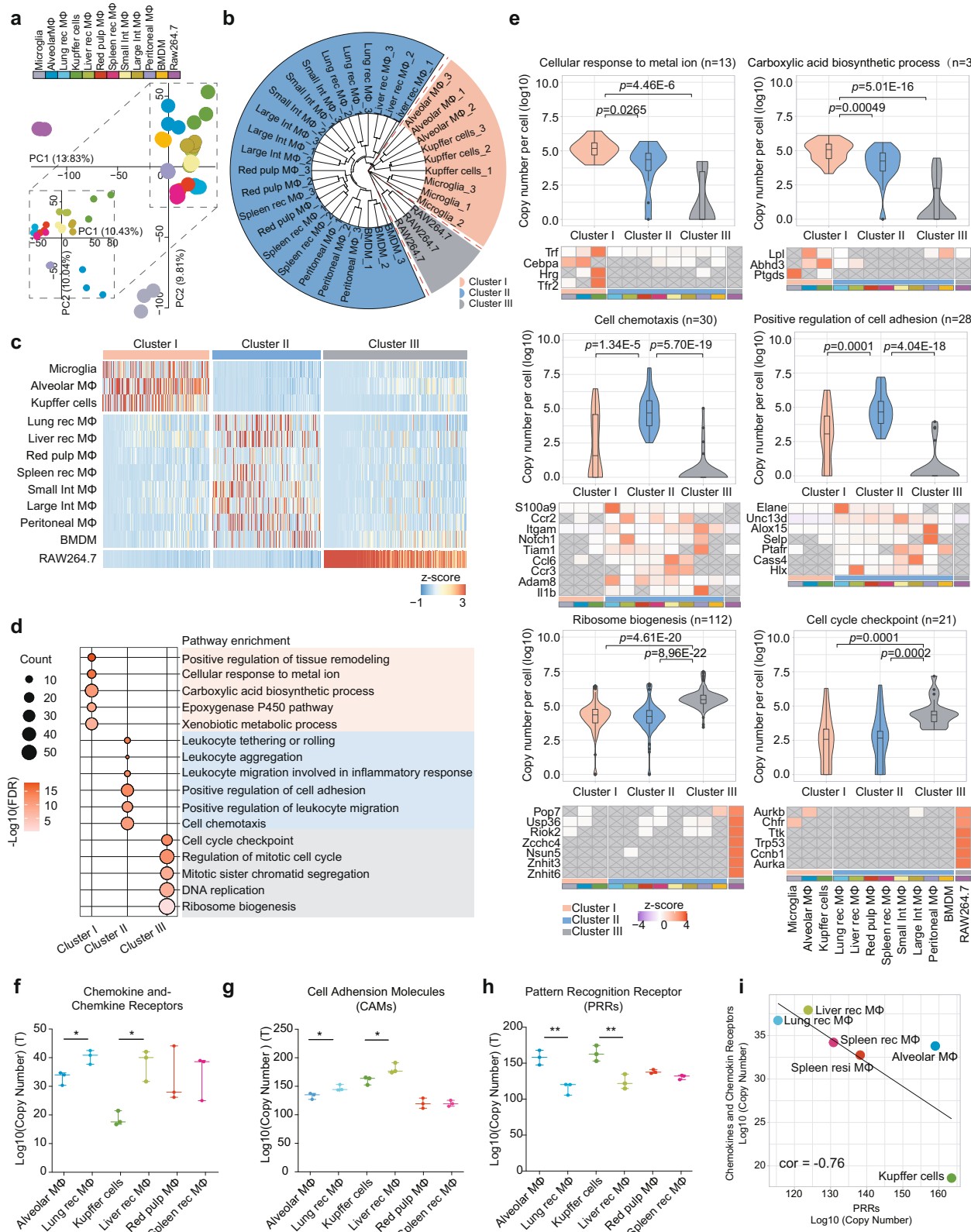

(Trfc, Tfr2, Hmox1, Atp13a2, etc.), cholesterol transport (Slc37a2, Plcl1, Lrp5, etc.), and LPS-mediated signaling (Tlr4, Nlrp3, Il18, etc.), while the liver-recruited macrophages were involved in immune regulation, including leukocyte migration (Thbs4, S100a8, S100a9, etc.), cell-cell adhesion (Selp, Ltf, Ctsg, etc.), and angiogenesis (Plcg1, Mmp9, etc.) (Fig. 6d). A similar phenomenon was also discovered for the lung. The alveolar macrophages were found to be characterized by Pparg

signaling (Slc27a1, Pparg, Acadm, etc.), regulation of pH (Tmem175, Car4, etc.), and PRR signaling (Rela, Nlrp3, Il18), indicating the involvement of alveolar macrophages in organ function assistance. In contrast, leukocyte migration (S100a8, S100a9, etc.), myeloid cell differentiation (Junb, Csf3r, Adam8, etc.), and negative regulation viral life cycle (Ltf, Ifitm6, etc.) were predominant in the lung-recruited macrophages (Fig. 6e, f).

**Fig. 5 | Hierarchical clustering of the proteome patterns distinguished different macrophage populations and revealed the significant differences between tissue-resident and recruited macrophages. a** Principal component analysis (PCA) of the protein expression patterns of the 12 macrophage populations. Inset: PCA of proteome pattern of macrophages except for microglia and RAW264.7. Different macrophages are indicated by colors in the top legend. **b** Unsupervised hierarchical clustering of the 12 macrophage populations (Euclidean distance). Three clusters were dissected and marked as red (cluster I), blue (cluster II), and gray (cluster III). **c** Heatmap of differentially expressed proteins (fold change ≥ 5, two-sided Student's *t* test, *p* value <0.05) of the three clusters. Values for each protein across all populations are color-coded based on the intensities, with low (blue) and high (red) *z*-scored copy numbers per cell. **d** Representative function annotations for differentially expressed proteins in cluster I, cluster II, and cluster III as obtained from the GOBP database (one-sided Fisher's exact test, BH FDR < 0.05). The dot size represents the number of proteins contributing to the indicated terms. The color bar shows the enrichment significance. **e** Violin plots showing the expression of DEPs in indicated cellular functions in cluster I (up panels), cluster II

(middle panels), and cluster III (bottom panels). The average copy number of each protein was calculated for each cluster. Interquartile ranges (IQRs) as boxes, with the median as a black line and the whiskers extending up to the most extreme points within 1.5-fold IQR, the outliers are shown as individual points. Exact *p* values (two-sided Student's *t* test) and protein numbers (*n*) are indicated in the boxplot, respectively. Heat maps depict the expression pattern of representative proteins for indicated functions across different populations. The gray blocks with a cross represent missing values. **f, g, h** Dot plot of the total expression levels of chemokine and chemokine receptors (**f**), cell adhesion molecules (CAMs) (**g**), and PRRs (**h**) in individual repeats of tissue-resident and recruited macrophages in the lung, liver and spleen. *n* = 3 biologically independent experiments, \**p* = 0.0363, \**p* = 0.0138, \**p* = 0.0489, \**p* = 0.0351, \*\**p* = 0.0052, \*\**p* = 0.0100 (two-sided Student's *t* test), from left to right. **i** Linear fitting of chemokine and chemokine receptors, and PRRs, across the tissue-resident and recruited macrophages in the lung, liver and spleen. The Pearson correlation coefficient is shown in the figure. Source data are provided as a Source Data file.

The analysis revealed the functional features and relevant molecular signatures between tissue-resident and recruited macrophages in immune- and non-immune-related bioprocess, especially in the liver and lung. The results highlighted: (1) the preferential involvement of tissue-resident macrophages in tissue functional regulation, and (2) the higher chemotaxis and adhesion capabilities and a lower PAMP recognition capability of tissue-recruited macrophages than did the resident macrophages.

**The diverse molecular signature and cellular functions between tissue-resident and recruited macrophages in the lung and liver**

Currently, F4/80 and CD11b serve as the relative indicators for distinguishing tissue-resident from recruited macrophage. The deep coverage of the proteome data in the current work enabled us to search for additional new protein markers to distinguish the two populations. As a result, four proteins, including Muc1, Marco, Pdl1 in tissue-resident macrophages, and Clec5a in recruited macrophages were identified as signature molecules (*p* < 0.05, fold change >10) to distinguish the tissue-resident and recruited macrophages in the lung and liver. We then verified their expression differences between the two macrophage populations using flow cytometry (Fig. 7a and Supplementary Fig. 13).

PRR pathways, such as Nlrp3 inflammasome signaling, were found to be overrepresented in the tissue-resident macrophages compared to the recruited macrophages. Il18, a cytokine participating in the Nlrp3 signaling pathway, ranked in this regard as one of the most differentially expressed proteins (fold change >100, *p* < 0.01), both in the lung and liver (Fig. 6d, f and Supplementary Fig. 12a, b). Furthermore, we also found that the other proteins in the inflammasome pathway, including Nlrp3, Nlrc4, Casp1, Naips, Gbps etc., were expressed at significantly higher levels in the tissue-resident macrophages than in the recruited macrophages (Fig. 7b; *p* < 0.05, fold change > 2). Using western blots, we confirmed the differences in the expression levels of Nlrp3, Casp1, Nek7, and Il18, proteins critically related to the inflammasome pathway, between the two types of macrophages (Fig. 7c). These results suggested the inflammatory response might be more active in the tissue-resident macrophages than in the recruited macrophages. To confirm this hypothesis, we compared the levels of inflammatory response factors between the two types of macrophages stimulated with LPS, and did so for the macrophages both in the lung and liver. Under LPS stimulation, the tissue-resident macrophages expressed Il18 and Nlrp3 inflammasome-related proteins at much higher levels than did the recruited macrophages (Fig. 7c). Other related proteins (Asc, Naips, and Gsdmd of inflammasome pathway; Akt and Nfkb of NF-κB pathway, etc.) were depicted through MS analysis, confirming that the inflammatory response was stronger in the tissue-resident macrophages (Supplementary Fig. 12c, d and Supplementary Data 9).

Il18 is a proinflammatory cytokine that enhances interferon (IFN)-γ production by anti-CD3-stimulated Th1 cells, particularly in association with Il12[49]. We therefore asked whether tissue-resident macrophages, which were shown above to produce high levels of Il18, could activate T cells more efficiently than could tissue-recruited macrophages. To this end, we constructed an in vitro co-culture system assay, in which CD4[+] T cells isolated from spleen were co-incubated with alveolar macrophages or lung-recruited macrophages, respectively. After 48 h, for each of these two experiments, the supernatant of the co-culture system was harvested and the IFN-γ level was measured to evaluate the level of activation of the T cells. IFN-γ was found to be expressed at significantly higher levels in T cells co-cultured with the alveolar macrophages than that in T cells co-cultured with the lung-recruited macrophages. However, after neutralization of Il18 with Il18 antibody, the expression level of IFN-γ in T cells of alveolar macrophage group declined (Fig. 7d). Under LPS stimulation, apart from the predominant expression of the inflammasome and Il18 in the alveolar macrophages, we noticed that the activity level of T cells dramatically increased in the presence of alveolar macrophages, compared to that in the presence of tissue-recruited macrophages, and declined after Il18 neutralization (Fig. 7e), suggesting the potential role of tissue-resident macrophage Il18 in activating T cells.

To further investigate the role of Il18 in distinguishing the molecular signature and cellular function features between two macrophage populations, we established *Il18[-/-]* mice (Methods). These mice showed intriguing phenomena. Using a co-culture system of CD4[+] T cells and macrophages, we found that the ability of alveolar macrophages activate T cells decreased from the relatively high level in wild-type mice to a relatively low level in *Il18[-/-]* mice, namely to a level similar to that of recruited macrophages, both under normal and LPS stimulation conditions (Fig. 7f, g). Similarly, under LPS stimulation, we found that Nlrp3, Caspase and Nek7 were consistently expressed in tissue-resident and recruited macrophages of the lung and liver in *Il18[-/-]* mice (Fig. 7c). We performed the proteome analysis on the two macrophage populations of the liver and lung, in both wild-type and *Il18[-/-]* mice under LPS stimulation, each in triplicate (Supplementary Data 9 and 10). Comparative proteome analysis revealed that the differential proteome patterns between tissue-resident and recruited macrophages identified in the WT mice, tend to diminish in the *Il18[-/-]* mice, both in the lung and liver (Fig. 7h). Meanwhile, compared to the wild-type mice, the *Il18[-/-]* mice also tended to show less of a difference in the expression levels of proteins related to Nlrp3 inflammasome and TLR pathways between the two macrophage populations (Supplementary Fig. 12e, f). Under LPS stimulation, the signature molecules identified above, including Marco, Muc1, Pdl1 and Clec5a, were expressed to equal extents in the alveolar and recruited macrophages in *Il18[-/-]* mice (Fig. 7i and Supplementary Fig. 13). These results indicated the potential role of Il18 in maintaining the molecular signature and cellular function features of tissue-resident

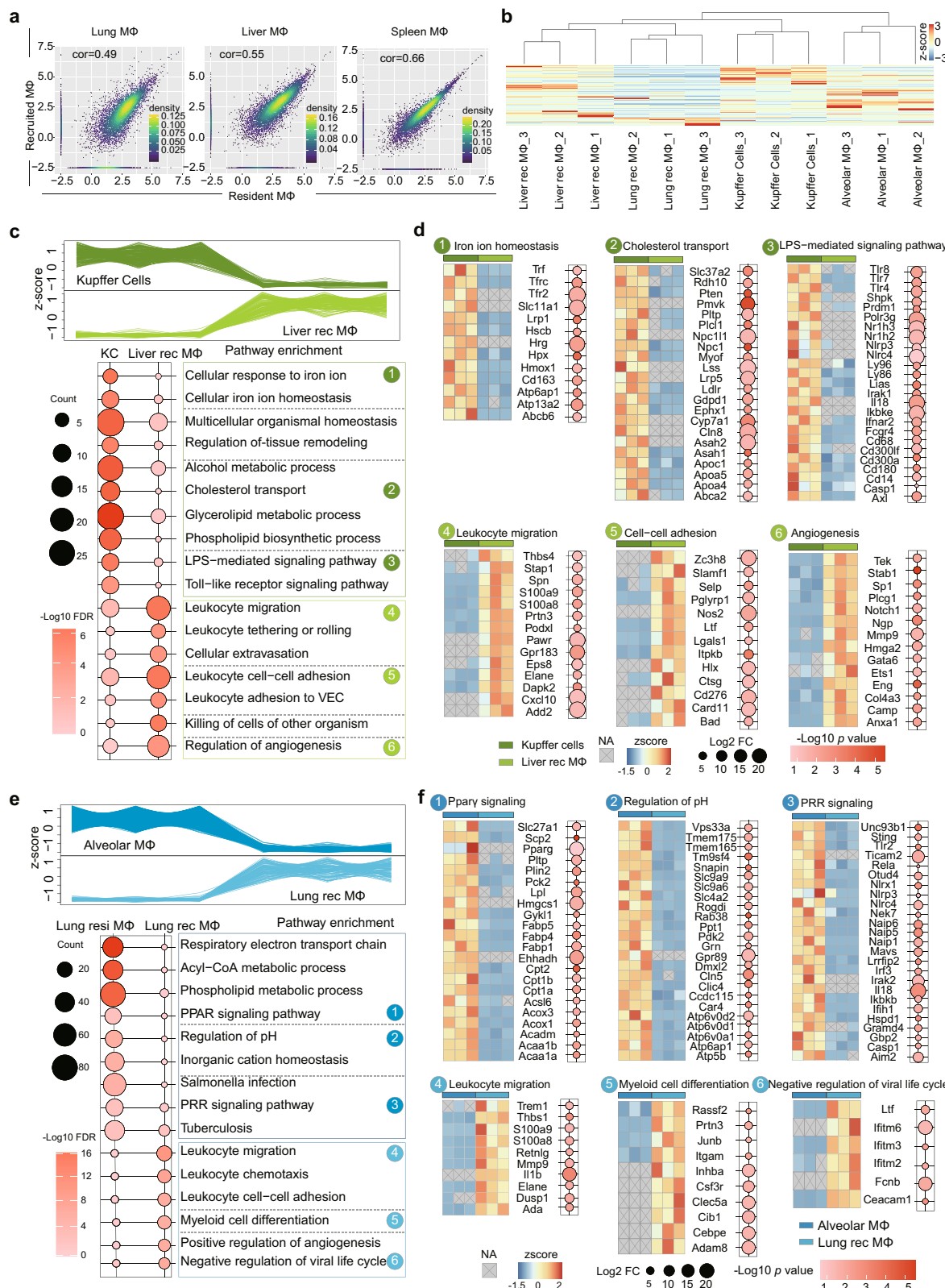

and recruited macrophages in the lung and liver, especially in inflammatory response and the capability to activate the T-cells.

### The liver-resident macrophages of Il18−/− mice tend to show an enhanced ability to recruit monocytes

In an LPS-induced acute liver injury model tested on wild-type and *Il18*−/− mice, we were surprised to notice a higher mortality of the *Il18*−/−

mice (Fig. 8a). The inflammation level was also higher in *Il18*−/− mice than in wild-type mice under LPS stimulation, both in the lung and liver (Fig. 8b). Meanwhile, the cell count of the liver-recruited macrophages was 7E5/mouse in wild-type mice, and increased to 1E6/mouse in *Il18*−/− mice; and the proportion of the recruited macrophages to total macrophages was 55% in the wild-type mice, and was increased to 77% in *Il18*−/− mice in the liver (70% in the wild-type mice and increased to 90%

**Fig. 6 | The functional diversity of the tissue-resident and tissue-recruited macrophages in the liver and lung. a** Pearson correlation coefficients between the proteome patterns of the tissue-resident ($X$ axis) and tissue-recruited ($Y$ axis) macrophages in the lung, liver and spleen. **b** Unsupervised hierarchical clustering (Euclidean distance) for proteome patterns of tissue-resident and recruited macrophages in the lung and liver. Values for each protein across all populations are color-coded based on the $z$-scored copy numbers per cell. **c, e** Expression pattern and representative function annotations (GOBP database, one-sided Fisher's exact test, BH FDR < 0.05) for differentially expressed proteins (fold change ≥ 5, two-sided Student's $t$ test, $p$ value < 0.05) in the tissue-resident and recruited macrophages in the liver (**c**) and lung (**e**). The dot size represents the number of proteins involved in the relevant terms. The color bar indicates the enrichment significance. **d, f** Expression patterns of representative proteins participating in the indicated cellular functions of the tissue-resident and recruited macrophages in the liver (**d**) and lung (**f**). Values for each protein in all populations are color-coded based on the $z$-scored copy numbers per cell. The gray blocks with a cross represent missing values. Bubble plots show the fold change and statistical significance for the indicated proteins (two-sided Student's $t$ test). Source data are provided as a Source Data file.

in the *Il18*[-/-] mice in the lung) after LPS stimulation (Fig. 8c, d and Supplementary Fig. 13). The results suggested a negative role of Il18 in regulating the monocyte-recruiting capability of the tissue-resident macrophages.

To test this suggestion of a negative role of Il18, we cultured the Kupffer cells isolated from both wild-type and *Il18*[-/-] mice treated with LPS or PBS. After 6 h, the cells were subjected to LC-MS measurements. For the macrophages of the mice treated with LPS, a total of 6599 proteins were identified (Supplementary Data 11). The bioinformatics analysis of the proteome data indicated that chemokines that recruit monocytes (such as Ccl7 and Cxcl12) were expressed at higher levels in the Kupffer cells in *Il18*[-/-] mice than in wild-type mice (Fig. 8e and Supplementary Fig. 12g). Therefore, we proposed that the ability of the Kupffer cells to recruit monocytes was upregulated in the *Il18*[-/-] mice. To provide more evidence for this proposal, we cultured and stimulated Kupffer cells in wild-type and *Il18*[-/-] mice, and used a trans-well assay to test the ability of RAW264.7 cells to migrate when under the treatment of the supernatant of Kupffer cells. Here, the transmembrane cell counts of RAW264.7 in the *Il18*[-/-] group were higher than those in the wild-type group, indicating the greater ability of the Kupffer cells to recruit monocytes in the *Il18*[-/-] mice than in the wild-type mice (Fig. 8f).

Based on the results taken together, we revealed the critical role of Il18 in maintaining molecular signature and cellular function features of these tissue-resident and recruited macrophages, especially in inflammatory response and the capability to activate T-cells. Also, we found an enhanced ability of the Kupffer cells to recruit monocytes in the *Il18*[-/-] mice, which may contribute to the high mortality of *Il18*[-/-] mice after LPS stimulation.

## Discussion

Macrophages represent a striking cell type that is present in nearly all tissues, beginning during embryonic development. In addition to their roles in innate immune defense and apoptotic cell clearance, macrophages are being increasingly recognized for their regulatory functions in relevant tissues. Here, we provided an in-depth description of the proteomes and transcriptomes of ten different primary macrophage populations and BMDMs derived from C57BL/6N mouse, providing comprehensive protein and mRNA expression patterns for dissecting the different functions of the different types of macrophages. Considering the absence of C57BL/6N homologous cell lines and the widespread use of RAW264.7, we selected RAW264.7 as the murine cell line for comparison with primary macrophages. Of course, the cell line RAW264.7 is not a C57BL/6 line, the influence derived from the mouse sub-strain should also be noted.

Comparisons of our proteomic and transcriptomic data with published data demonstrated that we captured the characteristics of the gene expression profiles of the relevant macrophage populations. It is worth to note that our data as well as almost all of previously available data of macrophage populations are mouse-specific, especially C57BL/6 strain-specific. The details of macrophage ontogeny, development, and homeostasis may vary across different species or even different mouse strains[11,50], and the caveats to the use of the C57BL/6 strain as a model for monocyte/macrophage research have

been well recognized and reviewed[4]. The quite radical difference between different species and mouse strains deserves further analysis in future study.

Our dataset provides an opportunity to explore differences between the proteome and transcriptome of the same cell type. Consistent with the published literature[51], we found that the expression levels of over 75% of the transcripts with missing proteins (accounting for 3339 genes out of 4269 missing proteins) were lower than the median expression level of the total transcriptome. These low-abundance transcripts might not be expressed on protein level, or their expression levels might be too low to be detectable by MS. The other 20% of the missing proteins were not detected due to specific post-translational modifications (ubiquitination) and extracellular localization. Some of the missing proteins were membrane proteins or TFs, which were hard to be detected by MS regardless of the cognate RNA expression abundance, might because they were difficult to extract by the lysis. In addition, we found some missing proteins were reported with short half-life, such as ribosomes[39]. One of the future directions for proteomics is to improve the coverage and sensitivity of proteome identification. The critical reasons causing missing protein or missing transcripts signals are also worth further investigation.

Comparative analysis of the macrophage proteomic landscape captured the representative gene signatures and functional heterogeneity of different primary macrophages in the physiological state, and profiled the characteristics of the immune- and non-immune-related bioprocess of each type of macrophage. The results provide a reference for performing novel investigations of the function and mechanism of macrophages. It is important to note that the cells used in our study were obtained through enzymatic digestion and FACS purification, although we performed this processing as quickly and gently as possible. These operations can activate macrophages under steady-state conditions, prompting the activation of immediately expressed genes and inflammatory cytokines. Furthermore, as macrophages are a type of phagocyte, tissue proteins phagocytosed by these cells were also captured by MS. This may influence the WGCNA-based gene cluster and functional analysis results in our study. This is an inevitable limitation of the current study. We will explore the effects of perturbations derived from different experimental procedures on gene expression profiles of primary cells in subsequent studies.

Another factor affecting protein or transcript quantification is cell contamination. Meta-analysis performed by Summers et al. indicated that the published macrophage transcriptomic datasets were influenced by extensive contamination of isolated preparations with other cell types straightforwardly or co-purification of cell types that may interact with MPS cells in vivo[22]. Here, to minimize the contamination of unrelated cells, we used 9–10 markers to sort each primary cell populations to ensure that the purities of our macrophage preparations were above 98%. The possible interference or contamination by unrelated cells can also impact mRNA/protein quantification results as well as subsequent data analysis, which is an inevitable limitation of the current study. First, the contamination derives from the phagocytosis activity of macrophages, in which RNA/protein from the engulfed cell may be detected. Second, the perturbation that arises from unrelated cells coated by macrophage remnants during tissue digestion. Third,

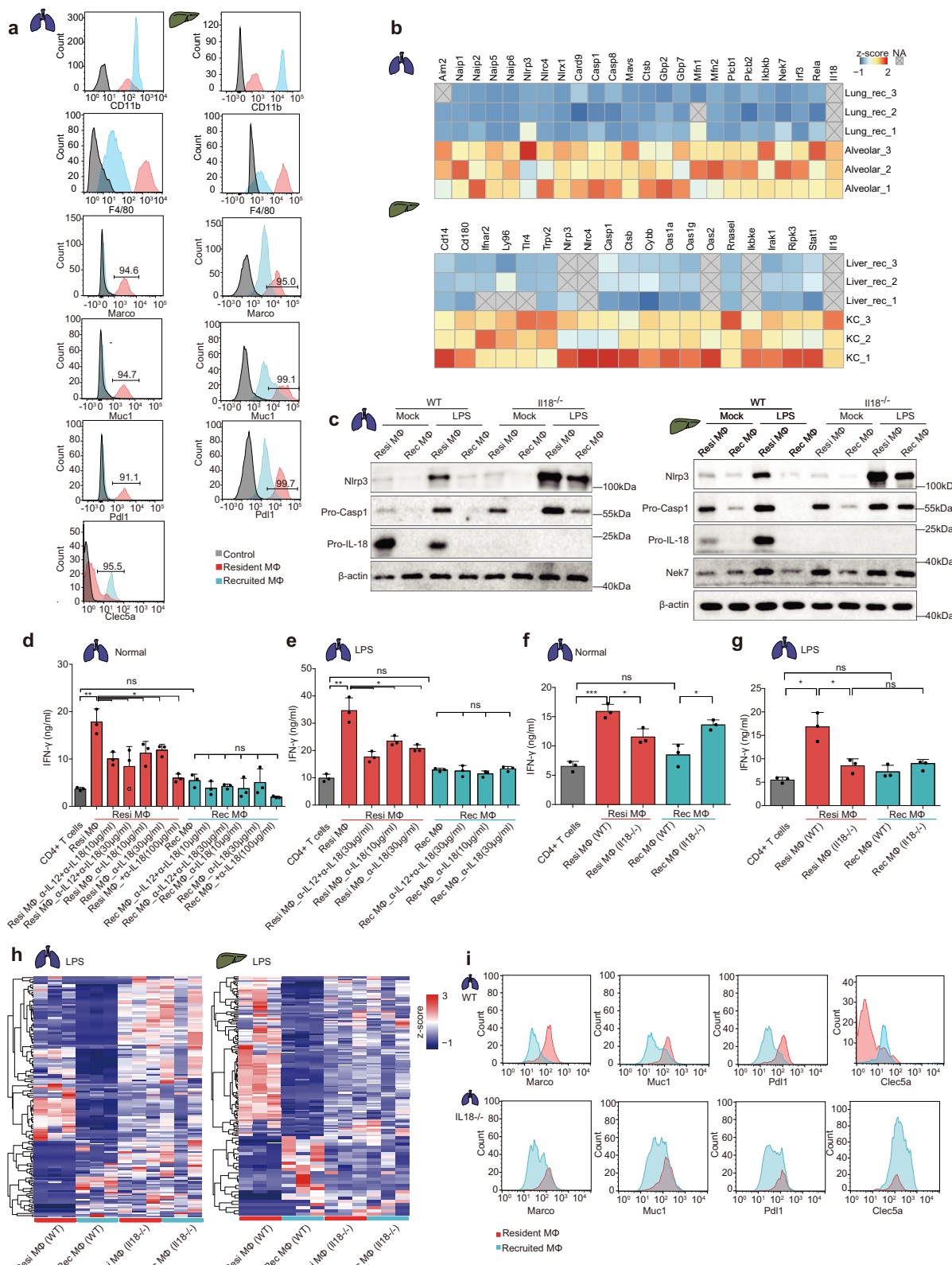

the tight interactions between macrophages and other cell populations also give rise to the consequence. These phenomena cannot be recognized or excluded by doublets gating in flow cytometry analysis. Such contamination can partly explain the phenomenon that specific proteins common in certain tissues may then be found in macrophages isolated from that same tissue. Also, the issue may influence the WGCNA-based gene cluster and functional analysis results in our

study. Therefore, it becomes worthwhile to explore the effects of perturbations derived from different experimental procedures on gene expression profiles of primary cells in subsequent studies.

Given the important functions of cardiac macrophages in tissue-supporting activities, we also archived the proteomic data of cardiac macrophages based on the sorting strategy of "CD45+F4/80hiCD11-blo Ly6g Ly6c CD11clo" (Supplementary Fig. 13c). The meta-analysis

**Fig. 7 | The diverse molecular signature and cellular functions between tissue-resident and recruited macrophages in the lung and liver. a** Representative histograms showing the fluorescence signals of markers, including F4/80, CD11b, Marco, Muc1, Pdl1 and Clec5a, in tissue-resident and recruited macrophages in the lung (left panel) and liver (right panel), as measured using FCM. **b** The heatmaps indicating the expression patterns of main proteins participating in the Nlrp3 inflammasome signaling pathway, in the tissue-resident and recruited macrophages. Values for each protein in all samples analyzed (rows) are color-coded based on the expression level, i.e., low (blue) and high (red) $z$-scored copy numbers. The gray blocks with a cross represent missing values. **c** WB validation of Nlrp3 inflammasome-related proteins in tissue-resident and recruited macrophages in the lung (left) and liver (right) of wild-type or $Il18^{-/-}$ mice, under normal or LPS stimulation conditions. The experiment was repeated three times independently with similar results. **d, e** Histogram of IFN-γ expression in CD4⁺ T cells co-cultured with alveolar macrophages (red) or lung-recruited macrophages (blue) in wild-type mice, under normal (**d**) or LPS stimulation conditions (**e**). α-Il12 and α-Il18 indicate neutralizing antibodies against Il12 and Il18, respectively. $n$ = 3 biologically independent

experiments (Data are presented as mean ± SD, **$p$ = 0.0093, *$p$ = 0.0162, *$p$ = 0.0404, *$p$ = 0.0321, *$p$ = 0.0362, **$p$ = 0.0083, **$p$ = 0.0073, *$p$ = 0.0135, *$p$ = 0.0393, *$p$ = 0.0280 (two-sided Student's $t$ test), from left to right. **f, g** Histogram of IFN-γ expression in CD4⁺ T cells co-cultured with alveolar macrophages (red) or lung-recruited macrophages (blue) in wild-type or $Il18^{-/-}$ mice, under normal (**f**) or LPS stimulation conditions (**g**). $n$ = 3 biologically independent experiments (Data are presented as mean ± SD), ***$p$ = 0.0007, *$p$ = 0.0152, *$p$ = 0.0256, *$p$ = 0.0203, *$p$ = 0.0260 (two-sided Student's $t$ test), from left to right. **h** The heatmaps indicating the proteome patterns of the tissue-resident and recruited macrophages in wild-type and $Il18^{-/-}$ mice, in the lung (left) and liver (right). Differentially expressed proteins between the tissue-resident and recruited proteins in wild-type mice with fold change > 5, $p$-value < 0.05 (Wilcoxon signed-rank test) were selected. Values for each protein in all macrophages are color-coded based on the expression level, i.e., low (blue) and high (red) $z$-scored copy numbers. **i** Representative histograms showing the fluorescence signals of indicated markers in alveolar and recruited macrophages in wild-type (upper) and $Il18^{-/-}$ mice (lower) under LPS stimulation, as measured using FCM. Source data are provided as a Source Data file.

based on 466 bulk RNA-seq datasets indicated that there were no unique expression profiles enriched in macrophages isolated from the heart[22] so that our proteome datasets provide a good opportunity for cardiac macrophage investigation. The dataset has been included in the online data portal (http://macrophage.mouseprotein.cn).

Previous researches have utilized different omic (epigenomic and transcriptomic) approaches to define the "key" TFs of macrophages. Lavin et al. screened the regulatory elements in eight different macrophage populations using genome-wide assays, including Assay for Transposase-Accessible Chromatin using sequencing (ATAC-seq), chromatin immunoprecipitation sequencing (ChIP-seq), and RNA-seq[37]. Based on their TF DNA-binding motif analysis results, they predicted many regulatory candidates, such as Runx2, Stats, Smads, etc. In our current study, we directly identified cell-specific TFs and "nominated" the ctmTFs responsible for maintaining the identities of different macrophages.

The functional features of the tissue-resident and recruited macrophages in the same tissue had not been fully characterized previously. In our current study, significant differences were observed between the proteome patterns of the tissue-resident macrophages and those of the recruited macrophages, allowing us to assess the functional diversities of these two macrophage populations, especially in the liver and lung: (1) the resident macrophages were predominantly involved in the tissue function regulation and had more connections to their tissues than did the recruited macrophages; (2) the recruited macrophages showed a greater chemotaxis and adhesion capabilities and a lesser PAMP recognition capability than did the resident macrophages; (3) IL18 may play important role in distinguishing feature identities of the two macrophage populations.

In summary, we presented comprehensive proteome and transcriptome datasets of macrophage populations residing in different tissues. This rich resource not only revealed the functional diversities of different macrophage populations but could also serve as a starting point for future studies of macrophage biology, providing a better understanding of the immune-related and non-immune-related functions of macrophages throughout the body.

## Methods
### Animals and tissue collection
Normal male C57BL/6N mice purchased from Shanghai Slac Laboratory Animal Co., Ltd., and $Il18^{-/-}$ mice provided kindly by Professor Rongbin Zhou, were kept in SPF conditions at the College of Pharmacy, Zhangjiang Campus, Fudan University. Eight- to twelve-week-old (20–25 g) mice were subjected to tissue collection or macrophage isolation. The euthanasia of animals was performed by carbon dioxide ($CO_2$) inhalation. The permission for animal experiments was granted by the Research Ethics Committees of Zhongshan Hospital.

### $IL18^{-/-}$ mice construction
$Il18^{-/-}$ mice have been previously reported[52]. To be more specific, IL-18 genomic DNA was screened from a 129/SvJ mouse genomic library (Stratagene), subcloned into the pBluescript vector, and characterized by restriction enzyme mapping and DNA sequencing. A targeting vector was designed to replace a 3.0 kb genomic fragment containing exons 3, 4, and 5 with pMC1-$neo$ (Stratagene). The targeting vector was flanked by the 5.2 kb fragment at the 3′ end and the 1.0 kb fragment at the 5′ end and contains an HSV-tk cassette at the 3′ end of the vector. The targeting vector was linearized with SalI and electroporated into embryonic day 14.1 embryonic stem (ES) cells. Homologous recombinant ES cells were identified by Southern blot analysis and microinjected into C57BL/6N blastocysts. Offspring were backcrossed to C57BL/6 mice and germline transmission was confirmed by PCR of tail genomic DNA. Littermate C57Bl/6 $Il18^{+/+}$ male mice were used as controls and were co-housed with experimental mice.

### Cell isolation
Macrophages were isolated and purified in reference to published studies with some modifications. The brief protocols for different macrophage populations were described as follows.

Microglia were isolated from the brain following enzymatic digestion and Percoll density gradient centrifugation[53]. Specifically, Mice were anaesthetized with $CO_2$ and immediately perfused intracardially with ice-cold $Ca^{2+}/Mg^{2+}$-free Hank's balanced salt solution (HBSS) to clear blood cells. Whole brains were then gently removed and minced in 15-ml dounce homogenizer containing 3 ml digestion cocktail (HBSS with 0.5 mg/ml collagenase IV (Sigma, cat. no. c5138), 0.01 mg/ml DNase I (Worthington, cat. no. b002004), 2 mg/ml dispase II (Sigma, cat. no. D4693), and 0.1 mg/ml Nα-tosyl-L-lysine chloromethylketone hydrochloride (TLCK, Sigma, cat. no. T7254)). Homogenates were then gently rocked and enzymatically digested in 15 ml digestion cocktail for 15 min at room temperature and centrifuged 7 min at 300 × $g$ at 18 °C. Then the cell pellet was washed once with HBSS and subjected to density gradient centrifugation (with the 0%, 30%, 37%, and 70% Percoll (Amersham, cat. no. 17-0891-02) density gradient) at 200 × $g$ at room temperature for 40 min (the centrifuge was stopped with minimal/no brake). Finally, the cells in 37–70% layer were collected using a transfer pipette and washed twice with HBSS. The cells were resuspended in staining buffer (ice-cold $Ca^{2+}/Mg^{2+}$-free HBSS containing 0.5% BSA (bovogen, cat. no. BSAS 0.1)) for later FACS antibody incubation and analysis.

Alveolar macrophages and lung-recruited macrophages were purified from the lung through enzymatic digestion[54]. Specifically, mice were anaesthetized with $CO_2$ and immediately perfused intracardially with ice-cold $Ca^{2+}/Mg^{2+}$-free Hank's balanced salt solation (HBSS) to clear blood cells. The lung was removed and transferred into

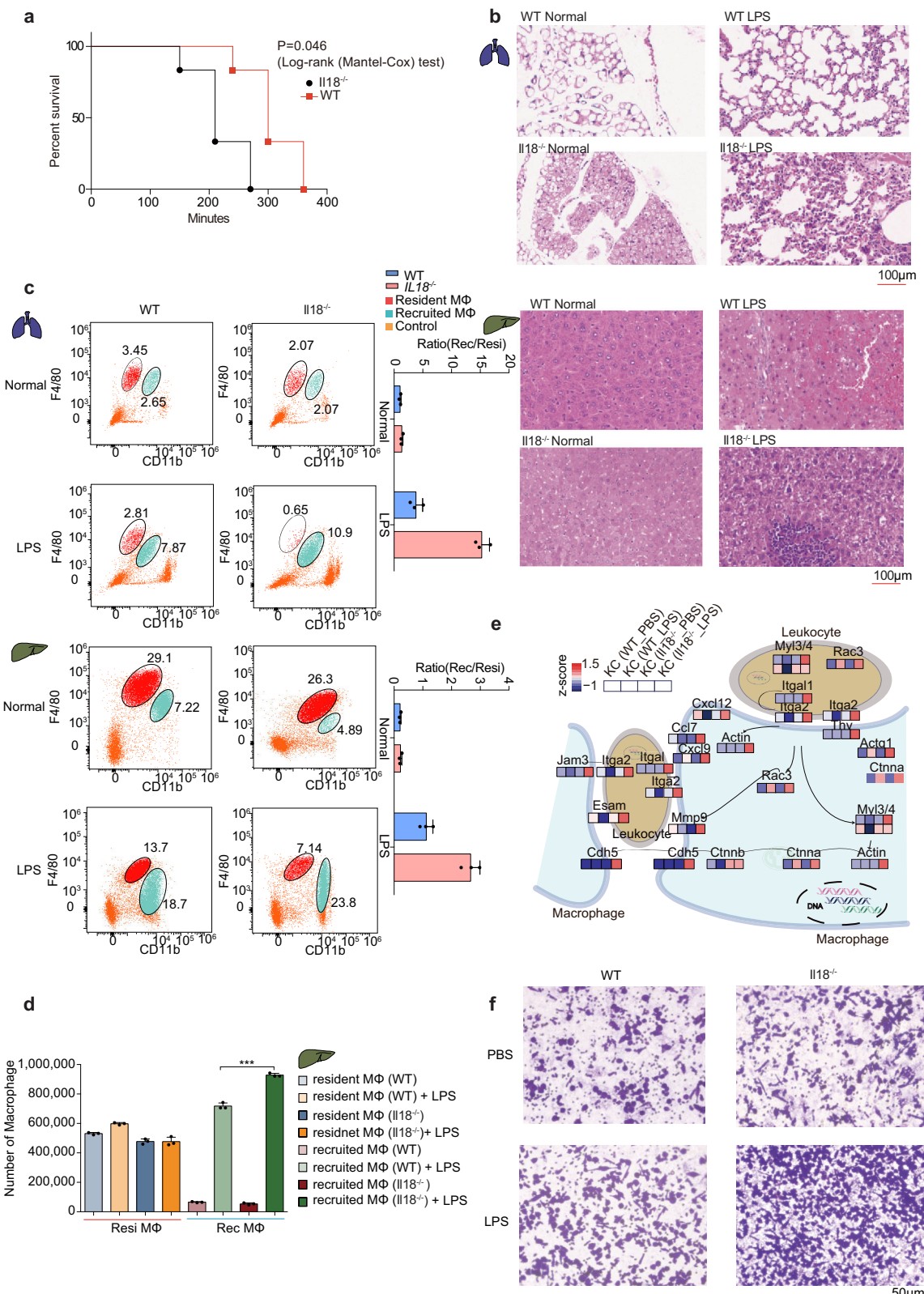

Petri dishes containing RPMI-1640 medium (HyClone, cat. no. SH30809.01), cut and minced into small pieces. The lung pieces were enzymatically digested with 15 ml RPMI-1640 medium containing 0.5 mg/ml collagenase IV and 0.02 mg/ml DNase I for 20 min, then further digested with additional 5 ml fresh enzyme solutions for 10 min, at 37 °C in a humidified atmosphere containing 5% $CO_2$ Cell aggregates were dispersed by repeated passage through a syringe and

filtered through a 100 μm cell strainer (BD Biosciences, cat. no. 352360) to obtain the single-cell suspension. Finally, cells were washed twice with HBSS and resuspended in staining buffer for later FACS antibody incubation and analysis.

Kupffer cells and liver-recruited macrophages were extracted from the liver by a two-step perfusion digestion in situ of the hepatic and Optiprep™ density gradient centrifugation[55]. Specifically, mice

**Fig. 8 | The liver-resident macrophages of *Il18*$^{-/-}$ mice tend to show an enhanced ability to recruit monocytes. a** Survival curves of *Il18*$^{-/-}$ mice and littermate wild-type mice (8 weeks-old C57BL/6 male mice) after LPS/D-GalN co-injection (*n* = 6 mice per group). **b** Haematoxylin and eosin (HE) staining of lung or liver sections in wild-type or *Il18*$^{-/-}$ mice, prepared after the LPS/D-GalN co-injection or under normal conditions. Scale bars each correspond to 100 μm. The experiment was repeated three times independently with similar results. **c** Representative FCM plots showing the percentage of lung- (upper panel) or liver- (lower panel) resident and recruited macrophages isolated from wild-type or *Il18*$^{-/-}$ mice, under normal or LPS stimulation conditions. The ratios of the cell number for tissue-resident macrophages to that of recruited macrophages are shown in right with bar plots. *n* = 3 biologically independent experiments (Data are presented as mean ± SD),

***p* = 0.0004, ***p* = 0.003 (two-sided Student's *t* test), from up to down. **d** Representative histograms show the cell counts of the Kupffer cells and recruited macrophages isolated from wild-type or *Il18*$^{-/-}$ mice with or without LPS stimulation. *n* = 3 biologically independent experiments (Data are presented as mean ± SD), ***p* = 0.0005 (two-sided Student's *t* test). **e** Schematic of the differential expression of migration-related proteins detected in Kupffer cells with PBS or LPS. Values for each protein in all analyzed samples are color-coded based on their intensities, i.e., low (blue) and high (red) *z*-scored copy numbers per cell. **f** Representative images showing the RAW264.7 cells that migrated through the upper chamber of the migration assay device used. The experiment was repeated three times independently with similar results. Source data are provided as a Source Data file.

were anaesthetized with $CO_2$ and subjected to a laparotomy to expose the liver, inferior vena cava (IVC) and portal vein (PV). Then the liver was perfused by the first step with warm $Ca^{2+}/Mg^{2+}$-free HBSS containing 1.9% EGTA (Sigma, cat. no. E4378) until free of blood by visual inspection, and perfused by the second step with warm HBSS containing 1% FBS (Gibco, cat. no. C11995500BT) and 0.5 mg/ml collagenase IV and 5.6% $CaCl_2 \cdot 2H_2O$ (Sigma, cat. no. C7902) until the tissue become nonelastic, through PV to IVC in situ with a peristaltic pump. The liver was then removed and transferred into a Petri dish containing DMEM medium (HyClone, cat. no. SH30022.01B) with 0.5% BSA, gently minced into cell aggregates and filtered through a 100 μm cell strainer to obtain the single-cell suspension. Subsequently, the parenchymal liver cell pellet was discarded after cell centrifugation at $50 \times g$ at 4 °C for 2 min. The supernatant was centrifugated at $500 \times g$ at 4 °C for 8 min to obtain nonparenchymal cell pellet (containing macrophage populations). Finally, the nonparenchymal cells were subjected to density gradient centrifugation (with the 0%, 11.2%, 17.6 and 24% Optiprep™ (Axis-shield, LYS3782) density gradient) at $1400 \times g$ at room temperature for 18 min (the centrifuge was stopped with minimal/no brake). Finally, the cells in 11.2–17.6% layer were collected using a transfer pipette and washed twice with HBSS. The cells were resuspended in staining buffer for later FACS antibody incubation and analysis.

Spleen red pulp and spleen-recruited macrophages were extracted after tissue grinding and red blood cell (RBC) lysis. Specifically, the spleen was gently grinded mechanically (without enzymatic digestion) into RPMI-1640 medium containing 5% FBS with the tip of a syringe plunger on a 100 μm cell strainer. Then the cell suspension was centrifuged and washed twice with ice-cold phosphate-buffered solution (PBS). The RBC was removed using cold RBC lysis buffer (BD Pharmingen, cat. no. 555899) for 3 min on ice. Finally, the cells were washed twice with HBSS and resuspended in staining buffer for later FACS antibody incubation and analysis.

Intestinal macrophage populations were extracted from small or large intestine tissue after epithelial segregation and enzymatic digestion[56]. Specifically, mice were anaesthetized with $CO_2$ and large and small intestines were transferred respectively into Petri dishes containing $Ca^{2+}/Mg^{2+}$-free PBS. The Peyer's patches along the anti-mesenteric surface of the intestines were dissected out with scissors and forceps. Intestines were then cut open longitudinally using scissors and washed to remove the fecal contents and mucus from the intestine lumen at room temperature in $Ca^{2+}/Mg^{2+}$-free PBS. Subsequently, the clean intestines were cut into small pieces (1.5 cm) into 50 ml conical tube containing 20 ml $Ca^{2+}/Mg^{2+}$-free HBSS with 5% FBS and 2 mM EDTA. Tubes were shaded horizontally at 250 rpm for 20 min at 37 °C in an orbital shaker in the humidified atmosphere containing 5% $CO_2$ (Repeat the step for 2–3 times to segregate epithelial cells). Then the tissues were collected and washed in 20 ml $Ca^{2+}/Mg^{2+}$-free HBSS with 5% FBS (without EDTA) and digested horizontally with 10 ml HBSS containing 5% FBS, 0.5 mg/ml collagenase IV and 0.02 mg/ml DNase I for 10 min at 150 rpm for 10–15 min at 37 °C in the orbital shaker (check the tissue

status every 3 min to avoid over-digestion). Subsequently, the tissue suspension was briefly vortexed and filter through a 100 μm cell strainer to obtain single-cell suspension. The cells were pelleted by centrifugation at $1000 \times g$ for 5–8 min at 4 °C. Finally, cells were washed twice with HBSS and resuspended in staining buffer for later FACS antibody incubation and analysis.

Peritoneal macrophages were isolated by carrying out peritoneal lavage. Mice were anaesthetized with $CO_2$ and quickly euthanized by cervical dislocation. Peritoneal cells were collected by lavages of the peritoneum with 5 ml ice-cold HBSS containing 1% FBS and 1 mM EDTA. Then the cells were washed and resuspended in ice-cold staining buffer for later FACS antibody incubation and analysis.

BMDMs were obtained from BM after 7 days of in vitro culture in DMEM with 10% FBS and M-CSF Specifically, Mouse BM was harvested from femur and tibia bones using 21 G needle and 10 ml syringe with 5 ml cold PBS containing 2% FBS. The marrow was dissociated by passage through a 21 G needle 4–6 times and a 100 μm cell strainer to obtain single-cell suspension. The RBC was removed using ice-cold RBC lysis buffer for 2 min on ice. Then the cells were pelleted through centrifugation at $500 \times g$ for 5 min at 4 °C, washed once in PBS and resuspended in BMDM culture medium containing 10% FBS, 25 ng/ml mouse M-CSF (R&D, cat. no. 416-ML-050/CF), and 100 U/ml penicillin/streptomycin. Subsequently, the cells were seeded in a Petri dish with BMDM culture medium at 37 °C in a $CO_2$ incubator. Change fresh BMDM culture medium on day 3. After 6–8 days of culture, non-adherent cells were washed off with PBS for 2–3 times. BMDM was digested by trypsin into single-cell suspension, washed twice with PBS and resuspended in staining buffer for further FACS antibody incubation and analysis.

### Flow cytometry and cell sorting

Macrophage cells were incubated on ice with CD16/32 blocking antibody (BD Biosciences, cat. no. 553142) for 20 min, and stained with fluorescently labeled antibodies (1:200 dilution in ice-cold staining buffer) directly against cell surface markers for 30 min. Cells were then washed twice, filtered through a 40 mm cell strainer (BD Falcon, 352340), and analyzed on Beckman Counter cytoFLEX LX, or sorted by BD influx or BD FACS AriaIII with 100-μm nozzle. flow cytometry (FCM) data were analyzed by FlowJo™ 10.7.1. Antibodies used in this study were presented in Supplementary tables (Supplementary Table 1).

### Cell culture

The macrophage cell line RAW264.7 was cultured in DMEM supplemented with 10% FBS, penicillin/streptomycin (100 U/ml) at 37 °C and 5% $CO_2$. The state of cell culture growth was recorded in real time with inverted phase contrast microscopy. The cells with concentration of 80% were digested by trypsin and collected for further analysis.

### Immunoblotting

Whole-cell lysates were prepared using 8 M Urea lysis buffer (Cell signaling Technology). The protein concentration was detected by Coomassie Brilliant Blue using a microplate spectrophotometer

(Infinite M200 PRO, Tecan). Equal amounts of total protein (50 μg) from cell lysates were loaded on a 10% SDS/PAGE gel, transferred to a PVDF membrane (Merck Millipore), and detected using an ECL Western Blotting Detection System (Bio-Rad). Beta-Actin was used as the loading control. Goat anti-rabbit IgG-HRP (1:5000, abcam, catalog No: ab6721)or goat anti-mouse IgG-HRP (1:5000, abcam, catalog No: ab6789) were used as the secondary antibodies. Antibodies: Il1β (1:1000, CST, catalog No:12703), Il18 (1:1000, CST, catalog No:57058), Nek7 (1:1000, CST, catalog No:3057S), Nlrp3 (1:1000, CST, catalog No:15101), Pro-caspase (1:1000, CST, catalog No:2225S), β-actin (1:1000, CST, catalog No:3700S).

### T cell stimulation assay with macrophages

Alveolar and lung-recruited macrophages were isolated and sorted following the same protocols as mentioned above. Then the macrophages were plated at 2.5E5 cells/well on 96-well plates and cultured for 2 h with DMEM containing 20% FBS, penicillin/streptomycin (100 U/ml) and 2 mM glutamine at 37 °C in a $CO_2$ incubator. Then, the anti-Il18 antibody (1:2000, BE0237, BioXCell) in PBS was added to each well. After 1 h, CD4$^+$ T cells isolated from the spleen through mechanical disruption were added to and co-cultured with the macrophages for 30 min in the incubator; the resulting T cells and macrophages were then stimulated with anti-CD3/CD28 antibody. After 48 h, the supernatant was harvested and IFN-γ in the supernatant was quantified by performing Elisa essay (Invitrogen, REF: 88-7314-88).

### Enzyme-linked immunosorbent assay (ELISA)

The concentration of IFN-γ was assessed by ELISA kits (USCN, China) according to the manufacturer's instructions. Briefly, samples were added into wells on an anti- IFN-γ microplate and incubated at 37 °C for 2 h. Then, the detector antibody was added to each well and incubated at 37 °C for 1 h. After three washes, 100 ml of the conjugate was added to each well and incubated at 37 °C for 1 h. After five washes, samples were measured immediately using a microplate reader (Bio-Rad Laboratory, Hercules, CA, USA) with absorbance at 450 nm.

### Trans-well assay

The migration ability of Raw264.7 was determined by carrying out a trans-well assay. Kupffer cells separated from the liver were plated at 4E5 cells/well on 24-well plates and cultured overnight with DMEM containing 20% FBS, penicillin/ streptomycin (100 U/ml) and 2 mM glutamine at 37 °C in a $CO_2$ incubator. Then the Kupffer cells were subjected to being stimulated by LPS (Invitrogen, 10 μg/ml), and medium with PBS was set as a control at the same time. After 6–8 h, supernatant of the Kupffer cells was harvested and added to the lower wells of the chambers of the migration assay device used (5 μm pore size, BD); meanwhile RAW264.7 macrophage cells were plated to the upper wells of the trans-well chambers at 2.5E5 cells/well in serum-free DMEM medium. After 36 h, RAW264.7 cells on the lower surface of the membrane were fixed in 4% paraformaldehyde and stained with Giemsa (Sigma Chemical Company, MO, USA). Cells in five microscopic fields were counted and photographed.

A549 invasion assays were also performed. Here, resident macrophages or recruited macrophages were cultured and treated with BMST or PBS in the lower chamber at 5E5 cells/well for 12 h before A549 cells were seeded on the upper chamber (8-μm pore size, BD) coated with Matrigel at 2E5 cells/well in serum-free DMEM medium. After 36 h, the cells on the lower surface of the membrane were fixed in 4% paraformaldehyde and stained with Giemsa (Sigma Chemical Company, MO, USA). Cells in five microscopic fields were counted and photographed.

### Primary cell sample preparation for LC-MS/MS

For each type of primary isolated macrophage and RAW264.7, equal numbers of cells (1.5E6) were collected, and the same amounts of protein (50 μg) were extracted by using 8 M urea (PH 8.0) with protease inhibitor (Pierce$^{TM}$, Thermo Fisher Scientific). Then, the proteins were subjected to trypsin digestion at an enzyme/protein mass ratio of 1:50 overnight following the FASP procedure[57]. Briefly, the proteins were subjected to reductive alkylation with dithiothreitol for 30 min at 56 °C and iodoacetamide (IAA) for 30 min at room temperature in the dark. Then, the samples were loaded into 10 kDa Microcon filtration devices (Millipore), and the urea was diluted and replaced by $NH_4HCO_3$ gradually after carrying out centrifugation twice with 50 mM $NH_4HCO_3$.

To obtain the global proteome, the peptides were separated into six fractions by carrying out the first dimension RPLC (sRP) before LC/MS analyses described in the next section. Briefly, the dried peptides were dissolved in $NH_4HCO_3$ (10 mM, pH 10.0) and loaded onto a homemade small C18 RP column (3 mg packing (3 μm, 150 Å, Agela)) in a 200 μl tip (T-400, Axygen) and eluted sequentially with nine elution buffers of increasing concentration (6, 9, 12, 15, 18, 21, 24, 30, and 35% of acetonitrile (ACN) in $NH_4HCO_3$ (10 mM, pH 10.0)). Finally, the nine fractions were combined into six fractions (6% + 24%, 9% + 30%, 12% + 35%, 15%, 18%, 21%) for the LC-MS/MS measurement.

### Tissue sample preparation of primary cell for LC-MS/MS

For the eight tissue samples, 100 μg of proteins of each tissue were extracted by 8 M urea with protease inhibitor. Then, the proteins were subjected to trypsin digestion at an enzyme/protein mass ratio of 1:50 overnight following the FASP procedure. Specifically, the proteins were subjected to reductive alkylation with dithiothreitol for 30 min at 56 °C and iodoacetamide (IAA) for 30 min at room temperature in the dark. Then, samples were loaded into 10 KD Microcon filtration devices (Millipore), and the urea was diluted and replaced by NH4HCO3 gradually after centrifugation for twice with 50 mM NH4HCO3. Proteins were digested with trypsin overnight at 37 °C. Finally, purified peptide was acquired after extraction with 50% ACN and 0.1% FA following desalination with 0.1% FA. The peptides were dried in a vacuum at 60 °C before LC-MS/MS analysis. Related to Fig. 4, Supplementary Figs. 8–11 and Supplementary Data 7 and 8.

### Cell and culture supernatant sample preparation for LC-MS/MS

For the samples with a small count of cells (lower than 1E6), such as samples in the LPS stimulation experiment, the protein digestion was carried out using an in-solution digestion procedure with trypsin and sodium deoxycholate (DOC) in a 37 °C incubator overnight, and then the peptides were extracted using 1% formic acid (FA) and desalination. For secretome investigation, the supernatants of Kupffer cells were collected and centrifuged at 10,000 × g at 4 °C for 20 min to remove cell debris. Then, the proteins were collected through acetone precipitation at a supernatant/acetone volume ratio of 1:3 at −20 °C overnight and digested by trypsin overnight at 37 °C. Finally, the product of the trypsin digestion was subjected to extraction with 50% ACN and 0.1% FA followed by desalination to obtain the purified peptides. Related to Figs. 7h and 8e and Supplementary Fig. 12, and Supplementary Data 9–11.

### LC-MS/MS analysis

For the 12 types of macrophages, the peptides were separated based on their different hydrophilic properties into six fractions using sRP as mentioned above, and were subjected each in the same manner to LC-MS/MS analysis using Orbitrap Fusion Lumos apparatus (Thermo Fisher Scientific) coupled with an Easy-nLC 1000 nanoflow liquid chromatography system (Thermo Fisher Scientific). Dried peptide was re-suspended in loading buffer (0.1% FA in water) and loaded onto a C18 trap column (100 μm × 2 cm, homemade; particle size, 3 μm; pore size, 120 Å; SunChrom, USA) with a maximum pressure of 280 bar using solution A (0.1% FA in water). Then, the peptides were separated on a C18 separation column (150 μm × 12 cm, homemade; particle size,

1.9 μm; pore size, 120 Å; Dr. Maisch, Ammerbuch, Germany) with a gradient of 5–35% mobile phase B (80% acetonitrile and 0.1% FA), and adjusted as a series of linear gradients following the different hydrophilic properties of six fractions, respectively, at a flow rate of 600 nl/min for 75 min. The MS analysis was performed on an Orbitrap mass analyzer by scanning $m/z$ values from 300 to 1400 and a resolution of 120,000 at 200 $m/z$. An automatic gain control (AGC) target value of $5 \times 10^5$ was used, with a maximum injection time of 50 s for full scans. The top-speed mode was selected in Quadrupole with a 1.6 $m/z$ window and fragmented by higher energy collisional dissociation (HCD) at a normalized collision energy of 35%. Then measurements were taken using ion trap analyzer with an AGC target of $5 \times 10^3$ and a maximum injection time of 35 ms for MS/MS scans. Finally, the dynamic exclusion time was set at 18 s, and data were acquired by Xcalibur software 2.2 (Thermo Fisher Scientific).

For the eight tissue samples and cell supernatant, peptides without prefractionation were subjected to LC-MS/MS analysis by Orbitrap Fusion (Thermo Fisher Scientific) with the same liquid chromatography system, trap column, MS analysis conditions, etc., as described above. The only exception is that the C18 separation columns here were 150 μm × 30 cm with a gradient of 5–35% mobile phase B (80% acetonitrile and 0.1% FA) at a flow rate of 600 nl/min for 150 min.

## Peptide and protein identification

MS raw files were processed with the "Firmiana" (a one-stop proteomic cloud platform)[28] against the mouse RefSeq protein database (updated on 04-07-2013) maintained by the National Center for Biotechnology Information (NCBI). The maximum number of missed cleavages was set to 2. Mass tolerances of 20 ppm for precursor and 0.5 Da for production were allowed. The fixed modification was cysteine carbamidomethylation and the variable modifications were N-acetylation and oxidation of methionine. For the quality control of protein identification, the target-decoy-based strategy was applied to confirm that the FDR (False Discovery Rate) of both peptide and protein were lower than 1%. The program percolator was used to obtain the probability value ($q$ value), and showed that the FDR (measured by the decoy hits) of every peptide-spectrum match (PSM) was lower than 1%. Then all peptides shorter than seven amino acids were removed. The cut-off ion score for peptide identification was 20. All of the PSMs in all fractions were combined for protein quality control, which was a stringent quality control strategy. The $q$ values of both target and decoy peptide sequences were dynamically increased until the corresponding protein FDR was less than 1% employing the parsimony principle. Finally, to reduce the false positive rate, the proteins with at least one unique peptide were selected for further investigation. The protein confidence and specific Exp. $q$ value ($q < 0.01$ for high-confidence threshold, $0.01 < q < 0.05$ for medium confidence threshold, and $q > 0.05$ for low confidence threshold) for each identification were clearly shown in Supplementary Data 1.

## Label-free-based MS quantification of proteins

The one-stop proteomic cloud platform "Firmiana"[28] was further employed for protein quantification. Here, the identification results and the raw data from mzXML file were loaded into the Firmiana platform. Then for each identified peptide, the XIC (extracted-ion chromatogram) was extracted by searching against the MS1 based on its identification information, and the abundance was estimated by calculating the area under the extracted XIC curve (AUC). For protein abundance calculation, the nonredundant peptide list was used to assemble proteins following the parsimony principle. Then, the protein abundances were firstly corrected by deploying a traditional label-free, iBAQ algorithm, which used number of theoretical peptides (tryptic digested peptides with length 7–30 amino acids) to correct the differences in signal intensity caused by protein size and sequence.

## Protein copy number for primary cells and FOT for tissue/culture supernatant calculation

For proteome datasets of the 12 macrophage populations, we applied the proteomic ruler[31] approach using a homemade Python script to estimate the copy number values of proteins in the macrophages based on iBAQ values. On the basis of assuming that the total mass of histones is approximately equal to the total mass of DNA, this approach uses the total corrected intensity of histones in each sample to estimate copy number per cell for every protein. To evaluate the accuracy of iBAQ-based copy number values in our current study, we also calculated the copy number based on the raw intensity. The result demonstrated that the iBAQ and intensity-based copy number were highly correlated with each other (with the average Pearson correlation coefficient $r = 0.94$) (Supplementary Fig. 14).

For tissue proteome and cell secretome quantification, the fraction of total (FOT), a relative quantification value that was defined as the iBAQ of a particular protein divided by the total iBAQ of all identified proteins in one experiment, was calculated as the normalized abundance of the particular protein so that its abundance can be compared across experiments. Finally, the FOT was further multiplied by 1E7 for easy presentation.

As for data integration, the proteins that had at least one unique peptide in each MS experiment were selected and merged into the proteome data matrix according to gene symbol, with null values left as NA. Three repeats of proteome data were used for protein quantification. The differential expression proteins (DEPs) presented in Figs. 4c, e and 5c–f were selected as the following criteria: (1) For proteins identified in both populations, the proteins with the threshold of Fold change ≥ 5 and $p$ value <0.05 (two-sided Student's $t$ test) were selected as DEPs; (2) For proteins for which the fold change or $p$ value could not be calculated due to null values, the proteins that were stably identified three times (with Coefficient of variation value CV ≤ 0.6) in one population but not identified in another population were selected as DEPs. In the heatmap or plot of Figs. 1h and 4c, e, the mean values for the copy numbers of three proteomic analyses of each macrophage population were calculated and shown.

## Total RNA extraction

Total RNA was extracted from the cells using Trizol (Invitrogen, Carlsbad, CA, USA) according to manual instruction. About 1E6 cells were collected in a 2 ml tube with 1 ml Trizol, followed by being homogenized for 2 min and rested horizontally for 5 min. The mix was centrifuged for 5 min at 12,000 × $g$ at 4 °C, then the supernatant was transferred into a new EP tube with 0.3 ml chloroform/isoamyl alcohol (24:1). The mix was shacked vigorously for 15 s, and then centrifuged at 12,000 × $g$ for 10 min at 4 °C. After centrifugation, the upper aqueous phase where RNA remained was transferred into a new tube with equal volume of supernatant of isopropyl alcohol, then centrifuged at 14,000 × $g$ for 20 min at 4 °C. After deserting the supernatant, the RNA pellet was washed twice with 1 ml 75% ethanol, then the mix was centrifuged at 14,000 × $g$ for 3 min at 4 °C to collect residual ethanol, followed by the pellet air dry for 5–10 min in the biosafety cabinet. Finally, 25–100 μl of DEPC-treated water was added to dissolve the RNA. Subsequently, total RNA was qualified and quantified using a Nano Drop and Agilent 2100 bioanalyzer (Thermo Fisher Scientific, MA, USA).

## mRNA library construction

Oligo (dT)-attached magnetic beads were used to purify mRNA. Purified mRNA was fragmented into small pieces with fragment buffer at the appropriate temperature. Then First-strand cDNA was generated using random hexamer-primed reverse transcription, followed by second-strand cDNA synthesis. afterward, A-Tailing Mix and RNA Index Adapters were added by incubating to end repair. The cDNA fragments obtained from the previous step were amplified by PCR, and products

were purified by Ampure XP Beads, then dissolved in EB solution. The product was validated on the Agilent Technologies 2100 bioanalyzer for quality control. The double-stranded PCR products from the previous step were heated denatured and circularized by the splint oligo sequence to get the final library. The single-strand circle DNA (ssCir DNA) was formatted as the final library. The final library was amplified with phi29 to make DNA nanoball (DNB) which had more than 300 copies of one molecular, DNBs were loaded into the patterned nanoarray and single end 50 bases reads were generated on MGI-SEQ2000 platform (The Beijing Genomics Institute, BGI).

### mRNA-seq data analysis
RSEM version 1.3.3 was used to quantify transcripts. First, the RSEM reference was generated by running rsem-prepare-reference against mouse reference genome (GRCm38); Then, the FPKM (fragments per kilobase of transcript per million mapped reads) was calculated by rsem-calculate-expression which handles both the alignment of reads against reference transcript sequences and the calculation of expression levels of each gene.

### mRNA copy numbers quantification
A similar principle of the "proteomic ruler" approach for protein quantification was also used for transcript copy number calculation[31]. Specifically, ribosomal RNA typically represents about 80% of total RNA[58], and in eukaryotic ribosomes, there is a ratio of about 1:1 between RNA and protein[59]. According to the ribosomal proteins' total copy number, the total cellular RNAs' copy number was calculated. Then, with the FPKM (Fragments Per Kilobase of transcript per Million mapped reads) value in the RNA-seq datasets, we calculated the copy number per cell for each RNA as the ratio of each RNA to the total cellular RNAs' copy number.

### Quality control of the proteome and transcriptome data
For the quality control of the triplicate and duplicate proteome and triplicate transcriptome data, the pairwise Pearson's correlation coefficients were calculated for all repeated runs using $\log_{10}$ transformed values in the statistical analysis environment R version 4.0.0.

### Bulk RNA-seq datasets screening for data comparison in published meta-analysis studies
In the meta-analysis research published by Summers et al.[22], We carefully screened and discarded 6 low-input RNA-seq data (GSM2784578, GSM2784579, GSM2784580, GSM2784588, GSM2784589, GSM2784590) and 6 Chip-seq data (GSM3983822, GSM3983823, GSM3983824, GSM3983825, GSM3983826, GSM3983827) because of their distinct data structure. Finally, a total of 249 bulk RNA-seq data of macrophage populations derived from the brain, liver, lung, spleen, small intestine, large intestine and peritoneal cavity were included in the study. Classical macrophages, including microglia, alveolar macrophage, Kupffer cells, spleen red-pulp macrophages, small and large intestinal macrophages, and peritoneal macrophages, were clearly defined in the original meta-analysis matrix. In order to evaluate our transcriptome of recently tissue-recruited macrophage populations, the following RNA-seq datasets were reclassified: firstly, we included nine datasets of liver macrophages in the meta-analysis data matrix as liver recruitment of macrophage populations, because these cells were identified as poorly differentiated and immature recruited liver monocytes in the original study[23]. Secondly, the cells labeled as pulmonary interstitial macrophages in the original studies[14,60] were compared with the lung-recruited macrophages, because these cells were thought to be derived from monocytes, and we considered that datasets may be relatively similar to lung monocytes. Thirdly, cells labeled as "splenic monocytes" in the meta-analysis data matrix were referred to as spleen-recruited macrophages. The classification of macrophages is clearly indicated in Supplementary Data 3 (with the column of "cell type").

### ChIP-seq and ATAC-Seq data collecting and processing
Briefly, raw ChIP-seq and ATAC-seq data were downloaded from GEO database (https://www.ncbi.nlm.nih.gov/geo) with the GEO accession number of GSE63339 and GSE63338, respectively, according to one study describing the enhancer landscape of the seven tissue-resident macrophage populations[37]. Processing of ChIP-seq and ATAC-seq data was carried out as described in previous study[37]. The raw reads were aligned to the mouse reference genome (UCSC mm10) using Bowtie2 (v2.3.4) with default parameters. To identify regions of enrichment and peaks, stacked by ChIP-seq reads of H3K4me1, we used the HOMER package makeTagDirectory followed by the findPeaks command with the histone parameter. We limited our chromatin analysis to high-confidence regions where the read density of both replicates was within the top 25th percentile and greater than two times the density of input reads (background). We overlapped the ATAC-seq and the ChIP-seq enriched peaks regions, and then extracted the sequence in the overlapped regions for motif analysis using known motif results in HOMER package.

### Weighted gene co-expression network analysis (WGCNA)
Proteins were clustered into functional modules using a WGCNA[40], with the R package WGCNA (1.71). All of the identified proteins were subjected to analysis. Standard parameters were changed to a soft threshold at power of 12 (based on scale free topology model fit, $R^2 = 0.85$), a "unsigned" network, average clustering, and a minimum module size of 30. The algorithm assigned 11,298 proteins to 40 modules (with Pearson correlation coefficients between module eigengene and trait, moduleTraitCor $\geq 0.5$) containing 35–1081 proteins as shown in Fig. 2. Only proteins with GS levels $\geq 0.6$ and MM $\geq 0.5$ were subjected to gene annotation analysis. Networks were exported to Cytoscape 3.8.0 for further visualization.

### Function annotation analysis
Protein modules, cell type signatures, and proteins participating in macrophage-tissue crosstalk were functionally characterized by carrying out an annotation enrichment analysis. For all three cases we used protein annotations from the GO (http://geneontology.org) database and KEGG (https://www.genome.jp/kegg/) with the R Bioconductor package "clusterProfiler" (R package v3.14.3). Enrichment scores were determined using one-sided Fisher's exact test. Both tests were corrected for multiple hypotheses using a Benjamini-Hochberg false discovery rate of 5%, if not stated otherwise.

### Hierarchical clustering and PCA
Unsupervised clustering was performed using the R package pheatmap (version 1.0.12). Briefly, the distances between the rows or columns of a data matrix were computed based on the Euclidean distance. The "complete" method was used in the agglomeration process. PCA was performed to visualize separation of different macrophage populations in the statistical analysis environment R version 4.0.0.

### Protein-protein interaction annotation
We used the STRING database (https://cn.string-db.org) to explore the protein-protein interactions between TFs in 12 macrophages. Each TF was annotated with the number of edges and the cell-specificity score (CSPS derived from TSPS)[61]. Briefly, the CSPS was computed based on the relative entropy, and specifically using the formula: $CSPS_i = \sum_{j=1}^{n} f_j^i \log_2(f_j^i/q^i)$. In this equation, $f_j^i$ denotes the fractional expression level of TF $i$ in macrophage $j$, and was computed as the ratio of the TF expression level in macrophage $j$ to its sum total expression level across all 12 macrophages; $q^i$ denotes the fractional expression of TF $i$ under a null model assuming uniform expression across macrophages. According to that definition, a minimum CSPS of 0 would be reported for TFs expressed uniformly across all macrophages, while a

maximum CSPS $\cong$ 3.59 would be reported for TFs expressed only in a single macrophage.

## TF classification

We defined a TF as a "specific" TF if its expression level in a certain macrophage was found to be 5 times that of the median of expression levels in the 12 tested macrophage populations, and belonged to the CTMs with the cut-off of GS ≥ 0.6 and MM ≥ 0.5. We defined a TF as a cell-type maintenance TFs (ctmTFs) if it satisfied the following two conditions in one type of macrophage: (1) it was found to be specific TF in one macrophage population. (2) The average $z$-score of the expression of its TG was greater than the max value of randomly selected proteins from the proteome data for 1000 times. The TF-TG regulatory network/relationship was archived from CellNET[45] database (http://cellnet.hms.harvard.edu).

## Crosstalk network

The method named CCCEXPLOR[48] was used to derive the hierarchical crosstalk network between macrophages populations and relevant tissue. Briefly, the KEGG signaling pathway was selected to be part of the crosstalk network if the (1) pathway included the interactions between tissue-type-specific ligands and receptors that were identified in the corresponding macrophage, and (2) the region downstream of the pathway was activated. The ligand-receptor interactions were downloaded from the DLRP[62] (http://www.hprd.org) and IUPHAR[63] (http://www.guidetopharmacology.org) databases. Mouse genes were mapped to human orthologs using the MGI database. Only the interactions between tissue-type-specific ligands and receptors that were identified in the corresponding macrophage with active downstream KEGG signaling pathways were selected. Tissue-type-specific ligands were identified based on the ratio of protein expression level in the particular tissue to the average level in the eight studied tissues being at least 1.5-fold. Active signaling pathways were defined as those with cell-type-specific TFs (with the expression level in the particular macrophage population being 1.5 times that of the average level in the 12 types of macrophages) and significantly enriched using pathway nodes between the receptors and TFs that could be detected in the proteomic profile (hypergeometric test, $p$ value <0.05).

## HE staining

Mouse lung and liver tissues were fixed in 10% formalin for histological and histomorphometric assessment, embedded in paraffin, then cut into 3-μm-thick section (Leica, RM2235). After deparaffinization and rehydration, 3 μm longitudinal sections were stained with hematoxylin solution for 5 min followed by 5 dips in 1% acid ethanol (1% HCl in 70% ethanol) and then rinsed in distilled water. They were next stained with eosin solution for 3 min followed by dehydration with graded alcohol and cleaning in xylene. The mounted slides were then examined and photographed using digital whole Slide Scanner (Leica, Aperio AT2).

## Data portal

We established a data portal [http://macrophage.mouseprotein.cn] to publish and describe our proteome and transcriptome data of the 12 macrophage populations. The data portal contains four main parts, including "explore", "compare", "analyze" and "download". The explore section consists of quantification value of 12,316 proteins and 16,413 gene transcripts of these macrophage populations. Users can search for the global proteome/transcriptome or specific gene expression patterns among the 12 macrophage populations in the "explore" section. In the "compare" section, the differences between proteome/transcriptome pattern of any two types of macrophage populations can be visualized interactively with a volcano plot. Quantification values of the differentially expressed proteins/transcripts between the two macrophage populations are easily accessible

for users. The "analyze" section contains eight main sets of results of this study with corresponding figures and datasets. In the 'download' section, we provided a link to directly download the datasets in this study for the user of the portal to analyze.

## Reporting summary

Further information on research design is available in the Nature Portfolio Reporting Summary linked to this article.

## Data availability

All data generated in this study, including the raw files and quantitative data matrix of proteomes and transcriptomes, have been deposited online. Specifically, proteome datasets of the 12 macrophage populations and the proteome profiles of the eight tissue/organs have been deposited to iProX with accession number IPX0001245000, and can be archived through accession number PXD021583 in PRIDE database. Proteome datasets of the macrophages in the liver and lung in wild-type or $Il18^{-/-}$ mice (related to Figs. 7 and 8) have been deposited to PRIDE with accession number PXD021657. The RNA-seq data of the 12 macrophage populations are accessible in SRA with accession number PRJNA482293. The bulk RNA-seq datasets for data comparison between published and our datasets were derived from meta-analysis research published by Summers et al.[22]. Raw ChIP-seq and ATAC-seq data were downloaded from GEO database with the GEO accession numbers GSE63339 and GSE63338, respectively, according to one study describing the enhancer landscape of the seven tissue-resident macrophage populations[37]. The ligand-receptor interactions were downloaded from the DLRP[62] [http://www.hprd.org] and IUPHAR[63] [http://www.guidetopharmacology.org] databases. Source data are provided with this paper.

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

## Acknowledgements

This work is supported by National Key R&D Program of China (2022YFA1303200, 2022YFA1303201), National Key R&D Program of China (2020YFE0201600, 2018YFA0507501, 2018YFE0201603, 2018YFE0201600, 2018YFA0507500, 2017YFA0505102, 2017YFA0505100, 2017YFA0505101, 2017YFC0908404, 2016YFA0502500), Shuguang Program of Shanghai Education Development Foundation and Shanghai Municipal Education Commission (19SG02), CAMS Innovation Fund for Medical Sciences (CIFMS) [2019-12M-5-063], National Natural Science Foundation of China (32201212, 31770886, 31770892, 31972933, 31700682, 81200355, 82272166), Chinese Ministry of Science and Technology (2016YFA0502500), National Program on Key Basic Research Project (973 Program, 2014CBA02000), National Institute of Health (Illuminating Druggable Genome, Grant U01MH105026), National Postdoctoral Program for Innovative Talents, The China Postdoctoral Science Foundation (2019M651268), The Major Project of Special Development Funds of Zhangjiang National Independent innovation Demonstration Zone (ZJ2019-ZD-004), Sponsored by Program of Shanghai Academic/Technology Research Leader (22XD1420100), Grant for Shanghai Municipal Science and Technology Major Project (2017SHZDZX01), International Science & Technology Cooperation Program (2012DFB30080), CAS-Youth Innovation Program Association (2016246), Shanghai Natural Science Foundation (18ZR1446300), and The Fudan original research personalized support project. In addition, we would like to thank the National Center for Protein Science Shanghai for the FACS analysis and we are grateful to Shufang He, Shuai Li, Yanke Wang, Jian Gao, and Jingxuan Han for their help.

## Author contributions

C.D., F.C.H., and W.J.Y. conceived the project and designed experiment. J.B.Q., Y.Z.W., F.Z., Z.Y.Q., K.L., W.H.S., L.S., M.W.L., T.G., X.W., Y.H., J.L., M.J.S., B.L.L., X.S.Z., and Y.X.T. conducted the experiments. J.B.Q., Y.L., and Y.Z.W. performed data analysis. C.D., J.B.Q., Y.L., Y.Z.W., and F.Z. wrote the manuscript. R.B.Z., P.H., C.S.D., Y.W., J.Q., and F.C.H. carefully edited the manuscript.

## Competing interests

The authors declare no competing interests.
