## [Peer Review File · Nature Communications]

Integrated proteomic and transcriptomic landscape of macrophages in mouse tissuesREVIEWER COMMENTS

Reviewer #1 (Remarks to the Author):

The manuscript by Qie et al is a proteomics tour de force to characterize macrophages from various mouse tissues. Overall the quality of the proteomic data and characterization is very strong. And while extensive expression profiling experiments manuscripts are present in the literature, this proteomics based data set is of significant value to the field.

Reviewer #2 (Remarks to the Author):

This manuscript described the profiling of the transcriptome and proteome patterns of bone-marrow derived macrophages (BMDM) and 10 primary macrophage populations from seven mouse tissues, including the brain, lung, liver, spleen, small intestine, large intestine, and peritoneum as well as a macrophage cell line. A large scale omics data was generated and explored by bioinformatic and biostatistical analyses. The data sets and the analyses provide resources for the understanding of microphase populations. The manuscript may be acceptable after addressing the following minor concerns.

Both raw data and processed data from transcriptomic and proteomic analysis should be made available in public repositories such as Scientific Data.

For proteomic data with 1% FDR, was it in protein level or peptide level? The FDRs for peptide level and protein level are generally different.

Three replicates of macrophage population were analyzed by proteomics, how reproducible of the three population analyses, Was one of the three proteomic analyses used for the quantification or all three were used? How was the data merged together?

Reviewer #3 (Remarks to the Author):

This paper is a complex analysis of a combined set of proteomic and transcriptomic data from several isolated mouse macrophage populations. It is a useful resource and there is some utility and novelty. But the novelty is not well-highlighted and set within existing knowledge and the short-comings are not recognised.

Specific comments.

The introduction really does not set the scene. It is a very dated and selective summary of the current literature on the phenotypic diversity and ontogeny of mouse mononuclear phagocytes. There is a need to recognize that almost all of the available information is mouse-specific and there is significant divergence between species. The study uses C57BL/6J mice. The authors need to say precisely which sub-strain, since there are significant differences (e.g. C57BL/6J has a deletion of the NNT gene, and there is a prevalent duplication of Dock2 in some commercial sub-strains; I suspect this is C57BL/6N, since NNT is highly-expressed). It is also relevant to note the major difference amongst mouse strains, since most public data is derived from C57BL/6J and there are quite radical differences between mouse strains (see PMID:31959980; 29779944). One should note that RAW264.7 is not a C57BL/6 line (it is BAB-14, related to BALB/c and it is capitalized, not

Raw264). References 5-10 are cited as dogma, but it is actually not the case that all tissue macrophages are derived from embryonic precursors and certainly not strongly supported that high F4/80 is a marker for embryo-derived cells (note the gene name is now Adgre1, not Emr1). Even in C57Bl/6J mice, many populations are progressively replaced from HSC-derived blood monocytes. I would suggest reading and citation of more recent literature and reviews (e.g. PMID: 30579704; 33139489). There is also a massive amount of mouse macrophage RNA-seq data from C57Bl/6J mice in the public domain. A large scale meta-analysis of available mouse data was published recently, against which the authors might compare their data (PMID:33031383). This analysis reveals two important artefacts; the contamination of tissue macrophages with other cells and their activation during isolation. Whilst there is less extensive comparative proteomic data for mouse macrophages, there is a significant literature on single isolated populations and the authors should note the development of single cell approaches up front (e.g. PMID: 33504367; cited in discussion). The statement that “the crosstalk networks of macrophages and tissues in different organs are still not well understood” is nonsense and should be removed.

The results starts with a description of the nature of the populations chosen lacking detail and without any justification (and also does not indicate why the animals were fasted, which does impact on monocyte and tissue macrophages). The description of the culture, isolation and purification procedures is inadequate. How were the BMDM harvested? Were the peritoneal macrophages purified by adherence or FACS?

Line 152-154 states. “We systematically compared our datasets with those published epigenome and transcriptome data and observed high consistency of the macrophage signatures across the three omics levels (Supplementary Fig. 2e). I cannot see this comparison in the figure and the relationship to published data needs much better validation (e.g. against the large metadata in PMID:33031383)

Lines 168-180 deal with the correlation between mRNA and protein data. As the authors indicate, the relative lack of quantitative correlation is well-known and there are straightforward mechanistic and also technical explanations (not least, for example, that macrophages do not express mRNA encoding major contaminants like albumin). However, the protein cannot exist without the prior existence of the mRNA. So, at a more straightforward level, Fig 1e shows that there is actually a large set of detected transcripts (around 1/3) where the corresponding protein is not detected. This speaks to the limits of detection of proteomics. Amongst the transcripts that lack corresponding proteins I suspect there will be a substantial cohort of immediate early response genes (e.g. fos) and cytokines induced at the mRNA level during cell isolation. The authors need to analyse the proteins that were not detected and understand the reasons. Conversely, the significant cohort of proteins with no corresponding mRNA are most likely contaminants.

Lines 197-233 deal with WGCNA of a set of putative PRR pathway genes. The rationale for this analysis is entirely unclear, and the choice of parameters for WGCNA is not justified. My feeling is that the analysis is excessively fine-grained and the inferred association with specific PAMP recognition is not supported by any of the literature (see also FACS data on tissue macs in PMID: 32840578).

Lines 235-259 describe the response of isolated KC to LPS, CpG DNA and flagellin. This is irrelevant and uninterpretable. The cells are incubated for 24 hrs without added CSF1 (which is required for maintenance of protein synthesis and regulates expression of TLR4 and TLR9) during which time their differentiation status changes. The stimuli are added at an arbitrary concentration and incubated for an arbitrary time after previous incubation in serum-free medium for 2 hours.

Lines 286-315. Figure 3 shows the identification of protein coexpression modules to identify cell-type specific proteins. As a preliminary analysis, it would be worthwhile to highlight the detection of markers that are known to be cell-type specific in these populations from FACS

analysis. This is actually quite evident in the proteome database (e.g. CLEC4F, CD5L, VSIG4 and TIMD4 are all very highly-expressed by KC; P2RY12 in microglia). However, the analysis and an overview of the related Table highlights a major issue that compromises any interpretation. All of the preparations have quite high levels of endothelial markers (PECAM1, VWF) and the high level of albumin contamination varies, which compromises relativity estimates. The microglia-enriched clusters include major myelin-associated proteins (MOBP, PLP1) and neuronal membrane transporters that are absolutely not expressed by microglia. Similarly, the alveolar macrophages have high levels of surfactant proteins and the gut macrophages have high levels of known intestinal epithelial markers (e.g. VIL1) transporters. This may reflect endocytic activity, but it is not possible to separate this possibility from straightforward contamination and the entire inference of functions for these genes in macrophages are invalid. The mention of RAW264 neglects the fact that it is a different mouse strain and the difference cannot be attributed to the lack of an immune environment.

Lines 319-398 deal with transcription factors detected and inferred TF networks. This analysis is also difficult to interpret because of contamination and activation during isolation. Many of the conclusions reiterate what is known for specific transcription factors (which could be reviewed more adequately) but it is also of note what is missing from tissue-specific data (e.g Spic, Id3, Runx1, Irf8, Tfec, Mafb) and what is likely artefact (e.g Elf3 is massively-expressed in intestine, and entirely restricted to epithelial cells, see proteinatlas.org and biogps.org)

Lines 400-446 deal with the difference between recently recruited and tissue resident macrophages and subsequently with apparent differences in their response to LPS in vitro. This comparison is based upon an arbitrary assignment of F4/80 and CD11b bright and dim as markers, which I personally feel is strongly debatable (e.g. F4/80 is absent from marginal zone and lymphoid follicle populations in spleen, and low on AM in lung). The progressive differentiation of monocyte-derived cells in multiple organs has been analysed quite extensively by others, and I was not compelled by any new insights from this analysis. In figure 7A, Clec5a is certainly known to be monocyte-enriched, but MUC1 mRNA is barely detectable in any published isolated macrophage population, so this is very likely due to contamination.

The final section in Figure 8 focusses on the ability of tissue-resident macrophages to produce IL18. The recently published meta-analysis of macrophage gene expression profiles indicated that all tissue resident macrophages in C57BL/6 mice express Il18 mRNA constitutively. The final section of this paper deals with the impact of an IL18 knockout; these are inadequately described in the methods, which require a reference and details of the genetic background and the use of littermate controls. However, a cursory PubMed search on IL18 and knockout reveals >1000 hits, many dealing with inflammasome activation and the lung, so the originality of this work is unclear and needs to be placed in some kind of context.

The discussion is somewhat rambling and unrelated to the results. The M1/M2 paradigm was never useful and most M2 markers are known to be expressed by resident tissue macrophages. The discussion of tumor-associated macrophages concludes that the diverse functions of tissue-resident macrophages and recruited macrophages in carcinogenesis deserve further investigation. There is already a massive literature on this topic!!

I had a look at the on-line resource, which is said to “provide a valuable resource for carrying out novel investigations of function and mechanism studies of macrophages”. The resource is quite easy to use and potentially useful as a resource but could use a brief guide. It would be useful to be able to see the proteomic and transcriptomic data for a particular on the

same view and to ensure the search recognises synonyms and protein names (e.g. Adgre1, F4/80, Emr1).

Reviewer #4 (Remarks to the Author):

In this manuscript, author carried out experiments to profile the proteome and transcriptome patterns of 11 macrophage populations from eight tissues and the cell line RAW264.7, with 12,316 proteins and 16,413 gene transcripts identified. They subsequently analyzed the proteome and revealed some interesting features. The resulting data thus will provides a valuable resource for studying functions of macrophages. This is interesting and comprehensive study. Nevertheless, the authors should addressed the concerns below before this paper can be published.

--The rational for selecting primary macrophage from 8 tissues is not clear. Why not heart? The macrophage is critically important for fat function. The authors should carry out the proteomic analysis of the two types of macrophages and include the data into the paper.

--As authors suggested, proteomics studies were carried out previously in macrophage derived from liver, brain and heart. What are the major differences between this study and previous ones? Innovation in terms of methodology and biology?

--This author does not believe the claim that the quantified protein copy numbers spanned over nine orders of magnitude. This conclusion therefore need to be validated by an orthogonal method.

--Too many papers were cited. Suggest to reduce reference number.

--The paper should be carefully edited by a native English speaker.

Point-to-point Response

Reviewer #1 (Remarks to the Author):

The manuscript by Qie et al is a proteomics tour de force to characterize macrophages from various mouse tissues. Overall, the quality of the proteomic data and characterization is very strong. And while extensive expression profiling experiments manuscripts are present in the literature, this proteomics-based data set is of significant value to the field.

Response: Many thanks for the appreciation and positive comments.

Reviewer #2 (Remarks to the Author):

This manuscript described the profiling of the transcriptome and proteome patterns of bone-marrow derived macrophages (BMDM) and 10 primary macrophage populations from seven mouse tissues, including the brain, lung, liver, spleen, small intestine, large intestine, and peritoneum as well as a macrophage cell line. A large-scale omics data was generated and explored by bioinformatic and biostatistical analyses. The data sets and the analyses provide resources for the understanding of microphase populations. The manuscript may be acceptable after addressing the following minor concerns.

Q1: Both raw data and processed data from transcriptomic and proteomic analysis should made available in public repositories such as Scientific Data.

Response 1: Thanks for the constructive comments. All of the data generated in this study, including raw files, processed data, and quantitative data matrix of proteomes and transcriptomes, have been deposited online. Specifically, the proteome datasets of the 12 macrophage populations and eight tissue/organs have been deposited to iProX (<https://www.iprox.org.cn>) with accession number IPX0001245000/PXD021583 (and can be accessed with URL: <https://www.iprox.cn/page/PSV023.html?url=1645013458256qmyI>; Password: a0v3). The proteome datasets of the macrophage populations of the liver and lung in wild-type or *Il18*^{-/-} mice (related to Fig. 7 and Fig. 8) have been deposited to PRIDE (<https://www.ebi.ac.uk/pride>) with accession number PXD021657 (and can be accessed with account: reviewer_pxd021657@ebi.ac.uk; Password: UFZuY3Pj). The RNA-seq data of the 12 macrophage populations are accessible in SRA (<https://www.ncbi.nlm.nih.gov/sra>) with accession number PRJNA482293. Please see the 'Data availability' in the 'Materials and Methods' sections in the manuscript for details. To allow for public access of the resource, we established a data portal at <http://macrophage.mouseprotein.cn> (Figure CL1).

Figure CL1. Data portal for the proteome and transcriptome landscape of BMDM and 10 primary macrophage populations in mouse tissue and cell line RAW264.7.

Q2: For proteomic data with 1% FDR, was it in protein level or peptide level? The FDRs for peptide level and protein level are generally different.

Response 2: Thanks for the comments, we apologized for unclear description. In our current study, firstly, peptide identification stringency was set at a maximum 1% FDR at peptide level. Peptides (minimum length of seven amino acid residues) with 1% FDR and a Mascot ion score greater than 20 were selected for protein identification. Then, proteins with 1% FDR (with at least one unique peptide) were selected for further analysis. The same cutoff strategies of FDR at protein/peptide level have been widely used in recently published researches^{1,2}. The FDR algorithms were referred to previous published papers (PMID: 17327847)³. **Please see Page 5 and the ‘Peptide and Protein Identification’ in the ‘Materials and Methods’ sections of the manuscript for details.**

Q3: Three replicates of macrophage population were analyzed by proteomics, how reproducible of the three population analyses, was one of the three proteomic analyses used for the quantification or all three were used? How was the data merged together?

Response 3: Thanks for the comments. In this study, we measured three biological replicates for each population. The proteome results showed that **the three replicates were highly reproducible**: for each macrophage population, 88% proteins were detected in all 3 replicates, with the average Pearson correlation among replicates greater than 0.9 (**Figure CL2a-c, also see Supplementary Fig. 2 in the revised**

manuscript). For data integration, the proteins that had at least one unique peptide in each MS experiment were selected and merged into the proteome data matrix according to gene symbol. The null values over the datasets were set as 1 to facilitate analysis. As for data analysis, three repeats of proteome data were used for protein quantification or differentially expression genes (DEGs) selection (bilateral Student's t test). In the heatmap or plot of Figure 1h, 4c, and 4e, the mean values for the copy numbers of three proteomic analyses of each macrophage population were calculated and showed. Please see the 'Label-free-based MS Quantification of Proteins' in the 'Materials and Methods' sections of the manuscript for details (Page 33, Lines 920-925).

Figure CL2. (a) Venn diagram of the numbers of identified proteins in triplicated proteome dataset among 12 macrophage populations. (b) The Pearson correlation coefficients matrix among triplicated proteome data. (c) Principal component analysis (PCA) of the protein expression patterns of the 12 macrophage populations.

Reviewer #3 (Remarks to the Author):

This paper is a complex analysis of a combined set of proteomic and transcriptomic data from several isolated mouse macrophage populations. It is a useful resource and there is some utility and novelty. But the novelty is not well-highlighted Nd set within existing knowledge and the short-comings are not recognized.

Specific comments.

INTRODUCTION

Q1: The introduction really does not set the scene. It is a very dated and selective summary of the current literature on the phenotypic diversity and ontogeny of mouse mononuclear phagocytes. There is a need to recognize that almost all of the available information is mouse-specific and there is significant divergence between species. The study uses C57BL/6J mice. The authors need to say precisely which sub-strain, since there are significant differences (e.g., C57BL/6J has a deletion of the NNT gene, and there is a prevalent duplication of Dock2 in some commercial sub-strains; I suspect this is C57BL/6N, since NNT is highly-expressed). It is also relevant to note the major difference amongst mouse strains, since most public data is derived from C57BL/6J and there is quite radical difference between mouse strains (see PMID: 31959980; 29779944). One should note that RAW264.7 is not a C57Bl/6 line (it is BAB-14, related to BALB/c and it is capitalized, not Raw264).

Response 1: Many thanks for the constructive comments.

About the summary of macrophage research in introduction

We have updated the introduction part and relevant references to make the introduction more accurate and progressive. The note of species or mouse strains identities as well as the use of C57BL/6 as a model for macrophage heterogeneity research were recognized and highlighted at the beginning of 'introduction' and discussed in 'discussion' part of the revised manuscript. All changes are highlighted **in blue** in the revised manuscript (**Please see discussion in Page 21, Line 562-569 of the revised manuscript**).

About the sub-strain of C57BL/6 used in this study

Thanks for the strictness and attention of the reviewer. Here, we used C57BL/6N in our current study. Just as the reviewer mentioned, there are significant differences between the sub-strain of C57BL/6J and C57BL/6N. We measured the proteome and transcriptome of C57BL/6N simultaneously for precise multi-omics comparison, although the transcriptome data of primary macrophage populations in different mouse tissues were published previously (for C57BL/6J). We carefully introduced the use of 'C57BL/6N' in the materials and methods part in the revised manuscript (**Please see Page 23 in the revised manuscript**).

About the concern of cell line RAW264.7

In order to investigate the proteome identities between primary cells and immortal cell lines, we performed the LC/MS-MS analysis of RAW264.7. In view of the reviewer's suggestion about murine origin of RAW264.7, we rephrased the explanation for the specificities showed by proteome data, removed the statement such as 'the lack of an immune environment'. Considering that no mouse macrophage cell lines derived from C57BL/6N are available for comparison with primary macrophage populations nowadays, we kept the proteome data of RAW264.7 in our study to provide a resource as a reference for comparison between immortal cell line and primary populations. We discussed the use of cell line RAW264.7 in the 'discussion' part of the revised manuscript (**Please see Page 20 in the revised manuscript**). The text of 'Raw264.7' in the full manuscript has been updated as 'RAW264.7'.

Q2: References 5-10 are cited as dogma, but it is actually not the case that all tissue macrophages are derived from embryonic precursors and certainly not strongly supported that high F4/80 is a marker for embryo-derived cells (note the gene name is now Adgre1, not Emr1). Even in C57Bl/6J mice, many populations are progressively replaced from HSC-derived blood monocytes. I would suggest reading and citation of more recent literature and reviews (e.g., PMID: 30579704; 33139489). There is also a massive amount of mouse macrophage RNA-seq data from C57BL/6J mice in the public domain. A large scale meta-analysis of available mouse data was published recently, against which the authors might compare their data (PMID:33031383). This analysis reveals two important artefacts: the contamination of tissue macrophages with other cells and their activation during isolation. single

isolated populations and the authors should note the development of single cell approaches up front (e.g., PMID: 33504367; cited in discussion).

Response 2: Thanks for the comments. Here, we response this comment as follows:

About the citation and writing of the introduction

We apologize for the dogmatic reference citation in the introduction part in previous manuscript firstly. In the revision, we accurately referred to and quoted the papers in the field of macrophages research recently, especially the papers mentioned by the reviewer, and updated the content of the introduction part of the manuscript. The revisions mainly include: **First**, highlighting the species and strain diversities for macrophage heterogeneity research; **Second**, summarizing the studies and theories on the development and differentiation of tissue-resident macrophages, including origin diversities of tissue-resident macrophages; **Third**, expanding the vision filed of omics data on macrophage studies according to the papers published recently; **Fourth**, adding the development of macrophage research field based on single cell technology.

Please see in Page 2-3 of the revised manuscript for details.

About the use of F4/80hi as a marker for embryo-derived macrophages.

F4/80 was previously proposed as a marker of embryo-derived macrophages⁴, while was also highly expressed in intestinal macrophages which was demonstrated derived from blood monocytes⁵. The expression of F4/80 was eventually elevated during monocyte differentiation to occupy a vacant KC niche⁶.⁷ These restricted the usage of F4/80hi as marker for embryo-derived macrophages. Through systematically surveying the literatures, we found that despite the limitation of using F4/80hi as marker for embryo-derived macrophages, F4/80hi were broadly used to identify tissue-resident macrophages within one individual tissue. Following we list representative studies within recent five years using F4/80 or using F4/80hi coupled with CD11b^{lo} as markers to detect or sort tissue-resident macrophages. (1) Cai, B. *et al.* tested the differential expression of MerTK in Kupffer cells gated as F4/80hiCD11b^{lo} and liver-recruited macrophages gated as F4/80^{lo}CD11b^{hi} (**Figure CL3**, Cell metabolism, 2020, PMID: 31839486)⁸; (2) Aditya Ambade *et al.* assessed the percentage of Kupffer cells gated by F4/80hiCD11b^{lo} in the liver of cenicriviroc (CVC, CCR2/5 inhibitor)-treated alcohol-fed mice (**Figure CL4**, Hepatology, 2019, PMID: 30179264)⁹; (3) Zhiqing Li. *et al.* employed microglia and Langerhans cell as internal labeling controls respectively for early erythro-

myeloid progenitors (EMP) and late EMP using CD11b^{lo}F4/80^{hi} to track the development of the tissue-resident mast cells in multiple organs (Immunity, 2018, PMID: 30332630)¹⁰; (4) Huang. *et al.* detected monocyte-derived and tissue-resident macrophages with F4/80^{int} and F4/80^{hi} respectively in synovial tissue and screened their role in a mouse model of rheumatoid arthritis (Science Advances, 2021, PMID: 33523968)¹¹; (5) Park, Jun-Gyu. *et al.* searched the percentage of kidney-resident (rMac: CD11b^{lo}F4/80^{hi}) and kidney-infiltrating (iMac: CD11b^{hi}F4/80^{lo}) subsets (**Figure CL5**, Scientific Reports, 2020, PMID: 32973321)¹²; (6) Ito, Tomoya. *et al.* searched the cell barrier function of resident peritoneal macrophages with F4/80^{hi}CD206⁻ in post-operative adhesions (**Figure CL6**, Nature Communications, 2021, PMID: 33854051)¹³; (7) Howard *et al.* detected the percentage alter ratio of resident macrophages in Duchenne muscular dystrophy with the cell feature of F4/80^{hi} (**Figure CL7**, The American Journal of Pathology, 2021, PMID: 33497702)¹⁴.

Figure CL3. Flow cytometric analysis of cell-surface MerTK expression on resident (F4/80^{high} CD11b^{low}) and recruited (F4/80^{low}CD11b^{high}) macrophages in Nonalcoholic steatohepatitis (NASH) liver. Related to the Supplementary Figure 1A in the published study (Cell metabolism, 2020, PMID: 31839486)⁸.

Figure CL4. Liver mononuclear cells were isolated via tissue digestion followed by density gradient, surface staining for CD11b and F4/80 for differentiating liver macrophages by flow cytometry. Cells were gated based on size, singlets, an amino-reactive dye for living cells and for CD45+ positivity (Hepatology, 2019, PMID: 30179264)⁹.

Figure CL5. Gating strategy for kidney-resident and infiltrating macrophages (rMac and iMac, respectively, Scientific Reports, 2020, PMID: 32973321)¹².

Figure CL6. Representative flow cytometry plots of F4/80^{High}CD206⁻ macrophages (red) and F4/80^{Low}CD206⁺ macrophages in the peritoneal fluid after ischemic button creation in mice (Nature Communications, 2021, PMID: 33854051)¹³.

Figure CL7. Representative flow cytometry dot plots for CD45+CD11b+LY6G-F4/80lo monocytes, and CD45+CD11b+LY6G-F4/80hi macrophages in C57 and mdx quadriceps (QUADs) and diaphragm (DIA) skeletal muscles (The American Journal of Pathology, 2021, PMID: 33497702)¹⁴.

In summary, F4/80hi is certainly not a marker for embryo-derived macrophages although, F4/80hi could serve as marker for tissue-resident and recently tissue-recruited macrophage populations.

Therefore, to avoid misleading, we removed the replaced ‘F4/80 and CD11b bright or dim were set to distinguish tissue-resident macrophages (EMPs-derived macrophages) and tissue-recruited macrophages (monocyte-derived macrophages) in the lung, liver, and spleen’ with ‘F4/80 and CD11b bright or dim were set to distinguish tissue-resident macrophages and recently tissue-recruited macrophages in the lung, liver, and spleen’ in the revision. Besides, in the revision we have updated the gene symbol of F4/80 in manuscript and dataset with Adgre1. Once again, we would like to thank the reviewer for the comments, which made the article more accurate and accessible.

About cell purity/potential contamination in this study

The reviewer mentioned that the contamination of macrophages with other cells may serve as an important artefact. To solve the problem, we evaluated the purity of all macrophage populations through flow cytometer (FCM) with the same gating strategy. The result showed that purities of all 11 populations were greater than 98% (**Figure CL8**), indicating that the cell populations were not contaminated significantly. **We presented**

the purity of 11 populations in Supplementary Figure 1 in the revision.

About possible cell activation during separation in this study

The reviewer mentioned the possible of cell activation during isolation may serve as another artefact. As for this question, firstly, we separated and isolated all cell populations at a low temperature of 4°C overall process, and controlled processing time within 2-3 hours, so as to obtain the omics data that best reflects the state of cells *in vivo*. Admittedly, the possibility of inducing activation of a small number of cells during isolation cannot be ruled out, and the concern of cell activation during separation/isolation is inevitable nowadays. Previously published meta-analysis based on 466 published RNA-seq datasets of macrophage populations also indicated that ‘all tissue disaggregation and separation protocols activate MPS cells’⁷. Thanks for the constructive comment, we were aware of this question and discussed the influence of cell activation perturbation on gene expression quantifications in the discussion part in the revision (**Please see page 21-22, Line 584-595 in the revised manuscript for detail**). We will take note of the concern in subsequent research.

Experiment Repeats

In order to demonstrate the reliability of our proteome data, freshly sorted 6 macrophage populations with high purity, including tissue-resident and recruited macrophages from the lung, liver and spleen, were subjected to LC-MS analysis. The new proteome data was highly correlated to the previous data (with the average Pearson correlation coefficient $r = 0.9$), indicating the high repeatability of proteome (**Figure CL9, new data as the evidence to demonstrate the reliability of our previous data mainly, only included here for the referees’ information**).

Figure CL8. Representative flow cytometry dot plots indicating the sorting strategy (a) and high purity (b, greater than 98%) of 10 primary macrophages in 7 tissues as well as BMDM. **Related to Supplementary Figure 1 in the revised manuscript.**

Figure CL9. Pearson correlation coefficient of FOT values between new MS data and the previous MS data. (New data as the evidence to demonstrate the reliability of our previous data mainly, only included here for the referees' information)

Q3: The statement that “the crosstalk networks of macrophages and tissues in different organs are still not well understood” is nonsense and should be removed.

Response 3: Thanks for the comment. We apologized for the unclear statement. We removed the statement from manuscript in the revision.

RESULT

Q4: The results start with a description of the nature of the populations chosen lacking detail and without any justification (and also does not indicate why the animals were fasted, which does impact on monocyte and tissue macrophages). The description of the culture, isolation and purification procedures is inadequate. How were the BMDM harvested? Were the peritoneal macrophages purified by adherence or FACS?

Response 4: Thanks for the comments. In the revision, we have rewritten the beginning of the result section in the revised manuscript. The first paragraph of the results reads as following (Please see page 5 in the revised manuscript):

To probe the global spectrum of protein expression profiles among tissue-resident macrophages in homeostasis, we isolated 7 primary tissue-resident macrophage populations including brain microglia, lung alveolar macrophages, liver Kupffer cells, splenic red pulp macrophages, peritoneal macrophages, and small and large intestinal macrophage populations from C57BL/6N mice (8 to 12 weeks old) through enzymatic digestion and fluorescence-activated cell sorting (FACS). To identify global differences between tissue-resident and recently recruited macrophages, we isolated three tissue-recruited macrophage populations in the lung, liver, and spleen simultaneously. A total of 13 surface markers, including CD45, F4/80, CD11b, CD117, Siglec-F, CD11c, Cx3cr1, MHCII, CD64, CD115, CD24, B220, and Ly6g, were used for cell sorting according to the published literature (Supplementary Fig. 1a, Supplementary Table 14). F4/80 and CD11b bright and dim phenotypes were used to distinguish tissue-resident and recruited populations in the lung, liver, and spleen (Supplementary Fig. 1a). Evidence obtained by FACS analysis confirmed that the purities of the different macrophage populations were all greater than 98% (Supplementary Fig. 1b). The cell line RAW264.7 and BMDMs were also collected, as these populations, although maintained in culture, are widely used as immortal cell lines or models of macrophage biology (Fig. 1a).

Moreover, according to reviewer's suggestion, we added the detailed isolation protocol for each macrophage one by one, especially for BMDM and peritoneal macrophages. **Please see the 'Materials and Methods' part in Page 25-28 the revised manuscript for details.** The revised methods for BMDM and peritoneal macrophages acquisition are as following:

Peritoneal macrophages were isolated by carrying out peritoneal lavage. Mice were anaesthetized with CO₂ and quickly euthanized by cervical dislocation. Peritoneal cells were collected by lavages of the peritoneum with 5 ml ice-cold HBSS containing 1% FBS and 1mM EDTA. Then the cells were washed and resuspended in ice-cold staining buffer for later FACS antibody incubation and analysis.

BMDMs were obtained from bone marrow after 7 days of invitro culture in DMEM with 10% FBS and M-CSF, as described in published researches¹⁵. Specifically, Mouse bone marrow was harvested from femur and tibia bones using 21G needle and 10 ml syringe with 5 ml cold PBS containing 2% FBS. The marrow

was dissociated by passage through a 21G needle 4-6 times and a 100 µm cell strainer to obtain single cell suspension. The red blood cells (RBCs) were removed using ice-cold RBC lysis buffer for 2 min on ice. Then, the cells were pelleted through centrifugation at 500 g for 5 min at 4°C, washed once in PBS and resuspended in BMDM culture medium containing 10% FBS, 25 ng/ml mouse M-CSF (R&D, cat. no. 416-ML-050/CF), and 100 U/ml penicillin/streptomycin. Subsequently, the cells were seeded in a Petri dish with BMDM culture medium at 37°C in a CO2 incubator. Fresh BMDM culture medium was changed on day 3. After 6-8 days of culture, non-adherent cells were washed off with PBS for 2-3 times. BMDM was digested by trypsin into single-cell suspension, washed twice with PBS and resuspended in staining buffer for further FACS antibody incubation and analysis.

As for the question of animal fasting, it's our misuse and omission in the writing process. The mice were only fasted during the pre-test process of intestinal macrophages isolation, while the mice employed for formal experiment were not fasted. Thanks for the reviewer's reminding, and we feel sorry for the omission. Thus, we removed the description of 'the mice were starved overnight' in the revision.

Q5: Line 152-154 states. “We systematically compared our datasets with those published epigenome and transcriptome data and observed high consistency of the macrophage signatures across the three omics levels (Supplementary Fig. 2e). I cannot see this comparison in the figure and the relationship to published data needs much better validation (e.g., against the large metadata in PMID:33031383).

Response 5: Thanks for the comments. In the revision, we systematically compared our transcriptome data with the datasets of a large-scale meta-analysis recommended by the reviewer⁷ (PMID: 33031383) through correlation calculation and quantification evaluation of signatures (in different clusters suggested by the meta-analysis). As a result, the spearman correlation coefficients between the two datasets of the same macrophage population were around 0.75 (p value < 0.05), which was significantly higher than the correlation coefficients between the two datasets of different cell types (**Figure CL10a, b**). We also compared published macrophage proteome¹⁶⁻¹⁸ and our proteome datasets. The results showed that the average correlation coefficient between the different proteome data of same macrophages was 0.7, which was significantly higher than that of the different macrophages (**Figure CL10c**). Also, the comparison captured consistent expression

of cell-type specific signatures among published transcriptome, our transcriptome and our proteome datasets (Figure CL10d, e). The reference was carefully marked in the revised manuscript and figure legends. Data collection procedure and datasets relationships were described in the materials and methods in the revised manuscript. Please see page 7 and Supplementary Figure 4 for details.

Figure CL10. Comparison analysis between our datasets and published data. (a) The violin diagrams of Spearman correlation coefficients r between transcriptome of indicated macrophage populations in published meta-analysis⁷ and in our study. Grey columns show coefficients between transcriptome of indicated populations in our study and transcriptome of other populations (indicated populations excluded) in meta-analysis. $*p < 0.05$, $**p < 0.01$, $***p < 0.001$ (bilateral Student's t test). **(b)** The violin diagrams show correlations between our transcriptome and different published datasets of microglia/small intestinal

macrophage populations. The BioProject ID of published datasets for groups labeled with E1 to E5 in microglia diagrams are PRJNA529096, PRJNA421946&PRJNA422281, PRJNA507265, PRJNA529095, PRJNA506249. The BioProject ID of published datasets for groups labeled with E1 to E4 in small intestinal macrophage diagrams are PRJNA471340, PRJEB27719, PRJNA591465, PRJNA325288. * $p < 0.05$, ** $p < 0.01$, *** $p < 0.001$ (bilateral Student's t test). (c) Spearman correlation coefficients matrix of proteome between our datasets and published studies¹⁶⁻¹⁸. (f, g) Heatmap of expression profiles of indicative gene clusters (f) and represent signatures (g) defined by meta-analysis⁷ in the published datasets as well as our transcriptome and proteome datasets. Mean values for each gene at all populations are color-coded based on the z-scored TPM (for transcriptome) and copy numbers per cell (for proteome). **Related to Supplementary Figure 4 in the revised manuscript.**

Q6: Lines 168-180 deal with the correlation between mRNA and protein data. As the authors indicate, the relative lack of quantitative correlation is well-known and there are straightforward mechanistic and also technical explanations (not least, for example, that macrophages do not express mRNA encoding major contaminants like albumin). However, the protein cannot exist without the prior existence of the mRNA. So, at a more straightforward level, Fig 1e shows that there is actually a large set of detected transcripts (around 1/3) where the corresponding protein is not detected. This speaks to the limits of detection of proteomics. Amongst the transcripts that lack corresponding proteins I suspect there will be a substantial cohort of immediate early response genes (e.g., fos) and cytokines induced at the mRNA level during cell isolation. The authors need to analyze the proteins that were not detected and understand the reasons. Conversely, the significant cohort of proteins with no corresponding mRNA are most likely contaminants.

Response 6: Thanks for the comment. According to the reviewer's suggestion, we explored the types and functions of missing proteins (**Figure CL11**). As mentioned by the reviewer, a total of 4,269 protein-coding-genes (accounting for 1/3 of identified protein-coding transcripts) were only detected in transcriptome. Consistently, previous studies also showed that the average co-identification number of proteome and transcriptome was 2/3 of transcriptome identifier number on gene product level^{1, 19, 20}, suggesting that transcriptome has an advantage on genome coverage than proteome. Even so, proteins are the main functional

components in all cells. Previous studies as well as our data indicated the poor correlation between quantities of proteins and transcripts^{16, 17, 19, 21}, emphasizing the importance of post-transcriptional processes that control protein synthesis and degradation.

According to the reviewer's suggestion, we explored the types and functions of 4,269 missing proteins (**Figure CL11**). We found that the expression levels of over 78% of the transcripts with missing proteins were less than the median of the total transcriptome. These low-abundance transcripts may not be expressed, or their expression levels were too low to be detected by mass spectrometry^{19, 22}. Through gene enrichment analysis, we found that the other 20% of missing proteins were not detected due to specific post-translational modification (the genes were annotated by ubiquitination), extracellular localization (membrane or secreted proteins), or biological function (transcription factors or assistors). Consistent with our results, Jiang, L. *et al.* compared the proteome and transcriptome data of 32 human tissues systematically, and showed that many ubiquitous transcripts are found to encode tissue-specific proteins without cognate proteins being detected, and discordance of RNA and protein enrichment provides evidence of protein secretion¹⁹. The results exactly reflected the limitation of proteome. We discussed the coverage limitation of proteome in the discussion (**Please see details in Page 8-9 and Page 21 in the revised manuscript**).

We also analyzed the function of proteins without corresponding mRNA identification (**Figure CL11**). As a result, we found that the protein products of 50% missing transcripts could be derived from specific macrophage phagocytic exosome granules or other apoptotic cells. Due to different half-life of protein and mRNA, immediate mRNA degradation could also contribute to leak detection of mRNA²³. For example, we found that many cytoskeleton gene products (which is more stable than other proteins according to previous reports²⁴) were detected by proteome while missed in transcriptome. Also, the protein products of 30% missing transcripts were identified only once in total or detected separately in different cells, reflecting the transient expression of proteins or noisy signal of mass spectrometry (**Please see details in Page 8-9 and Page 21 in the revised manuscript**). Consistently, previous literatures supported that biological processes, such as mRNA modification, mRNA degradation, protein ubiquitylation and protein degradation, are directly responsible for the inconsistency between transcriptome and proteome coverage²⁵ (**Figure CL12**).

In summary, these results and literatures emphasized that the identification of proteins with no corresponding mRNA could be attribute to the biological factors. Besides, the FACS test demonstrated that the cell purities in our current study were greater than 98%, the results were not significantly influenced by the cell contamination.

Figure CL11. Screening of missing proteins (up) or missing transcripts (down) in our proteome and transcriptome. **(a)** Density distribution for transcripts of 4,269 missing proteins (blue) and protein products of 878 missing mRNA (red) according to their average expression values and identification frequencies across the 12 macrophage populations. The genes were divided into 4 groups based on the cut-off of median value on expression level and 50% of on identification frequency level. **(b)** Representative functional annotations (GOBP and KEGG database, Fisher's exact test) for genes in indicative 4 regions in figure 2d. The dot size represents the number of proteins involved in the relevant terms. The color bar indicates the enrichment significance. **Related to Figure 1f, d, and Supplementary Figure 4g, h in the revised manuscript.**

Figure CL12. Overview of the gene expression pathway²⁵. The processes at different steps in the gene expression pathway that confer regulatory control are indicated at the bottom. miRNA, microRNA.

The analysis results of missed genes in proteome and transcriptome among the 12 macrophages were included in Figure 1 (**page 8-9 in the revised manuscript**) in the revised manuscript. We also discussed the limits of proteomics detection and the reason of the discordance identification in protein and mRNA levels in discussion part of the revised manuscript (**Please see page 21 for details**).

Q7: Lines 197-233 deal with WGCNA of a set of putative PRR pathway genes. The rationale for this analysis is entirely unclear, and the choice of parameters for WGCNA is not justified. My feeling is that the analysis is excessively fine-grained and the inferred association with specific PAMP recognition is not supported by any of the literature (see also FACS data on tissue macs in PMID: 32840578).

Response 7: Thanks for the comment. The PRR pathway plays a central role in immune regulation function of macrophage populations, and that the expression of molecules in PRRs pathways among different macrophages remain unclear. Besides, proteins involved in PRR signaling pathways showed relatively high coefficient variations (CVs) across the 12 macrophages. To systematically present the diverse protein expression patterns of PRR pathway genes, we performed WGCNA analysis in previous manuscript. The parameters for WGCNA were set as follows (in previous manuscript):

A total of 277 PRR-signaling-pathway-related proteins were clustered into functional modules using a weighted gene co-expression network analysis (WGCNA) package²⁶ in RStudio. A soft threshold at power

of 12 was selected based on scale free topology model fit ($R^2 = 0.8$). Module identification was performed with the 'cutreeDynamic' function using the 'tree' method and `minModuleSize = 5` (see methods and materials in previous manuscript).

As for specific PAMP recognition, our data revealed the expression variations of PRR across the different macrophages, and inferred the functional diversity of macrophage responses to PAMP. In consistent with our findings, previous researches have also indicated the wide variation in the expression of PAMP receptors from tissue to tissue (PMID: 18800157, <http://rstats.immgen.org/DataPage>)²⁷, for instance, the Tlr1, Tlr2, Tlr9, Nlrp6, P2x7, and Mefv, which showed elevated expression in intestine in our data were also showed significantly increased expression in the recent published single cell dataset (**Figure CL13**).

Figure CL13. Heatmap of expression patterns of indicated genes across different macrophage populations. Expression values for each gene at all populations are color-coded based on the log10-transformed quantification values in published research, red for high and blue for low.

During the revision, we took reviewer's advice, removed the WGCNA analysis of PRR pathway genes. In our revised manuscript, PRR pathway genes were included in the WGCNA analysis based on the global proteome data. Therefore, we could extract expression properties of PRR pathway genes from the whole proteome WGCNA analysis (**Figure CL14a**) and summarized them into a network diagram (**Figure CL14b**, also see Figure 3d in the revised manuscript). most of PRR-pathway proteins were widely expressed among populations although, their expression levels varied in different macrophages. For instance, Tlr5, Nlrc4, Naips, etc. were significantly identified CTMs of alveolar macrophages; Nod1, Trmps, etc. were highly expressed in Kupffer cells; Tlr1, Tlr12, P2x7, etc. were highlighted in intestinal macrophages. The tightly correlated genes in the network may suggested a functional diversity of macrophage responses to different pathogen-associated molecular patterns (PAMPs). To make the statement more precise, we removed our

statements on the inferred association of PRR expression with PAMP recognition in a certain macrophage population.

Figure CL14. Expression pattern of PRR-signaling-pathway related genes across the 12 macrophage populations. **(a)** Heatmap of expression patterns of PRR-signaling-pathway related genes across the 12 macrophage populations. Expression values for each gene at all populations are color-coded based on the z-scored copy numbers per cell. **(b)** Module network depicting the functionally diverse PRR proteins in different macrophages assigned by WGCNA. Proteins are labeled with the same color and with the indicated types of macrophages. Gene significance (GS) is represented by node size. Edges represent weighted Pearson correlation coefficients shown with gradually varied color. **(c)** Proteomic quantifications of TLRs across the

12 macrophage populations. The expression values are log₁₀-transformed copy numbers per cell. **Related to Figure 2d, e, f, and Supplementary Figure 6b in the revised manuscript.**

As for the TLR expression, we found the ubiquitous expression of TLR2, 3, 4 in tissue-resident macrophages, the relatively low expression of TLR7 in Kupffer cells, and the obviously high expression of TLR4 in Kupffer cells and TLR5 in alveolar macrophage populations, as well as the broad-spectrum expression of TLRs in intestinal macrophage populations (**Figure CL14c**, also see Figure 3e in the revised manuscript). The results also supported by the FACS data mentioned by the reviewer (PMID: 32840578)²⁸.

Q8: Lines 235-259 describe the response of isolated KC to LPS, CpG DNA and flagellin. This is irrelevant and uninterpretable. The cells are incubated for 24 hrs without added CSF1 (which is required for maintenance of protein synthesis and regulates expression of TLR4 and TLR9) during which time their differentiation status changes. The stimuli are added at an arbitrary concentration and incubated for an arbitrary time after previous incubation in serum-free medium for 2 hours.

Response 8: Thanks for the constructive comments. Accordingly, we removed the data and conclusions about this part from manuscript and figures in the revision. As most of the data were presented in supplementary materials in previous manuscript, the revision has no impact on the structure and conclusion of the paper.

Q9: Lines 286-315. Figure 3 shows the identification of protein co-expression modules to identify cell-type specific proteins. As a preliminary analysis, it would be worthwhile to highlight the detection of markers that are known to be cell-type specific in these populations from FACS analysis. This is actually quite evident in the proteome database (e.g., CLEC4F, CD5L, VSIG4 and TIMD4 are all very highly-expressed by KC; P2RY12 in microglia). However, the analysis and an overview of the related Table highlights a major issue that compromises any interpretation. All of the preparations have quite high levels of endothelial markers (PECAM1, VWF) and the high level of albumin contamination varies, which compromises relativity estimates. The microglia-enriched clusters include major myelin-associated proteins (MOBP, PLP1) and neuronal membrane transporters that are absolutely not

expressed by microglia. Similarly, the alveolar macrophages have high levels of surfactant proteins and the gut macrophages have high levels of known intestinal epithelial markers (e.g., VIL1) transporters. This may reflect endocytic activity, but it is not possible to separate this possibility from straightforward contamination and the entire inference of functions for these genes in macrophages are invalid. The mention of RAW264 neglects the fact that it is a different mouse strain and the difference cannot be attributed to the lack of an immune environment.

Response 9: Thanks for the comments. The reviewer mentioned: First, identification of proteins that should not be expressed by macrophage populations (including PECAM1, VWF, MOBP, PLP1, VIL1, etc.), and possibility of cell contamination; second, neglection of mouse strain influence on interpretation of RAW264.7 functions. Here, we answer these questions separately.

About identification of proteins that presumably should not be expressed by macrophages

To answer the question, we searched the transcriptome data-matrix of published meta-analysis⁷. In the published transcriptome dataset, as shown in **Figure CL15**, Firstly, both of PECAM1 and VWF were widely detected in macrophage populations, and prominently identified in spleen red pulp macrophages; Secondly, myelin-associated protein (like MOAP and PLP1) were detected in microglia with relatively high levels; Thirdly, VIL1 were also significantly identified in gut macrophage populations (including large and small intestinal macrophages); Fourthly, albumin (Alb) was obviously identified in liver macrophage populations. Besides, previous studies also showed the expression of albumin in macrophage populations²⁹. The result suggested the reliability of our proteome data. Regarding the cell contamination suspected by the reviewer, FACS test showed that the purity of macrophage populations in our study was greater than 98%, as mentioned above. Thus, the result was not possibly affected by cell contamination.

About interpretation of RAW264.7 functions

We recognized that we neglected the different mouse strain explanation for function feature identification of RAW264.7, as mentioned by the reviewer. Accordingly, we deleted the interpretation of 'lack of an immune environment' for the result about RAW264.7 function, and discussed the shortage of using RAW264.7 for comparison (**See Page 20 in the revised manuscript**), Considering that no mouse macrophage cell lines

derived from C57BL/6N are available for comparison with primary macrophage populations nowadays, we kept the proteome data of RAW264.7 in our study to provide a resource as a reference for comparison between immortal cell line and primary populations.

Figure CL15. Bar plots show the average quantification values of indicated gene transcripts across different macrophage populations in published meta-data analysis (PLoS Biol, 2020, PMID: 33031383)⁷

Q10: Lines 319-398 deal with transcription factors detected and inferred TF networks. This analysis is also difficult to interpret because of contamination and activation during isolation. Many of the conclusions reiterate what is known for specific transcription factors (which could be reviewed more adequately) but it is also of note what is missing from tissue-specific data (e.g Spic, Id3, Runx1, Irf8, Tfec, Mafb) and what is likely artefact (e.g Elf3 is massively-expressed in intestine, and entirely restricted to epithelial cells, see proteatlas.org and biogps.org)

Response 10: Thanks for the comment. The reviewer questioned the quantification patterns of specific transcription factors (TFs, including Spic, Id3, Runx1, Irf8, Tfec, Mafb, etc). In our study, a total of 510 TFs were successfully identified and quantified. Based on the deep coverage of TFs in proteome data, we screened out the cell-type specific TFs and cell-type-maintenance TFs (ctmTFs) respectively. The cell-type specific

TFs were determined based on the criterion that the expression level of a TF in a certain cell type was found to be 5 times greater than the geometric median of the expression levels in the other cell types, and belonged to the CTMs with the cut-off of gene significance (GS) ≥ 0.6 and module membership (MM) ≥ 0.5 . We reasoned that ctmTFs should not only be specifically enriched in the macrophage populations but also predominantly control the transcription of their downstream genes in the macrophages. We screened out ctmTFs according to following criteria (see materials and methods in the manuscript): (1) The TF was cell-type specific TFs in one macrophage population. (2) The average z-score of the expression of its target genes (TGs) was greater than the max value of randomly selected proteins from the proteome data for 1000 times.

Most of the transcriptional factors listed by the reviewer were obviously identified, and captured by relevant cell-type-specific TFs set and CTMs (cell-type-specific modules, with the cut-off of GS ≥ 0.6 and MM ≥ 0.5), rather than by ctmTFs sets. For example, Spic was captured in spleen-red-pulp CTMs, and also high expressed in the liver macrophage populations. Id3 was grouped into the CTMs of Kupffer cells, and also highly expressed in gut and peritoneal macrophage populations. Runx1 was included in CTMs of alveolar macrophages, Irf8 and Mafb were included in the CTMs of and microglia, relatively (**Figure CL16, Supplementary Figure 4a**). The result was consistent with previous studies, including published transcriptome datasets (**Figure CL16**)⁷, suggesting the high purity of the macrophages used in our study. **According to the comment, we carefully summarized the cell-type specific TFs with a heatmap (Supplementary figure 7a) and in Supplementary table 6.** Tfec was not identified by our proteome may due to the limitation of mass spectrum on TF identification. We discussed the detection limitation in the discussion in the revision (**Please see page 21 in the revised manuscript for detail**).

Figure CL16. Heatmap of quantification values of indicated TFs across different macrophage populations in our proteome (a) and published transcriptome datasets (in average) (b, PLoS Biol, 2020, PMID:

33031383)⁷. Expression values for each gene at all populations are color-coded based on the z-scored quantification values per cell.

As for the expression specificity of Elf3 shown in proteatlas.org or biogps.org, the data were derived from transcriptome studies that did not contain the case of macrophage populations in C57BL/6N mice. Taken proteatlas.org as an example, the bulk RNAseq data from Genotype-Tissue Expression (GTEx) project was used to determine the cell type specificity of all protein coding genes. The GTEx datasets were collected from multiple human post mortem tissues, but not the protein expression indicators from mice (<https://www.proteatlas.org/about/assays+annotation#transcriptomics>). Furthermore, in the published transcriptome concerning macrophage populations in mice (in meta-analysis)⁷, Elf3 was also obviously detected in intestinal macrophages (**Figure CL17**).

Figure CL17. Average quantification values of Elf3 transcripts across different macrophage populations in published meta-data analysis (left, *PLoS Biol*, 2020, PMID: 33031383)⁷ and our proteome datasets (right).

To assess the purity of macrophage populations, we evaluated the purity of macrophage populations through flow cytometer (FCM) with the same gating strategy. The result showed that purities of all 11 populations were greater than 98% (**Figure CL8**), indicating that the cell populations were not contaminated significantly. We summarized the gating strategies and purity levels of macrophage populations **Table CL1**. (**Please see Supplementary Table 14 in the revised manuscript for detail**).

Table CL1. Gating panels and purity for all macrophages. **Related to Supplementary Table 14 in the revised manuscript.**

Tissue	cell type	Markers for identification/purification	Purity
Brain	Microglia	CD45 ⁺ F4/80 ⁺ CD11b ⁺ Cx3cr1 ^{hi} MHCII ^{lo} Ly6g ⁻ CD117 ⁻ CD24 ⁻ B220 ⁻ Ly6c ^{lo}	≥98%
	Lung-resident MΦ	CD45 ⁺ F4/80 ^{hi} CD11b ^{lo} SiglecF ^{hi} CD11c ^{hi} Ly6g ⁻ CD117 ⁻ CD24 ⁻ B220 ⁻ Ly6c ^{lo}	≥98%
Lung	Lung-recruited MΦ	CD45 ⁺ F4/80 ^{lo} CD11b ^{hi} Ly6g ⁻ CD117 ⁻ CD24 ⁻ B220 ⁻ Ly6c ^{hi/lo}	≥98%
	Kupffer cells	CD45 ⁺ F4/80 ^{hi} CD11b ^{lo} MHCII ⁺ Ly6g ⁻ CD117 ⁻ CD24 ⁻ B220 ⁻ Ly6c ^{lo}	≥98%
Liver	Liver-recruited MΦ	CD45 ⁺ F4/80 ^{lo} CD11b ^{hi} Ly6g ⁻ CD117 ⁻ CD24 ⁻ B220 ⁻ Ly6c ^{hi/lo}	≥98%
	Spleen-resident MΦ	CD45 ⁺ F4/80 ^{hi} CD11b ^{lo} MHCII ⁺ Ly6g ⁻ CD117 ⁻ CD24 ⁻ B220 ⁻ Ly6c ^{lo}	≥98%
Spleen	Liver-recruited MΦ	CD45 ⁺ F4/80 ^{lo} CD11b ^{hi} Ly6g ⁻ CD117 ⁻ CD24 ⁻ B220 ⁻ Ly6c ^{hi/lo}	≥98%
SI	Small intestinal MΦ	CD45 ⁺ F4/80 ⁺ CD11b ⁺ MHCII ⁺ CD11c ⁺ Ly6g ⁻ CD117 ⁻ CD24 ⁻ B220 ⁻ Ly6c ^{lo}	≥98%
LI	Large intestinal MΦ	CD45 ⁺ F4/80 ⁺ CD11b ⁺ MHCII ⁺ CD11c ⁺ Ly6g ⁻ CD117 ⁻ CD24 ⁻ B220 ⁻ Ly6c ^{lo}	≥98%
Peritoneal	Peritoneal MΦ	CD45 ⁺ F4/80 ⁺ CD11b ⁺ MHCII ^{lo} CD115 ⁺ Ly6g ⁻ CD117 ⁻ CD24 ⁻ B220 ⁻ Ly6c ^{lo}	≥98%
Bone	BMDMs	CD45 ⁺ F4/80 ⁺ CD11b ⁺ CD64 ⁺ Ly6g ⁻ CD117 ⁻	≥98%

Q11: Lines 400-446 deal with the difference between recently recruited and tissue resident macrophages and subsequently with apparent differences in their response to LPS in vitro. This comparison is based upon an arbitrary assignment of F4/80 and CD11b bright and dim as markers, which I personally feel is strongly debatable (e.g. F4/80 is absent from marginal zone and lymphoid follicle populations in spleen, and low on AM in lung). The progressive differentiation of monocyte-derived cells in multiple organs has been analyzed quite extensively by others, and I was not compelled by any new insights from this analysis. In figure 7A, Clec5a is certainly known to be monocyte-enriched,

but MUC1 mRNA is barely detectable in any published isolated macrophage population, so this is very likely due to contamination.

Response 11: Thanks for the comments. The reviewer mentioned three questions: First, the feasibility of F4/80 and CD11b bright or dim as markers to distinguish tissue-resident and recruited macrophage populations; Second, the novelty of this part of research; Third, the protein expression patterns of Clec5a and Muc1 in macrophage populations.

About the use of F4/80 and CD11b bright or dim as markers to distinguish tissue-resident and tissue-recruited populations

To answer this question, we reviewed the literatures of the last decade to understand the development of markers for macrophage sorting. The concept of macrophage subtype identified with F4/80^{hi}CD11b^{lo} or F4/80^{lo}CD11b^{hi} has been widely used in the published literatures. For example, (1) Schulz *et al.* showed that the monocyte-derived-macrophage subpopulation is featured with CD11b^{hi}F4/80^{lo} macrophages, and another is originated from yolk sac, identified as the CD11b^{lo}F4/80^{hi} macrophages (**Figure CL18 and 19**, Science, 2012, PMID: 22442384)⁴; (2) Zanganeh, S. *et al.* revealed increased quantities of liver CD11b^{hi}F4/80^{lo} macrophages in ferumoxylol-pretreated mice compared to untreated controls through flow cytometry analysis (**Figure CL20**, Nat Nanotechnol, 2016, PMID: 27668795)³⁰; (3) Zhao, Y. *et al.* using the same gating strategy (F4/80 and CD11b high or low) to detect tissue-resident and monocyte-derived macrophages, and found the short-term mTOR deficiency after birth decreases CD11b^{hi}F4/80^{lo} macrophage development from the hemopoietic stem cells (HSCs), but had no detectable effects on the yolk sac-derived CD11b^{lo}F4/80^{hi} macrophages (**Figure CL21**, Blood, 2018, PMID: 29463562)³¹; (4) Pia Rantakari *et al.* compared the proportion of tissue-resident macrophages gated by F4/80^{hi}CD11b^{lo} and monocyte-derived macrophages gated by F4/80^{lo}CD11b^{hi} in multiple organs of wild-type and *Cav1*^{-/-} mice, in order to investigate the role of endothelium in tissue-resident macrophage seeding (**Figure CL22**, Nature, 2016, PMID: 27732581)³²; (5) So Yeon Kim *et al.* isolated F4/80^{hi}CD11b^{lo} Kupffer cells and F4/80^{lo}CD11b^{hi} monocyte-derived macrophages in the liver, and evaluated the expression of 7 pro-inflammatory genes in two macrophage populations (**Figure CL23**, Nature Communications, 2017, PMID: 29269727)³³.

Figure CL18. Gating strategy of primary tissue macrophages. CD45⁺ cells were gated from the P2 gate and macrophage populations were identified by F4/80 and CD11b expression (F4/80^{hi}CD11b^{lo} or F4/80^{lo}CD11b^{hi}). Related to Supplementary Figure 6C in the published study (Science, 2012, PMID: 22442384)⁴.

Figure CL19. Flow cytometric analysis of the indicated organs at E14.5 and E16.5 of WT (n = 11 and 6, respectively), *Myb*^{+/-} (n = 8 and 15), *Myb*^{-/-} (n=4and4), and *Pu.1*^{-/-} (n=3to5) mice, gated on CD45⁺ cells. F4/80^{bright} CD11b^{low} macrophages are color-gated in blue in dot plots and represented by white bars in histograms. F4/80^{low} CD11b^{high} macrophages are color-gated in red and represented by black bars in histograms. **P* < 0.05 of *Myb*^{-/-} versus WT. Related to the Figure 2A and Figure 2B in the published study (Science, 2012, PMID: 22442384)⁴.

Figure CL20. Livers of the indicated mice were further analyzed with FACS for infiltrating leukocyte populations. Related to the Figure 2A and Figure 2B in the published study (Nat Nanotechnol, 2016, PMID: 27668795)³⁰.

Figure CL21. Cells of spleen, peritoneal cavity, and colon from tamoxifen-treated WT and ER-mTOR KO mice were stained with anti-CD11b mAb and anti-F4/80 mAb and analyzed by flow cytometry. Related to the Figure 1A in the published study (Blood, 2018, PMID: 29463562)³¹.

Figure CL22. Flow cytometry analyses of yolk-sac-derived ($CD11b+F4/80^{high}$; red gates) and fetal liver-derived ($CD11b+F4/80^{intermediate}$; blue gates) macrophages. Related to the Supplementary Figure 3e in the published study (Nature, 2016, PMID: 27732581)³².

Figure CL23. Isolated cells were visualized by Giemsa staining. Bar = 10 μ m. Related to the Figure 3b in the published study (Nature Communications, 2017, PMID: 29269727)³³.

During literature consulting, we also noted that F4/80 is absent from marginal zone and lymphoid follicle populations in spleen as described by the reviewer. In this study, however, we analyzed spleen red pulp macrophages, in which F4/80 was positively expressed. Another detail is the low expression of F4/80 in AMs as mentioned by the reviewer. F4/80 is lower in AMs than in other tissue macrophages, even though, in the cell suspension of the lung, AMs express relatively higher levels of F4/80 than other cell types in the lung, and the molecular phenomenon of F4/80^{hi} was frequently used for AM identification^{34,35}. Meanwhile, other markers, including Siglec^f and CD11c, were also employed to identify/sort and purity validation alveolar macrophages in our study (Supplementary Figure 1).

About the innovation of this part of the research

As the reviewer mentioned, the differentiation of monocyte-derived cells in multiple organs has been analyzed quite extensively. However, the main point of this study is to explore the global proteome diversity of tissue-resident and recruited macrophages, rather than the mechanism (nor variation of gene expression pattern) of monocyte differentiation. The analysis revealed the functional features and relevant molecular signatures between tissue-resident and recruited macrophages in immune- and non-immune-related tissue

supporting activities, especially in the liver and lung. The result highlighted: (1) the preferential involvement of tissue-resident macrophages in tissue functional regulation, (2) the greater chemotaxis and adhesion capabilities and a lesser PAMP recognition capability of tissue-recruited macrophages, and (3) the critical role of Il18 in distinguishing feature identities of the two macrophage populations. Above all, our proteome resource provides the direct molecular references for the functional diversity between the tissue-resident and recruited macrophage populations. To make our interpretation more precise, we revised the language of this part and highlighted the significance of the research.

About expression of Clec5a and Muc1

Thanks for the approval of Clec5a expression. The reviewer also mentioned the expression of Muc1. Just as reviewer described, Muc1 was barely detected on mRNA level, while was obviously identified in alveolar macrophage on protein level. Consistently, a published study³⁶ showed that MUC1 was modestly expressed in normal mouse lung macrophages. ERK activity-mediated protein stabilization was required for MUC1 expression, but had little effect on MUC1 transcripts (**Figure CL24**). The result means that the posttranscriptional modification (PTM) and instability of MUC1 transcripts may influence the quantification values of Muc1 gene in proteome and transcriptome.

Figure CL24. Expression of MUC1 in macrophage. **(a)** THP-1 cells were treated with CSE (cigarette smoke extract, 20 mg/mL TPM). MUC1 mRNA level was detected by RT-PCR. β-Actin was detected as an input control. Related to Figure 3a in the published study. **(b)** U937 cells were treated by CSE (40 mg/mL TPM) with or without U0126 (inhibitor of ERK, 10 mmol/L). MUC1 expression was detected by Western blot analysis at the indicated time points. β-Actin was detected as an input control. The intensity of the individual bands was quantified by densitometry (ImageJ) and normalized to the corresponding input control bands.

MUC1 expression changes were calculated with the control taken as 100%. Related to Figure 3a and Figure 4c in the published study (Cancer Research, 2014, PMID: 24282280) ³⁶

As for the cell contamination concerned by the reviewer, FACS experiments showed that the cell purity in this study was greater than 98% without affecting on the protein quantification result, as mentioned above.

Q12: The final section in Figure 8 focusses on the ability of tissue-resident macrophages to produce IL18. The recently published meta-analysis of macrophage gene expression profiles indicated that all tissue resident macrophages in C57BL/6 mice express IL18 mRNA constitutively. The final section of this paper deals with the impact of an IL18 knockout; these are inadequately described in the methods, which require a reference and details of the genetic background and the use of littermate controls. However, a cursory PubMed search on IL18 and knockout reveals >1000 hits, many dealing with inflammasome activation and the lung, so the originality of this work is unclear and needs to be placed in some kind of context.

Response 12: Thanks for the comment. According to the suggestion, we described detailed information of the construction process (with reference), genetic background, and the use of littermates of IL18 knockout mice in the ‘Materials and Methods’ part. **Please see Page 24 in the revised manuscript for details.**

The reviewer also queried the originality of this part. Indeed, many published studies focused on immunoregulatory role of IL18 in the macrophages of the lung and liver. Nevertheless, the augment of monocyte recruitment capability of liver Kupffer cells in IL18^{-/-} mice has not been reported, and the variations of molecular expression patterns remain unknown. In our study, we proposed for the first time that the enhanced ability of monocyte recruitment by Kupffer cells may serve as a mechanism for inflammatory outburst in the liver of IL18^{-/-} mice, under acute inflammatory conditions. The LC-MS/MS data introduced in this part provide a resource to investigate the difference of secretome/proteome pattern between IL18^{-/-} mice and the littermate mice *in vitro*.

DISCUSSION

Q13: The discussion is somewhat rambling and unrelated to the results. The M1/M2 paradigm was never useful and most M2 markers are known to be expressed by resident tissue macrophages. The discussion of tumor-associated macrophages concludes that the diverse functions of tissue-resident macrophages and recruited macrophages in carcinogenesis deserve further investigation. There is already a massive literature on this topic!!

Response 13: Thanks for the constructive comments. According to the comment, we rewrote the discussion part of the manuscript in the revision. **Firstly**, we deleted the Figures and discussion text about M1/M2 polarization and tumor-associated macrophage (TAMs), as well as the description such as ‘the diverse functions of tissue-resident macrophages and recruited macrophages in carcinogenesis deserve further investigation’. **Secondly**, we added discussion about the novelty and advantages of the study. **Thirdly**, we discussed the identification bias between proteome and transcriptome, and highlighted the short-coming of the study, especially impacts derived from cell activation and phagocytosis. **Please see Page 20-22 in the revised manuscript for details.**

Q14: I had a look at the on-line resource, which is said to “provide a valuable resource for carrying out novel investigations of function and mechanism studies of macrophages”. The resource is quite easy to use and potentially useful as a resource but could use a brief guide. It would be useful to be able to see the proteomic and transcriptomic data for a particular on the same view and to ensure the search recognizes synonyms and protein names (e.g. Adgre1, F4/80, Emr1).

Response 14: Thanks for the constructive comments. Accordingly, we added a brief guide in the home page (<http://macrophage.mouseprotein.cn>), and provided a module with parallel exhibition of proteomic and transcriptomic data in a same view in the revision (**Figure CL25**). Now the on-line resource support direct search using synonyms or protein names (**Figure CL26**).

Figure CL25. On-line resource showing the quantification values of Adgre1 across 12 macrophage populations on protein and mRNA levels within a same view.

Gene Symbol

Macrophages

- Un-select All
- Microglia
- Alveolar MΦ
- Lung recruited MΦ
- Kupffer cells
- Liver recruited MΦ
- Red pulp MΦ
- Spleen recruited MΦ
- Small intestinal MΦ
- Large intestinal MΦ
- Peritoneal MΦ
- BMDM
- RAW264.7

Data

- Un-select All
- Proteome
- Transcriptome

Figure CL26. Alternative options of Searching box in our on-line proteome resource (<http://macrophage.mouseprotein.cn>). Searching directedly against synonyms or protein names were supported in our resource.

Reviewer #4 (Remarks to the Author):

In this manuscript, author carried out experiments to profile the proteome and transcriptome patterns of 11 macrophage populations from eight tissues and the cell line RAW264.7, with 12,316 proteins and 16,413 gene transcripts identified. They subsequently analyzed the proteome and revealed some interesting features. The resulting data thus will provide a valuable resource for studying functions of macrophages. This is interesting and comprehensive study. Nevertheless, the authors should address the concerns below before this paper can be published.

Q1: The rational for selecting primary macrophage from 8 tissues is not clear. Why not heart? The macrophage is critically important for fat function. The authors should carry out the proteomic analysis of the two types of macrophages and include the data into the paper.

Response 1: Many thanks for the positive and constructive comments. As the reviewer mentioned, besides the shared typical macrophage functions (such as phagocytosis and efferocytosis), the cardiac and adipose macrophages also play important homeostatic roles. We tried our best to get the sufficient number of cardiac/adipose macrophages with high purity from mice, following the protocols described in published researches^{37, 38}. However, based on the marker combination of CD45⁺F4/80⁺CD11b⁺(**Figure CL27a**), only about 3E4 and 4E4 primary macrophages were obtained respectively from the heart and inguinal adipose tissue of each normal C57BL/6N mouse (8-weeks to 10-weeks), consistent with the published literatures. The cell count is an order of magnitude less than the number of macrophages in other tissues (**Table CL2**) and falls far short of the minimum requirements for proteomic analysis (1.5E6-2E6 for each experiment). Furthermore, the purities of macrophage populations sorted from the heart and adipose were around 92%, much lower than the purity of macrophages (98%) obtained from other tissues (**Figure CL27b**). Such the low cell counts and purities of heart/adipose macrophages would seriously affect the credibility of quantitative proteomics. Therefore, we have not yet been able to achieve high-quality proteomes of heart and fat macrophage. In view of the important role of macrophages in the heart and fat, we will profile the proteome of the two macrophage populations in the future studies, and update the data in our data portal. **We**

discussed the limitation in our revised manuscript, please see page 22 and Supplementary Figure 11 and Supplementary Table 15 in the revised manuscript for details.

Figure CL27. Representative flow cytometric analysis of the single cell suspension of the heart (up) and inguinal adipose tissues (down). (a) Representative scatter of gating strategies for macrophage populations in indicated tissues. (b) Representative scatter of post-sort FCM analysis of macrophage populations in indicated tissues. **Related to Supplementary Figure 11 in the revised manuscript.**

Table CL2. Cell number and purity of post-sort macrophage populations in different tissues (per mouse).

Related to Supplementary Table 15 in the revised manuscript.

Tissue	Number of MΦ/mouse	Purity
Brain	1e5-2e5	98%
Lung	4e5-5e5	98%
Liver	8e5-1e6	98%
Spleen	3e5-5e5	98%
Small intestine	3e5-3.5e5	98%
Large intestine	2.5e5-3e5	98%
Peritoneal dropsy	2e6-3e6	98%
Heart	3e4-4e4	92%
Inguinal adipose	1.7e4-3e4	92%

Q2: As authors suggested, proteomics studies were carried out previously in macrophage derived from liver, brain and heart. What are the major differences between this study and previous ones?

Innovation in terms of methodology and biology?

Response 2: Thanks for the comments. In our previous manuscript, we mentioned three projects about cell type-resolved proteomes as ‘previous studies on cell-type-resolved liver, brain, and heart proteomes^{16, 17, 21, 39} have emphasized the poor correlation between mRNA and protein profiling, revealing the importance of post-transcriptional procedures that regulate protein synthesis and degradation’. In above projects, the authors performed proteome studies on macrophages in the liver (Kupffer cells) and brain (microglia). As for the heart, all of CD45⁺ immune cells, rather than macrophage populations, were subjected to proteomic analysis.

Compared with previous studies, **firstly**, this study possesses the innovation in biology. Although the previous researches performed proteome analysis of liver Kupffer cell^{16, 18} or brain microglia¹⁷, these studies focused on the function diversities or interactions between different cell types within one tissue. The main focus of our current study is on the proteome and function heterogeneity of macrophage populations from different tissues. Besides, our study also systematically explored the functional differences and molecular mechanisms of tissue-resident and recruited-macrophages, which were not covered by published literatures. **Secondly**, in terms of methods, our current study employed LC-MS/MS technology with six fractions detection in each experiment, supporting an extensive coverage of protein identification of macrophage proteomics. **Finally**, in terms of dataset itself, the omics data from different studies is of obvious batch effect, which makes parallel comparison difficult. Our study provides a comprehensive proteome database with the in-depth coverage of protein groups and cell types so far without batch effect between experiments, serving as a valuable resource for future studies of macrophage biology.

Q3: This author does not believe the claim that the quantified protein copy numbers spanned over nine orders of magnitude. This conclusion therefore needs to be validated by an orthogonal method.

Response 3: Thanks for the comment. Western Blot (WB) or Elisa may serve as ‘orthogonal method’ to validate the protein expression in sample. However, (1) both WB and Elisa are experiment based on specific antibodies, the titers of antibodies of different proteins are different, so that the dynamic range of the whole

proteome cannot be determined according to the strip intensities.; (2) the number of proteins which can be quantified is limited, and the protein dynamic range over the proteome scale cannot be successfully archived. In order to validate the dynamic range in our proteome data, BMDM and cell line RAW264.7 were collected and subjected to protein absolute quantification (for twice respectively) with QconCAT-based isotope approach^{40, 41}. In this approach, a synthetic gene encoding an artificial concatenation of tryptic peptides from a set of different proteins (a QconCAT protein) is cloned into an E. coli expression vector and expressed in a medium containing stable heavy isotope labeled lysine and arginine. A known amount (absolute quantification, ABQ) of the QconCAT protein is then co-digested with the analyte proteins, and the ratio of the digested proteins is determined by comparing the relative intensities of the analyte peptides that have been normalized to their corresponding isotope-labeled QconCAT peptides (**Please see Page 24 in the revised manuscript for details**). Because all the steps after protein extraction are performed simultaneously for both analytes and the QconCAT reference protein, the QconCAT method can minimize experimental variations. **Furthermore, one of the inherent advantages of QconCAT is that the molar ratio between each peptide can be set to 1:1 by design and used to normalize different mass spectrometric responses of peptides, allowing relative quantification of different proteins.** With the modified QconCAT method, we successfully obtained an equimolar ratio (1:1:1:1) for the core components of SMC1, SMC3, RAD21, and SA1/2 of the human endogenous cohesion complex in our previous study⁴¹, providing direct analytical evidence to support the ‘ring’ model for cohesion.

We carefully analyzed the QconCAT-based quantitative proteome data of the BMDM and RAW264.7. The label efficiency of heavy isotope-QconCAT standard is more than 99%. The average adjusted r^2 of absolute quantification (ABQ) values and intensity-based absolute quantification (iBAQ) values of QconCAT was 0.9 (**Figure CL28a**), representing the high confidence of the linear relationship between iBAQ and ABQ of the proteome. Based on the linear relationship, we calculated ABQ values and copy number values of proteins according to iBAQ values. **As shown in Figure CL28b-c, the quantified protein copy numbers spanned over nine orders of magnitude, consistent with the conclusion in our manuscript.** The data was included in the manuscript as a validation for proteome dynamic range. **Please see Line 150-153 and Supplementary Figure 3 in the revised manuscript for details.**

Figure CL28. Dynamic range of quantification values in proteome datasets of RAW264.7 and BMDM. **(a)** Linear regression relationship between iBAQ values and ABQ of QconCAT in proteome datasets of RAW264.7 and BMDM performed by QconCAT-based isotope labeling quantification. **(b, c)** Dynamic range of copy number values in proteome datasets of RAW264.7 and BMDM performed by label-free-quantification (in manuscript, b) and QconCAT-based isotope labeling quantification (newly archived, c). **Related to Supplementary Figure 3 in the revised manuscript for details.**

Q4: Too many papers were cited. Suggest to reduce reference number.

Response 4: Thanks for the constructive comments. During the revision process, we streamlined and updated the reference to make it more accurate and progressive. The current reference number is **64**.

Q5: The paper should be carefully edited by a native English speaker.

Response 5: We have carefully checked the manuscript to correct grammatical and spelling errors. The revised manuscript was sent to immunologists and professional science writers for their suggestions and editing. All changes are highlighted **in blue** in the revised manuscript.

References:

1. Xu, J.Y. *et al.* Integrative Proteomic Characterization of Human Lung Adenocarcinoma. *Cell* **182**, 245-261 e217 (2020).
2. Wang, L.B. *et al.* Proteogenomic and metabolomic characterization of human glioblastoma. *Cancer Cell* (2021).
3. Elias, J.E. & Gygi, S.P. Target-decoy search strategy for increased confidence in large-scale protein identifications by mass spectrometry. *Nature methods* **4**, 207-214 (2007).
4. Schulz, C. *et al.* A lineage of myeloid cells independent of Myb and hematopoietic stem cells. *Science* **336**, 86-90 (2012).
5. Schridde, A. *et al.* Tissue-specific differentiation of colonic macrophages requires TGFbeta receptor-mediated signaling. *Mucosal Immunol* **10**, 1387-1399 (2017).
6. Waddell, L.A. *et al.* ADGRE1 (EMR1, F4/80) Is a Rapidly-Evolving Gene Expressed in Mammalian Monocyte-Macrophages. *Front Immunol* **9**, 2246 (2018).
7. Summers, K.M., Bush, S.J. & Hume, D.A. Network analysis of transcriptomic diversity amongst resident tissue macrophages and dendritic cells in the mouse mononuclear phagocyte system. *PLoS Biol* **18**, e3000859 (2020).
8. Cai, B. *et al.* Macrophage MerTK Promotes Liver Fibrosis in Nonalcoholic Steatohepatitis. *Cell metabolism* **31**, 406-421 e407 (2020).
9. Ambade, A. *et al.* Pharmacological Inhibition of CCR2/5 Signaling Prevents and Reverses Alcohol-Induced Liver Damage, Steatosis, and Inflammation in Mice. *Hepatology (Baltimore, Md.)* **69**, 1105-1121 (2019).
10. Li, Z. *et al.* Adult Connective Tissue-Resident Mast Cells Originate from Late Erythro-Myeloid Progenitors. *Immunity* **49**, 640-653 e645 (2018).
11. Huang, Q.Q. *et al.* Critical role of synovial tissue-resident macrophage niche in joint homeostasis and suppression of chronic inflammation. *Sci Adv* **7** (2021).
12. Park, J.G. *et al.* Immune cell composition in normal human kidneys. *Sci Rep* **10**, 15678 (2020).
13. Ito, T. *et al.* Cell barrier function of resident peritoneal macrophages in post-operative adhesions. *Nat Commun* **12**, 2232 (2021).
14. Howard, Z.M. *et al.* Early Inflammation in Muscular Dystrophy Differs between Limb and Respiratory Muscles and Increases with Dystrophic Severity. *The American journal of pathology* **191**, 730-747 (2021).
15. Gosselin, D. *et al.* Environment drives selection and function of enhancers controlling tissue-specific macrophage identities. *Cell* **159**, 1327-1340 (2014).
16. Azimifar, S.B., Nagaraj, N., Cox, J. & Mann, M. Cell-type-resolved quantitative proteomics of murine liver. *Cell metabolism* **20**, 1076-1087 (2014).
17. Sharma, K. *et al.* Cell type- and brain region-resolved mouse brain proteome. *Nat Neurosci* **18**, 1819-1831 (2015).
18. Ding, C. *et al.* A Cell-type-resolved Liver Proteome. *Mol Cell Proteomics* **15**, 3190-3202 (2016).
19. Jiang, L. *et al.* A Quantitative Proteome Map of the Human Body. *Cell* (2020).
20. Gillette, M.A. *et al.* Proteogenomic Characterization Reveals Therapeutic Vulnerabilities in Lung Adenocarcinoma. *Cell* **182**, 200-225 e235 (2020).
21. Doll, S. *et al.* Region and cell-type resolved quantitative proteomic map of the human heart. *Nat Commun* **8**, 1469 (2017).
22. Nagaraj, N. *et al.* Deep proteome and transcriptome mapping of a human cancer cell line. *Mol Syst Biol* **7**, 548 (2011).
23. Baum, K., Schuchhardt, J., Wolf, J. & Busse, D. Of Gene Expression and Cell Division Time: A

- Mathematical Framework for Advanced Differential Gene Expression and Data Analysis. *Cell Syst* **9**, 569-579 e567 (2019).
24. Schroeter, C.B. *et al.* Protein half-life determines expression of proteostatic networks in podocyte differentiation. *FASEB J* **32**, 4696-4713 (2018).
 25. Buccitelli, C. & Selbach, M. mRNAs, proteins and the emerging principles of gene expression control. *Nat Rev Genet* **21**, 630-644 (2020).
 26. Langfelder, P. & Horvath, S. WGCNA: an R package for weighted correlation network analysis. *BMC Bioinformatics* **9**, 559 (2008).
 27. Heng, T.S. & Painter, M.W. The Immunological Genome Project: networks of gene expression in immune cells. *Nat Immunol* **9**, 1091-1094 (2008).
 28. Sato, R. *et al.* The impact of cell maturation and tissue microenvironments on the expression of endosomal Toll-like receptors in monocytes and macrophages. *Int Immunol* **32**, 785-798 (2020).
 29. Son, M. *et al.* Advanced glycation end-product (AGE)-albumin from activated macrophage is critical in human mesenchymal stem cells survival and post-ischemic reperfusion injury. *Sci Rep* **7**, 11593 (2017).
 30. Zanganeh, S. *et al.* Iron oxide nanoparticles inhibit tumour growth by inducing pro-inflammatory macrophage polarization in tumour tissues. *Nat Nanotechnol* **11**, 986-994 (2016).
 31. Zhao, Y. *et al.* mTOR masters monocyte development in bone marrow by decreasing the inhibition of STAT5 on IRF8. *Blood* **131**, 1587-1599 (2018).
 32. Rantakari, P. *et al.* Fetal liver endothelium regulates the seeding of tissue-resident macrophages. *Nature* **538**, 392-396 (2016).
 33. Kim, S.Y. *et al.* Pro-inflammatory hepatic macrophages generate ROS through NADPH oxidase 2 via endocytosis of monomeric TLR4-MD2 complex. *Nat Commun* **8**, 2247 (2017).
 34. Lavin, Y. *et al.* Tissue-resident macrophage enhancer landscapes are shaped by the local microenvironment. *Cell* **159**, 1312-1326 (2014).
 35. Sajti, E. *et al.* Transcriptomic and epigenetic mechanisms underlying myeloid diversity in the lung. *Nat Immunol* **21**, 221-231 (2020).
 36. Xu, X. *et al.* MUC1 in macrophage: contributions to cigarette smoke-induced lung cancer. *Cancer Res* **74**, 460-470 (2014).
 37. Xia, R. *et al.* Isolation and Culture of Resident Cardiac Macrophages from the Murine Sinoatrial and Atrioventricular Node. *Journal of visualized experiments : JoVE* (2021).
 38. Orr, J.S., Kennedy, A.J. & Hasty, A.H. Isolation of adipose tissue immune cells. *Journal of visualized experiments : JoVE*, e50707 (2013).
 39. Vogel, C. & Marcotte, E.M. Insights into the regulation of protein abundance from proteomic and transcriptomic analyses. *Nat Rev Genet* **13**, 227-232 (2012).
 40. Beynon, R.J., Doherty, M.K., Pratt, J.M. & Gaskell, S.J. Multiplexed absolute quantification in proteomics using artificial QCAT proteins of concatenated signature peptides. *Nature methods* **2**, 587-589 (2005).
 41. Ding, C. *et al.* Quantitative analysis of cohesin complex stoichiometry and SMC3 modification-dependent protein interactions. *J Proteome Res* **10**, 3652-3659 (2011).

REVIEWER COMMENTS

Reviewer #3 (Remarks to the Author):

The authors have responded comprehensively to my comments and have greatly improved the manuscript and the utility of the resource. However, there is one remaining area where I disagree with their response.

In the discussion the authors acknowledge the possibility of contamination.

"Another factor affecting protein or transcript quantification is cell contamination. Meta-analysis performed by Summer et al. indicated that the published macrophage transcriptomic datasets were influenced by extensive contamination of isolated preparations with other cell types straightforwardly or co-purification of cell types that may interact with MPS cells in vivo²² ."

(note, name should be Summers.)

However, the authors have not really appreciated the nature of contamination and they have not eliminated it. None of the genes shown in Figure CL15 is actually expressed by macrophages. To take just one example, Villin is clearly epithelial cell restricted in the gut and villin- reporter and villin-cre mice have been generated and very widely-used (e.g. PMID: 12065599). Notably, villin-cre did not delete fl/fl Csf1r in the intestine (PMID: 29593242)

Obviously, endocytosis of apoptotic cells can provide one source of contamination and is probably important in the gut. But a recent paper by Millard et al. (PMID: 34818538) builds on earlier studies (Lynch PMID: 29607532 , Gray PMID: 22675532) to demonstrate the major source of the contamination that occurs during disaggregation. Within the macrophage gate in any tissue digest there are unrelated cells coated with macrophage remnants. These are not recognised or excluded as doublets. So, whilst there may be 98% of cells positive for the macrophage-specific surface markers by FACS, it is still the case that there will be much greater contribution of unrelated cells to the total mRNA pool. The only way to reduce this contribution is to introduce a negative selection into the purification.

I suggest the authors tone down their assertions of purity and acknowledge that contamination is an issue, and is a likely explanation for some apparent tissue-specific detection where macrophages in a specific tissue appear to express proteins that are known to be very abundant only in that tissue or in unrelated cells (e.g Alb in liver, Villin in gut, Vwf in endothelial cells)

Reviewer #4 (Remarks to the Author):

The authors did significant works trying to address the reviewers' concerns. Unfortunately, this referee was not convinced by the data and rational.

Reviewer #5 (Remarks to the Author):

Response to reviewer 2 -

Q1 – data availability looks adequate and the web portal that has been established is straightforward to use and contains some valuable features.

Q2 – FDR of 1% is a fairly standard setting for protein and peptide cut-off. The authors state that proteins were filtered to only include those with at least 1 unique peptide. This needs to be explained more since a protein identified with a single peptide would normally be considered a weak hit. I would like to see a scoring introduced for each protein identified, so that people can clearly see whether an identification is of low, medium or high confidence.

Q3 – The authors state that null values in the datasets were set as 1 to facilitate analysis. If I understand this correctly then essentially all missing values have been replaced by 1. This is quite a crude way to deal with missing values and will lead to huge fold change differences when proteins were not detected in one population. A more accurate representation of the data would be to leave the null values and be clear that some proteins were detected in a presence/absence profile.

Other concerns

The methods for calculating copy numbers are not clear. Did the authors use iBAQ intensities to calculate copy numbers? If this is the case then this needs to be justified. Raw intensity would normally be used for copy number estimates.

In addition, I don't understand the logic of the following procedure:

Line 915: Fraction of total (FOT), a relative quantification value that was defined as the iBAQ of a particular protein divided by the total iBAQ of all identified proteins in one experiment, was calculated as the normalized abundance of the particular protein so that its abundance can be compared across experiments. Finally, the FOT was further multiplied by $1E7$ for the ease of presentation and FOTs less than $1E7$ were replaced with $1E7$ to adjust extremely small values.

Why did the authors do this? Given that copy numbers were calculated, I don't understand the logic for this step. If the proteomics data is of high quality then copy numbers can be compared between cell populations.

Line 857 to 881 – why did the authors fractionate some samples and not others? This will lead to differences in data depth which in turn will impact copy number estimates. This will make comparing between samples less reliable. I would like to understand why the authors fractionated only some samples.

Figure 1d – the labelling of transcript and protein is not correct. Also, I do not understand how the authors calculated transcript copies per cell? I would like this to be explained.

The rationale for doing absolute protein quantification using QconCAT is not clear. Why did the authors do this for the cell line? What does this add?

Point-to-point Response

Reviewer #3 (Remarks to the Author):

Q – The authors have responded comprehensively to my comments and have greatly improved the manuscript and the utility of the resource. However, there is one remaining area where I disagree with their response.

In the discussion the authors acknowledge the possibility of contamination.

"Another factor affecting protein or transcript quantification is cell contamination. Meta-analysis performed by Summer et al. indicated that the published macrophage transcriptomic datasets were influenced by extensive contamination of isolated preparations with other cell types straightforwardly or co-purification of cell types that may interact with MPS cells in vivo²²."

(note, name should be Summers.)

However, the authors have not really appreciated the nature of contamination and they have not eliminated it. None of the genes shown in Figure CL15 is actually expressed by macrophages. To take just one example, Villin is clearly epithelial cell restricted in the gut and villin- reporter and villin-cre mice have been generated and very widely-used (e.g. PMID: 12065599). Notably, villin-cre did not delete fl/fl *Csf1r* in the intestine (PMID: 29593242)

Obviously, endocytosis of apoptotic cells can provide one source of contamination and is probably important in the gut. But a recent paper by Millard et al. (PMID: 34818538) builds on earlier studies (Lynch PMID: 29607532, Gray PMID: 22675532) to demonstrate the major source of the contamination that occurs during disaggregation. Within the macrophage gate in any tissue digest there are unrelated cells coated with macrophage remnants. These are not recognised or excluded as doublets. So, whilst there may be 98% of cells positive for the macrophage-specific surface

markers by FACS, it is still the case that there will be much greater contribution of unrelated cells to the total mRNA pool. The only way to reduce this contribution is to introduce a negative selection into the purification.

I suggest the authors tone down their assertions of purity and acknowledge that contamination is an issue, and is a likely explanation for some apparent tissue-specific detection where macrophages in a specific tissue appear to express proteins that are known to be very abundant only in that tissue or in unrelated cells (e.g Alb in liver, Villin in gut, Vwf in endothelial cells)

Response: Thanks for the constructive comments. During the revision process, through careful consultation of the published literature, we have also become aware of the natural and inevitable nature of the existence of contamination. **We therefore fully agree with the reviewer's view. In line with the reviewer's suggestion, to appreciate the influence of the issue in our current study, we toned down the assertions of cell purity in the text and acknowledged the naturally occurring contamination and its possible effects. The causes of the contamination (such as specific proteins common in certain tissues identified in macrophages isolated from that same tissue) and the subsequent influence on the results of the study were also further discussed.**

The cell contamination may be derived from the following factors:

First of all, as far as flow cytometry is concerned, the technology itself, due to unavoidable technic defects such as fluorescence drift, fluorescence quenching after sorting, or cell fragmentation stimulated by high pressure during sorting, can cause some signal loss during back-testing post purification¹⁻⁴. In the published literature carried out by Xiaochun Liu et. al, Emily J. Lelliott et. al, Immanuel Kwok et. al, Fan Zhang et. al, and Hannah Van Hove et. al, Human-derived Langerhans cells, CD8+ T cells, eosinophils, synovial cells, and brain macrophages were sorted through fluorescence-activated cell sorter (FACS), with the purity of 98%⁵, 95%⁶, 98%⁷, 95%⁸, and 90%⁹ respectively. In our current study, we have demonstrated by back-testing based on 10 different markers that we obtained macrophage populations with a purity of 98%, which according to the

previous literature already represents a high purity. Therefore, the contamination is not straightforwardly resulted from purely technical reasons.

More importantly, as the reviewer notes, the following three naturally occurring sources of contamination cannot be completely circumvented due to the characteristics of macrophages: First, the contamination derives from phagocytosis by macrophages of dying (senescent/apoptotic) cells, in which RNA/protein from the engulfed cell may be detected. Second, the perturbation that arises from unrelated cells coated by macrophage remnants during tissue digestion. Third, the tight interactions between macrophages and other cell populations also give rise to the consequence. In a systematic meta-analysis of the macrophage transcriptome, Summers *et.al* provided an in-depth analysis and discussion of the reasons and cell types for contamination of macrophage populations¹⁰. the study suggested that such contamination occurs in almost all cellular material in the macrophage transcriptome research. To address the perturbation of the data brought about by the above factors, we need to rely on subsequent technological advances and the discovery of additional markers.

To assess the degree of possible cell contamination in our study, we referred to the cell-type signature matrix containing 3,028 markers and sc-RNAseq profiling of 894 mouse cell-types clusters from the database Mouse Cell Atlas (MCA, <http://bis.zju.edu.cn/MCA/atlas.html>)¹¹. Based on the reference matrix, we used scMCA^{11, 12}(**Figure CL1**), Gene Set variation analysis (GSVA)¹³ (**Figure CL2**) and CIBERSORTx¹⁴ (**Figure CL3**) respectively to assess the performance of each cell type in each macrophage protein profile dataset in our study.

scMCA demonstrates the most relevant cell types between the reference database and the test dataset (R packages: scMCA)^{11, 12}. The scMCA results suggested that both transcriptomic and proteomic data in our study were highly correlated with the signature matrix of myeloid cells in the scMCA reference database, and showed the strongest correlation with the corresponding macrophage reference signature profiles, in contrast to a lower correlation with tissue parenchymal or epithelial cells (**Figure CL1**). Furthermore, GSVA gives the enrichment degree of all cell type signature genes in a given tissue contained in the scMCA database for the tested dataset (R package: GSVA)¹³. Similarly, the GSVA results suggested that the expression profiles of the individual macrophage

types in our study were significantly enriched in the corresponding cell type signature gene sets of the scMCA reference database, and to a lesser enrichment degree in the parenchymal or epithelial cell reference gene pools (**Figure CL2**). To further assess the extent/likelihood of other cell contamination, CIBERSORTx, a digital cytometer (relative to flow cytometry nomenclature), was used to calculate the relative proportion of each cell type in the tissue for test datasets (online tool: <https://cibersortx.stanford.edu>)¹⁴. As shown in **Figure CL3**, our transcript expression profile better characterized the macrophages (with an average indicated percentage of 95%). Even with the low estimate percentage scores, the possible contamination was likely to be derived from the following cell types: Astrocytes in the brain (for microglia datasets), Endothelial cells in the lung (for alveolar macrophage), Erythroblast/ Hepatocyte cells in the liver (for Kupffer cells), Epithelial cells in the intestine (for intestinal macrophages). **Combining the flow cytometry data and the above digital assessment results, we inferred that tissue parenchyma cells, endothelial cells, or epithelial cells may have contributed to some source of contamination, but their proportion was low.**

Figure CL1. Heatmap shows the scMCA results of the proteome (A) and transcriptome (B) datasets across 12 macrophage populations. The top three cell types (in the reference database) with the highest Pearson correlation coefficients between scMCA reference database and each macrophage expression profile are shown. Each row represents one cell type in scMCA reference. Each column represents one experiment in proteome or transcriptome dataset. Red means high correlation; gray means low correlation.

Figure CL2. Heatmap of the GSEA scores (enrichment statistics, ES) of cell-type reference signatures of indicated tissues in scMCA signature matrix (y-axis) for transcriptome (A) or proteome (B) profiling of different macrophage populations. Only the macrophage types contained in the scMCA database were analyzed and shown. Each row represents one cell type in scMCA reference. Each column represents one experiment in the proteome or transcriptome dataset. Red means high GSEA score; blue means low GSEA score. Profiling for microglia, alveolar macrophages, Kupffer cells, small & large intestinal macrophages, BMDM were analyzed against cell-type reference signatures of the brain, lung, liver, small intestine and bone marrow from scMCA database.

Figure CL3. The stacked bar plot shows the estimated percentage of CIBERSORTx cell type enrichments based on scMCA reference datasets for macrophage transcriptome. Only the cell types with an estimated percentage > 0 were shown in the plot. Only the macrophage types contained in the scMCA database were analyzed and shown. Profiling for microglia, alveolar macrophages, Kupffer cells, small & large intestinal macrophages, and BMDM were analyzed against cell-type reference signatures of the brain, lung, liver, small intestine and bone marrow from scMCA database.

To this end, we revised the results and discussion part of the manuscript.

In the result (in Page 5 of the revised manuscript), we tone down the affirmation of cell purity and acknowledge the inevitable nature of the existence of contamination due to factors such as phagocytosis and encapsulation:

Evidence obtained by back-testing through flow cytometry suggested that the average purity of the macrophage populations was 98%. However, it is worth noting that the possible contamination due to factors such as phagocytosis activity of macrophages and encapsulation of other cells by macrophage remnants during tissue digestion is inevitable.

Also, in Page 10 of the manuscript, we have included the following description to explain the effect of such contamination on the results of tissue-specific detection (where macrophages in a specific tissue appear to express proteins that are known to be very abundant only in that tissue or in unrelated cells):

Undeniably, the ‘contamination’ derived from phagocytosis activity and re-encapsulation of unrelated cells during macrophage isolation may also partly explain the existence of abundant tissue-specific detection in different macrophage populations, such as Alb in the liver, Villin (Vil1) in the gut, Vwf in endothelial cells.

In Page 22 of the Discussion section, we provide an expanded discussion of the issue as follows:

The possible interference or contamination by unrelated cells can also impact mRNA/protein quantification results as well as subsequent data analysis, which is an inevitable limitation of the current study. First, the contamination derives from the phagocytosis activity of macrophages, in which RNA/protein from the engulfed cell may be detected. Second, the perturbation that arises from unrelated cells coated by macrophage remnants during tissue digestion. Third, the tight interactions between macrophages and other cell populations also give rise to the consequence. These three phenomena cannot be recognized or excluded by doublets gating in flow cytometry analysis. Such contamination can partly explain the phenomenon that specific proteins common in certain tissues may then be found in macrophages isolated from that same tissue. Also, the issue may influence the WGCNA-based gene cluster and functional analysis results in our study. Therefore, it becomes worthwhile to explore the effects of perturbations derived from different experimental procedures on gene expression profiles of primary cells in subsequent studies.

We greatly appreciate the seriousness of the reviewer to the issue of cell contamination and sincerely thank the reviewer for the careful and precise guidance on this work.

We have corrected the names of the authors (Summers) of the meta-analysis in the revision, please see Page 22 of the manuscript. We thank the reviewer for the erratum.

Reviewer #4 (Remarks to the Author):

The authors did significant works trying to address the reviewers' concerns. Unfortunately, this referee was not convinced by the data and rationale.

Q – This author does not believe the claim that the quantified protein copy numbers spanned over nine orders of magnitude. This conclusion therefore needs to be validated by an orthogonal method.

Response: During the revision process, we noted that the copy number of proteins in our proteome data ranged from 1 to 10^8 copies per cell, spanning a total of 8 orders of magnitude actually, whereas in the previous manuscript we mistakenly wrote that it spanned 9 orders of magnitude. We have corrected the presentation as “The quantified protein copy numbers spanned over eight orders of magnitude” in the revised manuscript (Page 6). We explain the result that the proteome spanning over 8 orders of magnitude as follows.

In the previous round of revisions, we carried out a QconCAT-based absolute quantification approach to demonstrate the confidence of data for protein copy numbers spanning eight (misspelled as nine previously) orders of magnitude. Probably because the QconCAT-based quantification method is still a mass spectrometry-based protein quantification technique rather than an orthogonal method, the reviewer commented that we had not fully answered the question. However, given the inappropriateness of the antibody-based experimental method in absolute copy number quantifications, we considered that there is not a suitable experiment to validate the range. In this round of revisions, by reviewing the published literatures discussing the dynamic range of the proteome, we found that a dynamic range spanning eight orders of magnitude is credible and is a frequent occurrence in previous studies.

We followed the ‘proteomic ruler’ approach for protein quantification. This approach was initially introduced by Prof. Matthias Mann’s group to quantify the proteome data of a series of cell lines¹⁵. Based on the assumption that the total mass of histones is approximately equal to the total mass of DNA, this approach used the total intensity of histones in each sample to estimate the copy number

per cell for every protein. Thus, the order of magnitude of the copy number reflects the dynamic range of the mass spectrum.

Prof. Matthias Mann has applied this approach extensively to subsequent proteomic studies. For example, in a proteomics study by Prof. Matthias Mann's group exploring the interactions network of myeloid and lymphoid cells in human whole blood, the authors used the 'proteomics ruler' approach to calculate the protein copy number of various cell types (*Nature immunology*, 2017, PMID: 28263321)¹⁶. Through examining the raw data of the study, we found that the dynamic range of protein copy numbers for certain cell types like neutrophils, monocytes, and dendritic cells in this study also spanned over eight orders of magnitude, ranging from a few to hundreds of millions (**Figure CL4A**)¹⁶. In another study, Prof. Matthias Mann performed LC-MS/MS analysis on five cell populations in the liver and calculated protein copy numbers using the 'proteomics ruler' approach, with protein copy numbers of Kupffer cells spanning eight orders of magnitude (**Figure CL4B**, *Cell Metabolism*, 2014, PMID: 25470552)¹⁷.

Figure CL4. The dynamic range of the proteomes of measured cell lineages (A) and Kupffer cells (B) performed by published researches^{16, 17}, based on copy number values per cell.

The causes and consequences of the large dynamic range in mass spectrometry data have also been discussed in previous literature^{18, 19}. The authors commented 'One of the most significant differences between transcriptomics and proteomics is in the dynamic range of mRNA and protein concentrations inside the cell'¹⁹. **The dynamic range of the cellular proteome approaches seven orders of magnitude—from one copy per cell to ten million copies per cell (The dynamic range of proteins expressed in body fluids even exceeds ten orders of magnitude), while the mRNA**

dynamic range covers only three or four orders of magnitude (*Nature*, 2011, PMID: 21593866)²⁰. Moreover, the dynamic range of signal intensities of peptides resulting from the proteome's digestion is at least an order of magnitude larger than that of the original proteome in shotgun proteomics¹⁹. Factors such as protein half-life, protein modification during sample preparation, mass spectrometry sensitivity, and protein coverage all contribute to the large dynamic ranges of protein spectra.

Consistently, the average level of protein copy number in our data across samples spanned eight orders of magnitude, ranging from 1 copy to hundreds of millions (nE8) copies. This dynamic range is in line with previous studies. **To avoid misunderstandings, we cited the above two papers from Prof. Matthias Mann's group in the manuscript to support the credibility of our proteomic dynamic range during the revision process. Please see Page 6 in the manuscript for detail.**

Reviewer #5 (Remarks to the Author):

Response to reviewer 2-

Q1 – data availability looks adequate and the web portal that has been established is straightforward to use and contains some valuable features.

Response 1: Thanks for the comments and appreciation.

Q2 – FDR of 1% is a fairly standard setting for protein and peptide cut-off. The authors state that proteins were filtered to only include those with at least 1 unique peptide. This needs to be explained more since a protein identified with a single peptide would normally be considered a weak hit. I would like to see a scoring introduced for each protein identified, so that people can clearly see whether an identification is of low, medium or high confidence.

Response 2: Thanks for the helpful comments. The score thresholds for q-value divided the identified proteins into high ($q < 0.01$), medium ($0.01 < q < 0.05$), and low confidence ($q > 0.05$) identifications. In the previous manuscript, only proteins with high confidence identifications were used for subsequent analysis. **In line with the reviewer's comments, we have updated Supplementary Table 1, giving the relevant score (Exp. q-value and protein score) and confidence (of low, medium or high) for each identified protein in each experiment in Sheet 2 of Supplementary Table 1.**

We paid much attention to the mentioned single peptide issue and applied the more stringent criteria (unique peptides ≥ 2 per protein) to minimize the impact of low-intensity peptides quantification. Based on protein patterns with the criterion of unique peptides ≥ 2 , we re-performed the weighted gene co-expression network analysis (WGCNA) to identify cell-type-specific modules (CTMs) and proteins (with the same cut-off as in the previous manuscript, i.e., Gene Significance (GS) ≥ 0.6 , $p.GS < 0.05$) and Module Membership (MM) ≥ 0.5 , $p.MM < 0.05$). Because of the stringent criteria for protein filtration, the number of specific proteins in CTMs of each macrophage is slightly reduced, but the proteins in each population were still similar to previous, with an overlap higher than 85% (**Figure CL5**). With similar gene sets, the functional enrichment analysis on the filtered

proteins in CTMs also remains the same. As shown in **Figure CL6B**, for example, genes in the CTMs of alveolar macrophages were enriched in the ‘response to oxidative stress’ and ‘regulation of lipid metabolism’, the CTMs of Kupffer cells were enriched in ‘fatty acid/xenobiotic metabolic process’ and ‘blood coagulation’, the CTMs of intestinal macrophages were characterized by the ‘mucosal immune response’ and ‘leukocyte migration’. The same results were also captured in **Figure CL6A** based on the protein pattern with the criterion of unique peptides ≥ 1 per protein. **The results suggested that based on the stress cut-off screening conditions (such as $GS \geq 0.6$, $p.GS < 0.05$, $MM \geq 0.5$, $p.MM < 0.05$) during data analysis processes, the setting of the unique peptide threshold did not have a significant effect on the results in our current study.**

Figure CL5. Venn diagram of the numbers of proteins with the cut-off of Gene Significance ($GS \geq 0.6$ ($p.GS < 0.05$) and Module Membership ($MM \geq 0.5$ ($p.MM < 0.05$)) in cell-type specific modules (CTMs) determined by WGCNA on protein patterns with the criteria of unique peptides ≥ 1 (Blue) and unique peptides ≥ 2 (Red).

Figure CL6. Representative functional annotations (GOBP databases, Fisher's exact test) for genes in the CTMs of indicated macrophage populations in Figure CL5, based on protein patterns with the criteria of unique peptides ≥ 1 (A) and unique peptides ≥ 2 (B). The dot size represents the number of proteins involved in the relevant term. The color bar indicates the enrichment significance.

Q3 – The authors state that null values in the datasets were set as 1 to facilitate analysis. If I understand this correctly then essentially all missing values have been replaced by 1. This is quite a crude way to deal with missing values and will lead to huge fold change differences when proteins were not detected in one population. A more accurate representation of the data would be to leave the null values and be clear that some proteins were detected in a presence/absence profile.

Response 3: Thanks for the comments. We apologized for the unclear description. In our research, proteins with less than 40% of missing values in one experiment were selected, and missing values were then imputed with the minimum value of the proteomic data, following the previous published work^{21, 22}. To avoid systematic bias in comparative analysis among different macrophage populations, we utilized two tails student *t*-test. The significance of differential expression proteins was determined based on a *p*-value cut-off of 0.05.

Following the reviewer's suggestion, we replaced all the missing values with NA, and re-performed

the same analysis as illustrated in Figure 5c-f (in which the proteins were selected according to the cutoff based on fold change and p -value). For proteins identified in both populations, the criteria for the selection of differential expression proteins (DEPs) remained the same as in the previous manuscript: i.e., Fold change ≥ 5 & p -value < 0.05 (two tails student t-test). For proteins for which the fold change or p -value could not be calculated due to null values, the proteins that were stably identified three times (with Coefficient of variation value $CV \leq 0.6$) in one population but not identified in another population were selected as DEPs.

As shown in the Venn diagram, the significantly altered proteins were similar, with an overlap higher than 95% (**Figure CL7A**). The results of the functional enrichment analysis of DEPs also remain the same (**Figure CL7B, C**), except those individual genes, as in **Figure CL8**, were not included (because of their spark expression in one population). For example, proteins that enriched in iron ion homeostasis, cholesterol transport, and LPS-mediated signaling processes/pathways, were dominantly expressed in Kupffer cells, whereas proteins that participated in leukocyte migration, and cell adhesion pathways showed elevated expression in liver-recruited macrophage population, under the two data processing procedures (**Figure CL7B, C**). **To avoid misunderstanding, we followed the reviewer's suggestion, left the null values as NA in Supplementary Table 1, Supplementary Table 7, and Supplementary Table 9-11, and presented missing values clearly with grey blocks with a cross in heatmap or mosaic plots, including Figure 2e, Figure 4e, Figure 5d, Figure 5f, and Figure 6b, Supplementary Figure 6b, Supplementary Figure 7a, Supplementary Figure 10c-d. The methods for DEPs selection based on the datasets with Null values were also updated, please see Page 34 in the Materials and Methods of the revised manuscript.**

Figure CL7. Comparison of differential expression proteins (DEPs) and their functions in macrophage proteome datasets with or without null value (NA). **(A)** Venn diagram of the numbers of DEPs for tissue-resident macrophage and tissue-recruited macrophage populations in the protein patterns with (red) or without (blue) null value (NA), in the liver (left) and lung (right). The DEPs in protein patterns without NA were selected with the cut-off of Fold change ≥ 5 & p .value < 0.05 (two tails student t -test). In protein patterns with NA, the proteins identified in all populations with the cut-off of Fold change ≥ 5 & p .value < 0.05 (two tails student t -test), as well as the proteins stably identified in all three repeats in one population (with Coefficient of variation value $\text{CV} \leq 0.6$)

but not in another, were selected as DEPs. **(B, C)** Representative functional annotations (GOBP databases, Fisher's exact test) for DEPs of tissue-resident macrophage and tissue-recruited macrophage populations, in the liver (left) and lung (right), based on protein patterns with **(B)** or without **(C)** null value, respectively. The dot size represents the number of proteins involved in the relevant term. The color bar indicates the enrichment significance.

Figure CL8. Expression patterns of proteins successfully identified as DEPs participating in functions in Figure CL7 in proteome data without NA but not in proteome data with NA, in the liver (left) and lung (right). Values for each protein in all populations are color-coded based on the z-scored copy numbers per cell. The grey blocks with a cross represent missing values.

Other concerns

Q4 – The methods for calculating copy numbers are not clear. Did the authors use iBAQ intensities to calculate copy numbers? If this is the case then this needs to be justified. Raw intensity would normally be used for copy number estimates.

Response 4: Thanks for the constructive comments. We used iBAQ intensities for copy number calculation. Consistent with the reviewers' comments, according to the original research on the 'proteomic ruler' approach for protein quantification introduced by Prof. Matthias Mann's group¹⁵, the protein copy number was estimated based on the Raw intensity. Prior to proposing the 'proteomic ruler' approach, Prof. Matthias Mann's group invented a total protein approach (TPA) to estimate the absolute copy number of proteins in one sample²³. In the research, the authors demonstrated that a fractional value of the MS signal (LFQ intensity) of a protein compared with the total MS signal is a good proxy of the percentage of its protein mass to total protein mass. This

can then be converted into numbers of molecules per cell by measuring or estimating the volume and protein content of the analyzed cells. Based on the TPA, Prof. Matthias Mann introduced the ‘proteomic ruler’ approach (the MS signal of histones can be used as a “proteomic ruler” because it is proportional to the amount of DNA in the sample, which in turn depends on the number of cells.) to calculate protein copy number per cell directly from deep eukaryotic proteome datasets without any additional experimental steps. **They showed that both LFQ intensity and iBAQ intensity were highly correlated with the concentration of standard proteins (UPS2; Sigma)²³ (Figure CL9).** In addition, they applied the iBAQ values to estimate protein copy numbers in a study entitled ‘cell type- and brain region-resolved mouse brain proteome’ (*Nature Neuroscience*, 2015, PMID: 26523646)²⁴. To evaluate the accuracy of iBAQ intensity-based copy number values in our current study, we also calculated the copy number using the raw intensity. **The result demonstrated that the iBAQ and intensity-based copy number were highly correlated with each other (with the average Pearson correlation coefficient $r = 0.94$) (Figure CL10, also see Page 34 and Supplementary Figure 12 in the revised manuscript).**

Figure CL9. TPA calculation applied to analysis of a mixture of standard proteins (UPS2; Sigma). Protein standards were solubilized in SDS containing buffer and processed with the FASP method using trypsin. The digest was analyzed by LC-MS/MS using 4 h acetonitrile gradient. The protein concentrations were calculated either using directly protein intensities or iBAQ values. Related to Figure 7A of the published literature²³.

Figure CL10. The point charts show the Pearson correlation coefficients between the copy number values based on iBAQ (x-axis) and raw intensity (y-axis) in 36 proteome patterns for the 12 macrophage populations in our current study.

In addition, I don't understand the logic of the following procedure:

Q5 – Line 915: Fraction of total (FOT), a relative quantification value that was defined as the iBAQ of a particular protein divided by the total iBAQ of all identified proteins in one experiment, was calculated as the normalized abundance of the particular protein so that its abundance can be compared across experiments. Finally, the FOT was further multiplied by $1E7$ for the ease of presentation and FOTs less than $1E7$ were replaced with $1E7$ to adjust extremely small values. Why did the authors do this? Given that copy numbers were calculated, I don't understand the logic for this step. If the proteomics data is of high quality then copy numbers can be compared between cell populations.

Response 5: Thanks for the comments. According to the principle of copy number calculation, as well as the comments on the non-applicability of copy number for tissue samples in the original publication, (the original description is 'in tissues, not only are cell sizes variable, but visual

counting of cells is also problematic'), the copy number is only applicable to the quantification of cellular samples, but not to sample materials such as tissues, body fluids, or culture supernatant¹⁵.

Therefore, in this study, we performed copy number calculations on protein profiles of macrophage cells, while applying FOT to the analysis of tissue expression profiles. The strategy of performing different normalizations on data from different sample types for subsequent analysis was also applied in the published researches. For example, in a proteomics study by Prof. Matthias Mann's team exploring the interactions network of myeloid and lymphoid cells in human whole blood, the authors used a copy number approach to quantify various cell types and a median-based normalization approach to quantify secreted proteins (*Nature immunology*, 2017, PMID: 28263321)¹⁶.

In our study, no direct parallel comparison was made between the macrophage data based on copy number and the tissue data from the FOTs quantification algorithm, which was only used to construct the cell-tissue interaction network (in Figure 4).

We apologize for the absence of a detailed sub-declaration of data normalization for cell and tissue samples in material methods. **During the revision process, to avoid misunderstandings, we delineated separately the data normalization approach for macrophage populations (based on the copy-number approach) and tissues (based on FOTs values) in the materials and methods. Please see page 34 in the revised manuscript.**

Q6 – Line 857 to 881 – why did the authors fractionate some samples and not others? This will lead to differences in data depth which in turn will impact copy number estimates. This will make comparing between samples less reliable. I would like to understand why the authors fractionated only some samples.

Response 6: In this study, given the small number of cells obtained by fluorescence-activated cell sorter (FACS) and the time duration taken to obtain them, a small number of macrophage materials available were used to obtain the deep coverage of protein expression profiles through sample fractionation (into 6 fractions). For tissue samples, where the material itself is easily accessible and

of high quality, deeper coverage of protein expression profiles could also be obtained by a single-shot, long-gradient experimental strategy.

The fraction-based macrophage data were not directly compared in parallel with the single-run tissue data, but were only used to construct a cell-tissue interaction network (in Figure 4), so the difference in coverage depth due to sample fractionation did not affect the results.

During the revision process, to avoid misunderstandings, we have described the process of cell and tissue sample preparation separately in the materials and methods. We explicitly stated that cells underwent an sRP-based gradient separation approach, while for tissue samples a single-shot mass spectrometry analysis strategy was used. Please see pages 30 and 31 of the revised manuscript.

Q7 – Figure 1d – the labelling of transcript and protein is not correct. Also, I do not understand how the authors calculated transcript copies per cell? I would like this to be explained.

Response 7: Thanks for the comments. We apologize for the mislabeling of datasets in Figure 1d. The label has been corrected during the revision process.

For transcript copy number calculation, we followed the ‘proteomic ruler’ approach for protein quantification, as stated above. This approach was also initially introduced by Prof. Matthias Mann’s group to quantify the proteome data of a series of cell lines (including A549, Hep-G2, PC-3, etc.)¹⁵. On the basis of the assumption that the total mass of histones is approximately equal to the total mass of DNA, this approach used the total intensity of histones in each sample to estimate the copy number per cell for every protein. **In this published study, Prof. Matthias Mann also proved that ribosomal proteins can be served as a ruler for estimating cellular RNA copy number per cell.** Ribosomal RNA typically represents about 80% of total RNA²⁵, and in eukaryotic ribosomes, there is a ratio of about 1:1 between RNA and protein²⁶. According to the ribosomal proteins’ total copy number, the total cellular RNAs’ copy number can be calculated. Then, with the FPKM (Fragments Per Kilobase of transcript per Million mapped reads) value in the RNA-seq datasets, we can calculate the ratio of each RNA to the total cellular RNAs’ copy number. Thus, we

calculated each RNA's copy number per cell. This method was also used in Prof. Matthias Mann's study to estimate the RNA's copy number per cell (*Cell Metabolism*, 2014, PMID: 25470552)¹⁷.

During the revision process, we further described the method for copy number calculation for transcripts in the materials and methods. Please see page 36 in the manuscript for details.

Q8 – The rationale for doing absolute protein quantification using QconCAT is not clear. Why did the authors do this for the cell line? What does this add?

Response 8: In the last round of revision, the reviewer asked us to validate the dynamic range spanning eight orders of magnitude, so we carried out additional absolute protein quantification studies based on QconCAT on BMDM and RAW264.7 cell line. The result further demonstrated the reliability of eight orders of magnitude of protein quantification. However, we subsequently found that the principle of QconCAT, which is also based on LC-MS/MS techniques (not an orthogonal method), did not support the dynamic range of the mass spectra, so **we removed these QconCAT experimental data and related descriptions during the revision process.**

Thanks again for the constructive and precise comments and suggestions.

References:

1. Box, A. *et al.* Evaluating the Effects of Cell Sorting on Gene Expression. *J Biomol Tech* (2020).
2. Burel, J.G. *et al.* The Challenge of Distinguishing Cell-Cell Complexes from Singlet Cells in Non-Imaging Flow Cytometry and Single-Cell Sorting. *Cytometry. Part A : the journal of the International Society for Analytical Cytology* **97**, 1127-1135 (2020).
3. Jayasinghe, S.N. Reimagining Flow Cytometric Cell Sorting. *Adv Biosyst* **4**, e2000019 (2020).
4. Song, X. *et al.* Improved strategy for jet-in-air cell sorting with high purity, yield, viability, and genome stability. *FEBS Open Bio* **11**, 2453-2467 (2021).
5. Liu, X. *et al.* Distinct human Langerhans cell subsets orchestrate reciprocal functions and require different developmental regulation. *Immunity* (2021).
6. Lelliott, E.J. *et al.* CDK4/6 inhibition promotes anti-tumor immunity through the induction of T cell memory. *Cancer Discov* (2021).
7. Kwok, I. *et al.* Combinatorial Single-Cell Analyses of Granulocyte-Monocyte Progenitor Heterogeneity Reveals an Early Uni-potent Neutrophil Progenitor. *Immunity* **53**, 303-318 e305 (2020).
8. Zhang, F. *et al.* Defining inflammatory cell states in rheumatoid arthritis joint synovial tissues by integrating single-cell transcriptomics and mass cytometry. *Nat Immunol* **20**, 928-942 (2019).
9. Van Hove, H. *et al.* A single-cell atlas of mouse brain macrophages reveals unique transcriptional identities shaped by ontogeny and tissue environment. *Nat Neurosci* **22**, 1021-1035 (2019).
10. Summers, K.M., Bush, S.J. & Hume, D.A. Network analysis of transcriptomic diversity amongst resident tissue macrophages and dendritic cells in the mouse mononuclear phagocyte system. *PLoS Biol* **18**, e3000859 (2020).
11. Han, X. *et al.* Mapping the Mouse Cell Atlas by Microwell-Seq. *Cell* **172**, 1091-1107 e1017 (2018).
12. Sun, H., Zhou, Y., Fei, L., Chen, H. & Guo, G. scMCA: A Tool to Define Mouse Cell Types Based on Single-Cell Digital Expression. *Methods Mol Biol* **1935**, 91-96 (2019).
13. Hanzelmann, S., Castelo, R. & Guinney, J. GSEA: gene set variation analysis for microarray and RNA-seq data. *BMC Bioinformatics* **14**, 7 (2013).
14. Steen, C.B., Liu, C.L., Alizadeh, A.A. & Newman, A.M. Profiling Cell Type Abundance and Expression in Bulk Tissues with CIBERSORTx. *Methods Mol Biol* **2117**, 135-157 (2020).
15. Wisniewski, J.R., Hein, M.Y., Cox, J. & Mann, M. A "proteomic ruler" for protein copy number and concentration estimation without spike-in standards. *Mol Cell Proteomics* **13**, 3497-3506 (2014).
16. Rieckmann, J.C. *et al.* Social network architecture of human immune cells unveiled by quantitative proteomics. *Nat Immunol* **18**, 583-593 (2017).
17. Azimifar, S.B., Nagaraj, N., Cox, J. & Mann, M. Cell-type-resolved quantitative proteomics of murine liver. *Cell metabolism* **20**, 1076-1087 (2014).
18. Wu, L. & Han, D.K. Overcoming the dynamic range problem in mass spectrometry-based

- shotgun proteomics. *Expert Rev Proteomics* **3**, 611-619 (2006).
19. Zubarev, R.A. The challenge of the proteome dynamic range and its implications for in-depth proteomics. *Proteomics* **13**, 723-726 (2013).
 20. Schwanhauser, B. *et al.* Global quantification of mammalian gene expression control. *Nature* **473**, 337-342 (2011).
 21. Xu, J.Y. *et al.* Integrative Proteomic Characterization of Human Lung Adenocarcinoma. *Cell* **182**, 245-261 e217 (2020).
 22. Zhou, Q. *et al.* A mouse tissue transcription factor atlas. *Nat Commun* **8**, 15089 (2017).
 23. Wisniewski, J.R. *et al.* Extensive quantitative remodeling of the proteome between normal colon tissue and adenocarcinoma. *Mol Syst Biol* **8**, 611 (2012).
 24. Sharma, K. *et al.* Cell type- and brain region-resolved mouse brain proteome. *Nat Neurosci* **18**, 1819-1831 (2015).
 25. Warner, J.R. The economics of ribosome biosynthesis in yeast. *Trends Biochem Sci* **24**, 437-440 (1999).
 26. Melnikov, S. *et al.* One core, two shells: bacterial and eukaryotic ribosomes. *Nat Struct Mol Biol* **19**, 560-567 (2012).

REVIEWERS' COMMENTS

Reviewer #4 (Remarks to the Author):

This referee has serious doubt with the claims this paper made. Specifically,
1. 8 orders of dynamic range: So, let's say we want to detect a spectrum in which we see two peaks, i.e. one is 100%, and on other is 0.000001% (8 orders of magnitude). If both came from representing peptides, we need at least 10^9 ions for peptide 1 and at least 10 ions for peptide 2 (to detect with S/N ~ 10).

Presently we cannot do this because

1. AGC max for ion detector is around 10^6 ions.

2. We cannot send 10^9 ions into a c-trap, because of ion losses while transmitting ESI ions to it (that is a typical number of ions after the heated capillary).

Ok, one would claim that it is a combination of HPLC with a mass spectrometry that may enable us to detect that dynamic range. Then one must load onto a column say 50 attomoles of peptide 2, and $10^8 \times 50$ amoles of peptide 1 (50nmole! Which, in case of 50kDa protein is 250ug!). Column overload!

Additionally, to my memory, the histones, likely most abundant proteins in cells, have around millions of copies per cells. Do we have proteins whose abundance is around 100 million molecules per cell, I doubt.

In sum, the author's claim and rational are very misleading.

2. Proteomics of macrophages from heart.

Single cell proteomics is getting routine. With 100,000 cells, you can easily detect 5000 proteins, which is well within the range of the expertise the authors have. Accordingly, this exp must be done and included in the paper.

Reviewer #5 (Remarks to the Author):

The authors have made clear changes to the manuscript and I feel that these help to clarify the issues that I had.

Point-to-point Response to Reviewer

Reviewer #4 (Remarks to the Author):

This referee has serious doubt with the claims this paper made. Specifically,
1. 8 orders of dynamic range: So, let's say we want to detect a spectrum in which we see two peaks, i.e. one is 100%, and on other is 0.000001% (8 orders of magnitude). If both came from representing peptides, we need at least 10^9 ions for peptide 1 and at least 10 ions for peptide 2 (to detect with S/N ~10).

Presently we cannot do this because

1. AGC max for ion detector is around 10^6 ions.

2. We cannot send 10^9 ions into a c-trap, because of ion losses while transmitting ESI ions to it (that is a typical number of ions after the heated capillary).

Ok, one would claim that it is a combination of HPLC with a mass spectrometry that may enable us to detect that dynamic range. Then one must load onto a column say 50 attomoles of peptide 2, and $10^8 \times 50$ amoles of peptide 1 (50 nmole! Which, in case of 50 kDa protein is 250 μ g!). Column overload!

Additionally, to my memory, the histones, likely most abundant proteins in cells, have around millions of copies per cells. Do we have proteins whose abundance is around 100 million molecules per cell, I doubt.

In sum, the author's claim and rational are very misleading.

Response 1. Thanks to the reviewer's comments. We are aware of and totally agree with the reviewer's comments, the statement about the dynamic range has been removed in the revision.

2. Proteomics of macrophages from heart.

Single cell proteomics is getting routine. With 100,000 cells, you can easily detect 5000 proteins, which is well within the range of the expertise the authors have. Accordingly, this exp must be done and included in the paper.

Response 2. Thanks to the reviewer's comments. Given the important functions of cardiac macrophages in tissue-supporting activities, we have archived the proteomic data of cardiac macrophages based on the sorting strategy of " $CD45^+F4/80^{hi}CD11b^{lo}Ly6g^-Ly6c^-CD11c^{lo}$ " (**Figure CL1**). The dataset now has been included in the online data portal (<http://macrophage.mouseprotein.cn>).

While additional data may be of interests, the authors agree with the editor (in email exchange) that sacrificing 40 mice for this supporting data that is not the main focus of the work may not be optimal considering animal welfare, so will not perform this experiment as requested, but will instead consider it as an integral part of future studies.

Figure CL1. Gating strategy of cardiac macrophage populations.